# Learning Equilibria from Data:
# Provably Efficient Multi-Agent Imitation Learning

**Till Freihaut**[*]
University of Zurich
freihaut@ifi.uzh.ch

**Luca Viano**[*]
EPFL
luca.viano@epfl.ch

**Volkan Cevher**
EPFL
volkan.cevher@epfl.ch

**Matthieu Geist**
Earth Species Project
matthieu@earthspecies.org

**Giorgia Ramponi**
University of Zurich
ramponi@ifi.uzh.ch

## Abstract

This paper provides the first expert sample complexity characterization for learning a Nash equilibrium from expert data in Markov Games. We show that a new quantity named the *all policy deviation concentrability coefficient* is unavoidable in the non-interactive imitation learning setting, and we provide an upper bound for behavioral cloning (BC) featuring such coefficient. BC exhibits substantial regret in games with high concentrability coefficient, leading us to utilize expert queries to develop and introduce two novel solution algorithms: MAIL-BRO and MURMAIL. The former employs a best response oracle and learns an $\varepsilon$-Nash equilibrium with $\mathcal{O}(\varepsilon^{-4})$ expert and oracle queries. The latter bypasses completely the best response oracle at the cost of a worse expert query complexity of order $\mathcal{O}(\varepsilon^{-8})$. Finally, we provide numerical evidence, confirming our theoretical findings.

## 1 Introduction

Learning in systems with multiple agents is common in real-world applications, such as autonomous driving [Shalev-Shwartz et al., 2016], traffic light control [Bakker et al., 2010], and games [Samvelyan et al., 2019]. Designing reward functions in these applications is challenging, as it requires defining multiple, potentially opposing, objectives. However, expert data are often available, making Multi-Agent Imitation Learning (MAIL) an important approach for learning policies that perform well in underlying Markov Games (MGs) with unknown reward functions. MAIL has the potential to ensure the alignment of agents with the original experts' goals and to avoid potentially exploitable policies that can lead to socially undesirable behavior [Hammond et al., 2025].

A key distinction between Multi-Agent Imitation Learning and Single-Agent Imitation Learning (SAIL) is that the performance of a strategy in MAIL depends on the strategies of other agents. This means that an expert need not maximize reward directly; instead, the goal is to reach a state where no agent benefits from unilaterally deviating from its strategy, typically referred to as an equilibrium. The most common equilibrium concept is the Nash equilibrium (NE). To evaluate how close a given strategy is to an NE, the objective must consider strategic deviations of one agent while holding the others fixed.

In this work, we consider 2-player Zero-Sum Markov Games[1] where the agents' rewards are perfectly opposing, i.e., $r_1(s, a, b) = -r_2(s, a, b)$. In this setting, denoting the state value function of a strategy

---

[*]Equal contribution.

[1]For the sake of simplicity, the main text will focus on this case. The appendix outlines the extension to $n$ player general-sum games.

Table 1: For simplicity, we report results for the two-player zero-sum with discount factor $\gamma$, finite state space $|\mathcal{S}|$, finite action spaces $\mathcal{A}$, $\mathcal{B}$. Let $|\mathcal{A}_{\max}| = \max |\mathcal{A}|, |\mathcal{B}|$. Being consistent with Tang et al. [2024], we denote $\beta = \min_{s \in \mathcal{S}} d^{\mu^{\mathrm{E}}, \nu^{\mathrm{E}}}(s)$, by $u$ the recoverability coefficient and with $H$ the finite horizon of the considered game. Moreover, we refer to Tang et al. [2024] for the definition of the convex functions $\ell_{\mathrm{MALICE}}$ and $\ell_{\mathrm{BLADES}}$. Additionally, for the behavioral cloning (BC) output pair $\widehat{\mu}, \widehat{\nu}$ with an input dataset $\mathcal{D}$ we define $\mathcal{C}_{\max} = \max_{\mu,\nu} \max \left\{ \max_{\nu^{\star} \in \mathrm{br}(\mu)} \left\| \frac{d^{\mu, \nu^{\star}}}{\rho} \right\|_{\infty}, \max_{\mu^{\star} \in \mathrm{br}(\nu)} \left\| \frac{d^{\mu^{\star}, \nu^{\mathrm{E}}}}{\rho} \right\|_{\infty} \right\}$, where $\rho$ is any state state distribution, in BC we set it to $\rho = d^{\mu^{\mathrm{E}}, \nu^{\mathrm{E}}}$. For this comparison, notice that the main text by Tang et al. [2024] focuses on learning correlated equilibria, but as specified in their appendix, the same proofs can be performed for the problem of learning Nash equilibria. For the algorithms presented by Tang et al. [2024], we can not specify the bound on the number of expert queries since their analysis as an error propagation only flavor. As a final minor difference, we apply our analysis to the infinite horizon discounted setting, which is more relevant for practical settings. Finally, we abbreviated Queriable Expert by QE.

| Algorithm | MG assum. | Computational Cost | Nash$-$Gap | Expert Data | Required Computational Oracles | QE |
|---|---|---|---|---|---|---|
| BC Tang et al. [2024] | $\beta > 0$ | 0 (analytical solution is available) | $\mathcal{O}\left(uH\varepsilon\beta^{-1}\right)$ | Not specified | $\varepsilon$-accurate TV minimizer | ✗ |
| MALICE Tang et al. [2024] | $\beta > 0$ | $\exp\left(|\mathcal{S}|\right)$ | $\mathcal{O}\left(uH\varepsilon\right)$ | Not specified | $\varepsilon$-accurate $\ell_{\mathrm{MALICE}}$ minimizer | ✗ |
| BLADES Tang et al. [2024] | None | $\exp\left(|\mathcal{S}|\right)$ | $\mathcal{O}\left(uH\varepsilon\right)$ | Not specified | $\varepsilon$-accurate $\ell_{\mathrm{BLADES}}$ minimizer | ✓ |
| **BC (Our analysis)** | $\mathcal{C}_{\max} < \infty$ | 0 (analytical solution is available) | $\mathcal{O}\left(\varepsilon\right)$ | $\widetilde{\mathcal{O}}\left(\frac{|\mathcal{S}||\mathcal{A}_{\max}|\mathcal{C}_{\max}^2}{(1-\gamma)^4\varepsilon^2}\right)$ | None | ✗ |
| **MAIL-BRO (Ours)** | None | $\mathrm{poly}(|\mathcal{S}|, |\mathcal{A}_{\max}|, (1-\gamma)^{-1}, \varepsilon^{-1})$ | $\mathcal{O}\left(\varepsilon\right)$ | $\widetilde{\mathcal{O}}\left(\frac{|\mathcal{S}||\mathcal{A}_{\max}|^2}{(1-\gamma)^4\varepsilon^4}\right)$ | Best response oracle | ✓ |
| **MURMAIL (Ours)** | None | $\mathrm{poly}(|\mathcal{S}|, |\mathcal{A}_{\max}|, (1-\gamma)^{-1}, \varepsilon^{-1})$ | $\mathcal{O}\left(\varepsilon\right)$ | $\widetilde{\mathcal{O}}\left(\frac{|\mathcal{S}|^4|\mathcal{A}_{\max}|^5}{(1-\gamma)^{12}\varepsilon^8}\right)$ | None | ✓ |

pair $\mu, \nu$ as $V^{\mu,\nu} : \mathcal{S} \to \mathbb{R}$, we measure the gap of a strategy pair to an NE by the following metric

$$\text{Nash-Gap}(\mu, \nu) := V^{\mu^{\star},\nu}(s_0) - V^{\mu,\nu^{\star}}(s_0),$$

where $s_0$ is the starting state of the game[2] and $\nu^{\star}$ denotes one of the strategies from the set of best-response strategies to $\mu$. That is, $\mu^{\star} \in \mathrm{br}(\nu) := \mathrm{argmax}_{\mu} V^{\mu,\nu}(s_0)$ and $\nu^{\star} \in \mathrm{br}(\mu) := \mathrm{argmin}_{\nu} V^{\mu,\nu}(s_0)$. This objective has been widely adopted in Multi-Agent Reinforcement Learning (see, e.g., [Cui and Du, 2022a,b]) and it is easily motivated by the fact that any strategy profile output by an algorithm under study $(\mu_{\mathrm{out}}, \nu_{\mathrm{out}})$ such that $\text{Nash-Gap}(\mu_{\mathrm{out}}, \nu_{\mathrm{out}}) \leq \varepsilon$ is an $\varepsilon$-approximate Nash equilibrium, often shortened as $\varepsilon$-NE. However, it remained largely unexplored in the MAIL setting until the seminal work of Tang et al. [2024], who showed that minimizing the Nash Gap is fundamentally hard in MAIL since deviations in out-of-distribution states can incur linear regret.

A limitation of Tang et al. [2024] is that they provide an error propagation analysis only. While their analysis has the advantage of suggesting meaningful losses that can be minimized to ensure small Nash-Gap, it falls short in characterizing the amount of expert samples needed to learn a $\varepsilon$-NE from expert data. Moreover, their BLADES and MALICE algorithms have computational complexity that scales exponentially with the number of states in the game due to their for loops over the set of all possible deviations. This set has cardinality exponential in $|\mathcal{S}|$.

This work presents the first theoretical analysis of sample complexity in MAIL, and notably, it achieves this without exponential dependencies. Specifically, our contributions are as follows:

1. We provide a sample complexity analysis for BC, revealing the emergence of an *all deviation concentrability coefficient* (Theorem 3.1).
2. We formally separate MAIL from SAIL, proving in Theorem 3.2 that even with fully known transitions, for any non-interactive imitation learning algorithm (like BC) there exists a Markov Game with infinite single deviation concentrability coefficient where the Nash Gap remains constant even with infinite expert data.

---

[2]We will relax this to a stochastic starting state in the next sections.

3. On the positive side, we show that the dependence on the concentrability coefficient can be avoided if an interactive expert is available. In particular, assuming access to a Best Response Oracle, we propose an algorithm that achieves an $\varepsilon$-NE with $\mathcal{O}(\varepsilon^{-4})$ expert queries and oracle calls (Algorithm 2).

4. Additionally, we develop an algorithm that avoids the Best Response oracle and the concentrability coefficient simultaneously, achieving an $\varepsilon$-NE with $\mathcal{O}(\varepsilon^{-8})$ expert queries. Moreover, the algorithm is computationally efficient. Its design is based on the novel principle of maximum uncertainty response.

For clarity, we report a comparison of our results with existing MAIL algorithms in Table 1.

## 2 Preliminaries

We start by formalizing the concept of Two-Player Zero-Sum Markov Games. Then, we define the imitation learning settings considered in this work dubbed interactive and non-interactive respectively.

**Two-Player Zero-sum Markov Game.** An infinite-horizon two-player zero-sum Markov game is defined by the tuple $\mathcal{G} = (\mathcal{S}, \mathcal{A}, \mathcal{B}, P, r, \gamma, d_0)$, where $\mathcal{S}$ is the finite (joint-)state space, $\mathcal{A}$ is the finite action space of the first player, $\mathcal{B}$ is the finite action space of the second player, $P \in \mathbb{R}^{|\mathcal{S}||\mathcal{A}||\mathcal{B}| \times |\mathcal{A}|}$ is the (unknown) transition function, $r \in [-1, 1]^{|\mathcal{S}||\mathcal{A}||\mathcal{B}|}$ the reward vector, a discount factor $\gamma \in [0, 1)$ and $d_0$ a distribution over the state space from which the starting state is sampled. In a zero-sum Markov Game there is one player trying to maximize the rewards and one player aims to minimize the rewards. We assume that the first player is maximizing the reward and the second player aims to minimize it. It holds that $r^1(s, a, b) = -r^2(s, a, b)$ $\forall (s, a, b) \in \mathcal{S} \times \mathcal{A} \times \mathcal{B}$. Therefore, we can omit the superscript in the reward and simply refer to the reward as $r$. We define a policy of player 1 as $\mu : \mathcal{S} \to \Delta_{\mathcal{A}}$ and the policy of player 2 as $\nu : \mathcal{S} \to \Delta_{\mathcal{B}}$, where $\Delta$ is the probability simplex over the finite action spaces $\mathcal{A}$ and $\mathcal{B}$, respectively. Next, we define the value function for a given state $s \in \mathcal{S}$ and the state-action value function for a given state $s \in \mathcal{S}$ and joint actions $(a, b) \in \mathcal{A} \times \mathcal{B}$ for a given policy pair $(\mu, \nu)$. To this end, let us denote by $\{S_t, A_t, B_t\}_{t=0}^{\infty}$ the stochastic process generated by the interaction of the policy pair $(\mu, \nu)$ in the Markov Game, then we can define the value functions as follows $V^{\mu,\nu}(s) := \mathbb{E}_{\mu,\nu}\left[\sum_{t=0}^{\infty} r(S_t, A_t, B_t) \mid S_0 = s\right]$ and $Q^{\mu,\nu}(s, a, b) := \mathbb{E}_{\mu,\nu}\left[\sum_{t=0}^{\infty} r(S_t, A_t, B_t) \mid S_0 = s, A_0 = a, B_0 = b\right]$.

Additionally, we define the state visitation probability induced by a policy pair $(\mu, \nu)$ as $d^{\mu,\nu}(s') := (1 - \gamma)\mathbb{E}_{\mu,\nu}\left[\sum_{t=0}^{\infty} \gamma^t \mathbf{1}_{\{S_t = s'\}} \mid s_0 \sim d_0\right]$. If one player's policy is fixed, then the Markov Game induces a Markov decision process (MDP). Assuming that player 2 fixes their strategy, the induced transition function for a given state-action pair $(s, a) \in \mathcal{S} \times \mathcal{A}$ to new state $s' \in \mathcal{S}$ is given by $P_\nu(s' \mid s, a) := \sum_{b \in \mathcal{B}} \nu(b \mid s) P(s' \mid s, a, b)$. It is analogously defined if the policy of player 1 is fixed. Additionally, for a fixed strategy of the opponent player, we define the best response set as $\mathrm{br}(\nu) = \mathrm{argmax}_{\mu \in \Pi} \langle d_0, V^{\mu,\nu} \rangle$ and $\mathrm{br}(\mu) = \mathrm{argmin}_{\nu \in \Pi} \langle d_0, V^{\mu,\nu} \rangle$, respectively, where $\Pi$ denotes the set of all possible policies and $\mu^\star, \nu^\star$ as elements of these sets. It is important to note that the best response may not be unique, but the value is. A pair of policies is called a Nash equilibrium if both policies are best responses to each other. Last, we introduce the *Nash gap*, which measures how close a given policy pair $(\mu, \nu)$ is to a NE:

$$\mathrm{Nash-Gap}(\mu, \nu) := \left\langle d_0, V^{\mu^\star, \nu} - V^{\mu, \nu^\star} \right\rangle. \tag{1}$$

The Nash-Gap has the desirable property, that $\mathrm{Nash-Gap}(\mu, \nu) = 0$, if $(\mu, \nu)$ is a NE and $\mathrm{Nash-Gap}(\mu, \nu) > 0$, otherwise.

**Non-interactive Multi-Agent Imitation Learning.** In *non-interactive* MAIL, the learner observes a dataset $\mathcal{D} := \{\tau_k\}_{k=1}^N$ containing $N$ trajectories collected in the two-player zero-sum Markov Game, where the actions are sampled from the NE expert policy pair $(\mu^{\mathrm{E}}, \nu^{\mathrm{E}})$. For each trajectory $\tau_k$, a random length $H \sim \mathrm{Geo}(1 - \gamma)$ is sampled and then the sequence of states and (joint-)actions up to time $H$ are saved, i.e. $\tau_k := \{(s_t, a_t, b_t)\}_{t=1}^H$. After such dataset is collected, the learner can no longer collect new expert data. For this reason, we refer to this setting as non-interactive. Moreover, the learner might know the transition function of the Markov Game. The learner's goal is to adopt an algorithm $\mathrm{Alg}$ that takes as input $\mathcal{D}$, and outputs a pair of policies $(\widehat{\mu}, \widehat{\nu})$ such that $\mathbb{E}_{\mathrm{Alg}}[\mathrm{Nash-Gap}(\widehat{\mu}, \widehat{\nu})] < \varepsilon$.

**Interactive Multi-Agent Imitation learning.** In *interactive* MAIL, there is no initial dataset $\mathcal{D}$. The learner interacts with the environment for a certain number of rounds. At each round, the learner can collect a trajectory with a chosen policy pair and decide to query the expert at the visited states. The learner's goal is to adopt an algorithm Alg that after $\text{poly}(\varepsilon^{-1})$ main expert queries outputs a pair of policies $(\widehat{\mu}, \widehat{\nu})$ such that $\mathbb{E}_{\text{Alg}}[\text{Nash}-\text{Gap}(\widehat{\mu}, \widehat{\nu})] < \varepsilon$. Compared to the non-interactive setting, the expert can be queried during learning.

In the following, we present the theoretical results concerning the two above settings. In the next section, we study the non-interactive setting.

## 3  On the sample complexity of Multi-Agent Behavior Cloning

In this section, we give our first result, which concerns the sample complexity of Behavior Cloning.

Interestingly, our upper bound depends on a novel quantity $\mathcal{C}_{\max} \in \mathbb{R}$ dubbed *all policy deviation concentrability* coefficient, which is an infinite norm ratio between the occupancy distributions related to the notion of data coverage assumptions needed in Offline Zero-Sum Markov Games [Cui and Du, 2022a, Zhong et al., 2022] and concentrability coefficients in approximate dynamic programming [Scherrer et al., 2012, Geist et al., 2019, Vieillard et al., 2020]. Contrary to the analysis of Tang et al. [2024], we do not require that the occupancy measure of the equilibrium policy pair used to collect $\mathcal{D}$ is lower bounded by $\beta$. Therefore, our analysis also applies to the realistic setting where some states have zero probability to appear in $\mathcal{D}$.

We conclude this section with a lower bound inspired by the construction of Tang et al. [2024], which separates Multi-Agent Imitation Learning from Single-Agent Imitation Learning, showing the necessity of the concentrability coefficient in the multi-agent non-interactive setting even with a known transition model.

**Behavioral cloning in Markov Games.** In the context of Markov games, BC aims to recover a pair of policies $(\widehat{\mu}, \widehat{\nu})$ from expert demonstrations $\mathcal{D}$ based on maximum likelihood estimation. Formally, we have that $(\widehat{\mu}, \widehat{\nu}) = \text{argmax}_{(\mu,\nu)} \sum_{\tau \in \mathcal{D}} \log(\mathbb{P}(\tau; \mu, \nu))$, where $\mathbb{P}(\tau; \mu, \nu) = d_0(s_0) \prod_{h=1}^{H} \mu(a \mid s)\nu(b \mid s)P(s' \mid s, a, b)$ is the probability of generating trajectory $\tau$ under policies $(\mu, \nu)$, where $H \sim \text{Geo}(1 - \gamma)$. In the tabular set-up, we can obtain the closed-form solution of the above optimization problem $\widehat{\mu}(a \mid s) = \frac{N(s,a)}{N(s)}$, if $N(s) > 0$ and $\widehat{\mu}(a \mid s) = \frac{1}{|\mathcal{A}|}$ otherwise. Similarly, this holds for $\widehat{\nu}(b \mid s)$ by replacing $N(s, a)$ with $N(s, b)$. Here $N(s, a), N(s, b)$ and $N(s)$ denote the number of times that state-action pair $(s, a), (s, b)$ and state $s$ appear in $\mathcal{D}$.

Now, we can state our result for the upper bound of Behavior Cloning when minimizing the Nash Gap (1). We give a proof sketch below the theorem and the full proof can be found in Appendix C.

**Theorem 3.1.** *Let $(\mu^E, \nu^E)$ denote a Nash equilibrium policy pair in a two-player zero-sum Markov game, and let $\mathcal{D}$ contain trajectories from this expert policy pair. Let $(\widehat{\mu}, \widehat{\nu})$ be the policies obtained via Behavior Cloning from $\mathcal{D}$ of size $N$. Then, with probability at least $1 - \delta$, it holds:*

$$\text{Nash}-\text{Gap}(\widehat{\mu}, \widehat{\nu}) \leq \mathcal{C}_{\max} \frac{8}{(1 - \gamma)^2} \sqrt{\frac{|\mathcal{S}||\mathcal{A}_{\max}|\log^2(2|\mathcal{S}|/\delta)}{N}},$$

*where $C_{\max} = \max_{\mu,\nu} \max\left\{\max_{\nu^\star \in \text{br}(\mu)} \left\|\frac{d^{\mu^E, \nu^\star}}{\rho}\right\|_\infty, \max_{\mu^\star \in \text{br}(\nu)} \left\|\frac{d^{\mu^\star, \nu^E}}{\rho}\right\|_\infty\right\}$ and we set $\rho = d^{\mu^E, \nu^E}$.*

*Proof Sketch.* In the first step of the proof, we add and subtract the value function of the Nash equilibrium expert. Additionally, we use the definition of the Nash equilibrium, in particular that the policies are best responses to each other, to upper bound it by replacing it with the best responding policies to $\widehat{\mu}$ and $\widehat{\nu}$ respectively.

$$V^{\mu^\star, \widehat{\nu}}(s_0) - V^{\widehat{\mu}, \nu^\star}(s_0) \leq \underbrace{V^{\mu^\star, \widehat{\nu}}(s_0) - V^{\mu^\star, \nu^E}(s_0)}_{:=\text{Error}(\widehat{\nu})} + \underbrace{V^{\mu^E, \nu^\star}(s_0) - V^{\widehat{\mu}, \nu^\star}(s_0)}_{:=\text{Error}(\widehat{\mu})},$$

where $\mu^\star \in \text{br}(\widehat{\nu}), \nu^\star \in \text{br}(\widehat{\mu})$. Next, we can upper bound the two error terms separately. Note that the error terms each share one fixed policy, therefore, we can apply a version of the performance

difference lemma, the triangle inequality, and fix one best response for each player to obtain

$$\text{Error}(\widehat{\nu}) \leq \frac{2}{1-\gamma} \max_{\mu \in \text{br}(\widehat{\nu})} \mathbb{E}_{\mu,\nu^{\text{E}}} \left[ \sum_{t=0}^{\infty} \gamma^t \text{TV}\left(\nu^{\text{E}}(\cdot \mid s), \widehat{\nu}(\cdot \mid s)\right) \right]. \tag{2}$$

We note that $\max_{\mu \in \text{br}(\widehat{\nu})}$ contains the random variable $\widehat{\nu}$ but can be upper bounded with the best response to with respect to the general policy class $\max_{\nu \in \Pi} \max_{\mu \in \text{br}(\nu)}$. Last, we do a change of measure to get the expectation with respect to the expert policy pair. Then, we bound the ratio of the state visitation distribution and bound the expectation with concentration inequalities to obtain with probability of at least $1 - \delta$

$$V^{\mu^\star,\widehat{\nu}}(s_0) - V^{\mu^\star,\nu^{\text{E}}}(s_0) \leq \frac{8}{(1-\gamma)^2} \max_{\nu \in \Pi} \max_{\mu \in \text{br}(\nu)} \left\| \frac{d^{\mu,\nu^{\text{E}}}}{d^{\mu^{\text{E}},\nu^{\text{E}}}} \right\|_\infty \sqrt{\frac{|\mathcal{S}| |\mathcal{B}| \log^2(2 |\mathcal{S}| / \delta)}{N}}.$$

Analogous calculations for the second player complete the proof. The full proof is given in Appendix C. $\qquad \square$

We now discuss several important implications of the derived theorem, particularly focusing on the quantity $\mathcal{C}_{\max}$, referred to as the *all policy deviation concentrability* coefficient (see, e.g., [Cui and Du, 2022a, Zhong et al., 2022]). Intuitively, the theorem indicates that if a best response against a possible recovered policy shifts the support of the state visitation distribution away from the one induced by the observed Nash equilibrium, the corresponding objective becomes unbounded.

**Remark 3.1.** *While restricting, this requirement is weaker than a uniform lower bound on the equilibrium state occupancy measure assumed by Tang et al. [2024], that is $d^{\mu_E,\nu_E} \geq \beta$. In particular, it always holds that $\mathcal{C}_{\max} \leq \beta^{-1}$.*

Next, we introduce a lower bound on $\mathcal{C}_{\max}$, that intuitively describes how different the coverage of different Nash equilibrium policies can be. Formally, we introduce $\mathcal{C}(\mu^{\text{E}}, \nu^{\text{E}}) :=$ $\max \left\{ \max_{\mu \in \text{br}(\nu^{\text{E}})} \left\| \frac{d^{\mu,\nu^{\text{E}}}}{d^{\mu^{\text{E}},\nu^{\text{E}}}} \right\|_\infty, \max_{\nu \in \text{br}(\mu^{\text{E}})} \left\| \frac{d^{\mu^{\text{E}},\nu}}{d^{\mu^{\text{E}},\nu^{\text{E}}}} \right\|_\infty \right\}$, which we refer to as the *expert policy deviation concentrability* coefficient. Clearly, we have that $\mathcal{C}_{\max} \geq \mathcal{C}(\mu^{\text{E}}, \nu^{\text{E}})$ as the latter only consider the best responses to the Nash equilibrium policies.

Unfortunately, we show that even for this smaller quantity there exists a zero-sum Markov game in which $\mathcal{C}(\mu^{\text{E}}, \nu^{\text{E}})$ is unbounded, and we show that in such a game, no non-interactive algorithm can recover a Nash profile even with an infinite amount of data. We present this result in the next section. Note, that it remains an open question if it is possible to have a bounded $\mathcal{C}(\mu^{\text{E}}, \nu^{\text{E}})$ but an unbounded $\mathcal{C}_{\max}$. We discuss this further in Section 6.

The observations are similar in spirit to those obtained in the offline setting [Cui and Du, 2022a, Zhong et al., 2022]. In these works, the authors derive a lower bound that shows the necessity of a *unilateral concentration* assumption to minimize the Nash gap. However, their construction does not apply to the Imitation Learning setting.

### 3.1 Necessity of $\mathcal{C}(\mu_E, \nu_E)$ in non-interactive MAIL

In this section, we provide the negative result, that a Markov Game exists, such that the expert deviation concentrability coefficient of Theorem 3.1 is unbounded.

The first hardness results to minimize the Nash Gap in Multi-Agent Imitation Learning were derived by Tang et al. [2024, Thm. 4.3]. Next, we will give a stronger result, showing that even in the case of full knowledge of the transition model and perfect recovery of the state visitation distribution of the expert, the Nash gap is of the order $(1-\gamma)^{-1}$. A detailed discussion on the difference between the following result and the one obtained by Tang et al. [2024, Thm. 4.3] can be found in Appendix I. An illustration of the Zero-Sum Markov game can be found in Fig. 1 and the full proof in Appendix D.

**Theorem 3.2** (Construction of MG). *For any learning algorithm* Alg *in the non-interactive imitation learning setting, there exists a zero-sum Markov game with $\mathcal{C}(\mu^{\text{E}}, \nu^{\text{E}}) = \infty$ such that the output policies $\hat{\mu}, \hat{\nu}$ satisfy $\mathbb{E}_{\text{Alg}} \left[ \langle d_0, V^{\mu^\star,\widehat{\nu}} - V^{\widehat{\mu},\nu^\star} \rangle \right] \geq (1-\gamma)^{-1}$. The result continues to hold even if* Alg *is aware of the transition dynamics of the game.*

This theorem illustrates a fundamental limitation of BC in zero-sum Markov games. Specifically, it reveals that even perfect recovery of the Nash expert's state visitation distribution, along with complete knowledge of the transition model, is insufficient for minimizing the Nash gap. The key insight is that a Nash equilibrium only guarantees robustness against *unilateral* deviations. As a result, regions of the Markov game that require *joint* deviations to be visited may remain underexplored by the expert, leaving the learner vulnerable in those regions. This can be seen in Fig. 1, if the learner has a (jointly) inaccurate policy in state $s_1$, the best response of the agents can change the expert path to exploit the opponent in the red path of the Markov Game and the green one respectively. Notably, this phenomenon persists even when the transition model is known. This can be seen as $S_{\text{xplt1}}, S_{\text{xplt2}}$ and $S_{\text{copy}}$ are sets of states, and each action combination leads to a different unique state, i.e. $|S_{\text{xplt1}} \cup S_{\text{xplt2}} \cup S_{\text{copy}}| = |\mathcal{A}| |\mathcal{B}|$. This highlights the necessity of *interactive* Imitation Learning to explore strategically important but unobserved regions of the state space.

This issue marks a critical distinction between multi-agent and single-agent imitation learning. In single-agent settings, BC suffices to achieve a good performance [Rajaraman et al., 2020, 2021a, Foster et al., 2024].

Moreover, it is important to notice that in the construction used by Tang et al. [2024, Thm. 4.3], knowledge of the transition model enables learners to steer toward expert-visited trajectories. In contrast, our result establishes a hardness construction in the zero-sum setting showing that the guidance provided by transition knowledge is insufficient.

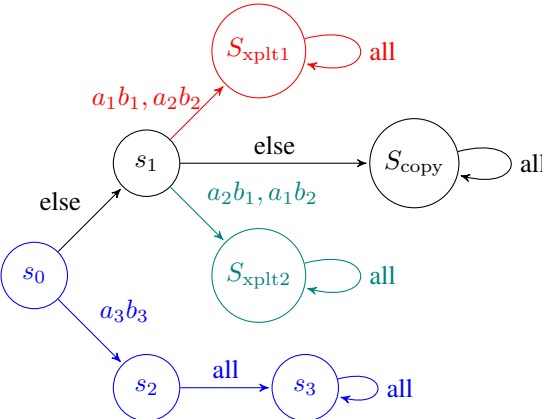

Figure 1: 2 Player Zero-Sum Game with Linear Regret in case of full knowledge of transition.

**Possible ways to learn under unbounded** $\mathcal{C}(\mu^{\text{E}}, \nu^{\text{E}})$. The above theorem makes clear that no algorithm can learn in the non-interactive setting if $\mathcal{C}(\mu^{\text{E}}, \nu^{\text{E}}) = \infty$. We can think of several remedies to this fact. First, we could require to observe data from the possible strategies in the set of Nash equilibria. In this case, we would encounter a smaller concentrability coefficient which features the average of the equilibria occupancy measures in the denominator. A second remedy is to move to the interactive MAIL setting which allows the learner to collect reward free trajectories in the Markov Game and query the expert policy pair along the visited states.

Since the former assumption is rarely realistic, we later propose an interactive algorithm (Algorithm 2) that actively queries expert demonstrations to reduce the Nash gap of the resulting policy.

## 4   Avoiding the single deviation concentrability in interactive MAIL

In this section, we introduce two algorithms (MAIL-BRO and MURMAIL) designed to avoid dependence on $\mathcal{C}(\mu^{\text{E}}, \nu^{\text{E}})$ at the cost of moving to the interactive imitation learning setting.

Both of our algorithms address key limitations of the approaches proposed in BLADES and MALICE by Tang et al. [2024], as their methods require exponential compute and focus solely on error propagation analysis without providing convergence guarantees for the resulting policies. In contrast, our algorithms are accompanied by both convergence guarantees and polynomial computational cost. To motivate these algorithms, let us briefly revisit the structure of the original proof. In offline BC,

we lack data corresponding to the best responses against the estimated expert policies. As a result, it is not feasible to directly estimate the expectation in Eq. (2). To circumvent this, we apply a change of measure at the cost of introducing the all deviation concentrability term.

Our first approach to overcome this limitation is to introduce a *Best Response oracle*, which enables sampling from the distributions $(\mu, \nu^{\mathrm{E}})$ and $(\mu^{\mathrm{E}}, \nu)$, where $\mu \in \mathrm{br}(\widehat{\nu})$ and $\nu \in \mathrm{br}(\widehat{\mu})$, thereby allowing us to estimate the relevant expectations without incurring the concentrability coefficient. Formally, we have the following definition, also used in previous works (see e.g. Hellerstein et al. [2019]).

**Definition 4.1** (Best Response Oracle). *Let $(\widehat{\mu}, \widehat{\nu})$ be a pair of policies for a Markov Game $\mathcal{G}$. Then, a* Best Response Oracle *generates policies $\mu \in \mathrm{br}(\widehat{\nu})$ and $\nu \in \mathrm{br}(\widehat{\mu})$.*

However, it is not straightforward to use the policies given by the Best Response Oracle. Starting from (2), we derive the following optimization problem

$$\min_{\widehat{\mu} \in \Pi} \max_{\nu \in \mathrm{br}(\widehat{\mu})} \mathbb{E}_{\mu^{\mathrm{E}}, \nu} \left[ \sum_{t=0}^{\infty} \gamma^t \mathrm{TV} \left( \mu^{\mathrm{E}}(\cdot \mid s), \widehat{\mu}(\cdot \mid s) \right) \right]. \tag{3}$$

Even under the assumption of being able to generate samples from $\mu^{\mathrm{E}}, \nu$, where $\nu \in \mathrm{br}(\widehat{\mu})$, two problems remain. First of all, the optimization problem Eq. (3) is non-convex in $\widehat{\mu}$. Secondly, in order to estimate the minimizer $\widehat{\mu}$, we need to collect data from the occupancy measure of the policy pair $\mu^{\mathrm{E}}, \nu$ for $\nu \in \mathrm{br}(\widehat{\mu})$, which depends on the minimizer itself.

To overcome this issue, we make use of the following bound, here only obtained for fixing $\mu_k$:

$$\frac{1}{K} \sum_{k=1}^{K} \left\langle d_0, V^{\mu_E, \nu_k^{\star}} - V^{\mu_k, \nu_k^{\star}} \right\rangle \leq \sqrt{\frac{|\mathcal{A}_{max}|}{K(1-\gamma)^2} \sum_{k=1}^{K} \mathbb{E}_{s \sim d^{\mu_k, \nu_k^{\star}}} \left[ \| \mu_E(\cdot \mid s) - \mu_k(\cdot \mid s) \|^2 \right]}, \tag{4}$$

where $\nu_k^{\star} \in \mathrm{br}(\mu_k)$. This expression can be derived via the performance difference Lemma, Cauchy-Schwarz, and eventually Jensen's inequality, and it analogously holds for $\nu_k$ fixed. The above inequality is crucial for the design of our algorithms in the interactive setting, as shown next.

### 4.1 Efficient algorithm with a best response oracle

In this section, we present our statistically and computationally efficient algorithm with a best response oracle defined as follows.

---

**Algorithm 1:** Multi-Agent Imitation Learning with Best Response Oracle (MAIL-BRO)

---

**Input:** number of iterations $K$, learning rates $\eta$, BR oracle, initial policies $(\mu_1, \nu_1)$
**Output:** $\varepsilon$-Nash equilibrium $(\hat{\mu}, \hat{\nu})$
**for** $k = 1$ *to* $K$ **do**
    | **Update policies:**;
    | Query BR oracle to obtain $\mu_k^{\star} \in \mathrm{br}(\widehat{\nu}_k), \nu_k^{\star} \in \mathrm{br}(\widehat{\mu}_k)$ ;
    | Sample $S_k^{\mu} \sim d^{\mu_k, \nu_k^{\star}}, A_k^{\mu} \sim \mu_E(\cdot \mid S_k^{\mu}), S_k^{\nu} \sim d^{\mu_k^{\star}, \nu_k}, A_k^{\nu} \sim \nu_E(\cdot \mid S_k^{\nu})$ ;
    | $g_k^{\mu}(s, a) = \mu_k(a \mid S_k^{\mu}) \mathbb{1}_{S_k^{\mu}=s} - \mathbb{1}_{A_k^{\mu}=a}$ ;
    | $g_k^{\nu}(s, a) = \nu_k(a \mid S_k^{\nu}) \mathbb{1}_{S_k^{\nu}=s} - \mathbb{1}_{A_k^{\nu}=a}$ ;
    | $\mu_{k+1}(a \mid s) \propto \mu_k(a \mid s) \exp \left( -\eta g_k^{\mu}(s, a) \right)$ ;
    | $\nu_{k+1}(b \mid s) \propto \nu_k(b \mid s) \exp \left( -\eta g_k^{\nu}(s, a) \right)$
**end**
**return** $\mu_{\widehat{k}}$, $\nu_{\widehat{k}}$ *for* $\widehat{k} \sim \mathrm{Unif}([K])$

---

With the best response oracle and the bound in Eq. (4) in place, we can aim at applying a no-regret algorithm to the loss sequence $\left\{ \mathbb{E}_{s \sim d^{\mu_k, \nu_k^{\star}}} \left[ \| \mu_E(\cdot \mid s) - \mu_k(\cdot \mid s) \|^2 \right] \right\}_{k=1}^{K}$. Since these losses are not directly observable, MAIL-BRO (see Algorithm 1) at each iteration performs a step of exponential weights updates with a stochastic unbiased gradient denoted by $g_k^{\mu}$ and $g_k^{\nu}$ for the two players respectively. These gradient estimates can be shown to have almost surely bounded noise too.

Exploiting these facts in the analysis of MAIL-BRO, we can attain the following formal result.

**Theorem 4.1.** *Let us run Algorithm 1 for* $K = \mathcal{O}\left(\frac{|\mathcal{S}||\mathcal{A}_{\max}|^2 \log|\mathcal{A}_{\max}| \log(1/\delta)}{(1-\gamma)^4 \varepsilon^4}\right)$ *iterations with learning rate* $\eta = \frac{2|\mathcal{S}| \log|\mathcal{A}_{\max}|}{K}$. *Then, the sequence of policies* $\{\mu_k, \nu_k\}_{k=1}^K$ *satisfies with probability at least* $1 - 5\delta$ *that* $\frac{1}{K}\sum_{k=1}^K \max_{\mu \in \Pi} \langle d_0, V^{\mu,\nu_k}\rangle - \min_{\nu \in \Pi} \langle d_0, V^{\mu_k,\nu}\rangle \leq \mathcal{O}(\varepsilon)$. *Therefore, setting* $\delta = \mathcal{O}(\varepsilon)$ *ensures that for a certain* $\hat{k} \sim \mathrm{Unif}([K])$ *it holds that* $\mathbb{E}\left[\mathrm{Nash-Gap}(\mu_{\hat{k}}, \nu_{\hat{k}})\right] \leq \varepsilon$. *That is,* $\mu_{\hat{k}}, \nu_{\hat{k}}$ *is an* $\varepsilon$-*Nash equilibrium in expectation.*

The proof can be found in Appendix E. We observe that, compared to standard BC, the sample complexity now is of the order $\mathcal{O}(\varepsilon^{-4})$, which is worse by a factor of $\varepsilon^{-2}$. However, this trade-off allows us to completely avoid dependence on the all policy deviation concentrability coefficient in the MAIL-BRO upper bound. That is, MAIL-BRO is able to effectively recover an approximate equilibrium from expert data in a larger class of games.

Unfortunately, assuming a best response oracle might be limiting in some cases. For those cases, we can replace the call to the oracle with the maximum uncertainty responding policy as we explain in the next section.

## 4.2 Avoiding the best response oracle thanks to the maximum uncertainty response

Here, we introduce our algorithm MURMAIL (Algorithm 2) which can be applied in the most general setting where $\mathcal{C}(\mu^{\mathrm{E}}, \nu^{\mathrm{E}}) = \infty$ and the best response oracle is not available. The idea is again to start from (4), but instead of querying the Best Response Oracle, the objective is upper bounded by the maximum uncertainty policy. It is important to note that the exploration follows in a decentralized way, avoiding the *curse of multi-agents* by exploring induced MDPs instead of the original Markov Game.

---

**Algorithm 2:** Maximum Uncertainty Response Multi-Agent Imitation Learning (MURMAIL)

**Input:** number of iterations $K$, learning rates $\eta$, inner iteration budget $T$, initial $(\mu_1, \nu_1)$
**Output:** $\varepsilon$-Nash equilibrium $(\hat{\mu}, \hat{\nu})$
**for** $k = 1$ **to** $K$ **do**

    **Inner Single-Agent RL Updates:**
    % Maximum uncertainty response to $\mu$-player update
    Define single agent transition $P_{\mu_k}(s' \mid s, b) = \sum_{a \in \mathcal{A}} \mu_k(a \mid s) P(s' \mid s, a, b)$;
    Define single agent stochastic reward $R_{\mu_k}(s) \to \mathbb{1}_{\{A_E = A'_E\}} - 2\mu_k(A_E \mid s) + \|\mu_k(\cdot|s)\|^2$
      where $A_E, A'_E \sim \mu_E(\cdot \mid s)$;
    $y_k = \mathrm{UCBVI}(T, P_{\mu_k}, R_{\mu_k})$;
    % Maximum uncertainty response to $\nu$-player update
    $P_{\nu_k}(s'|s,a) = \sum_{b \in \mathcal{B}} \nu_k(b|s) P(s' \mid s, a, b)$;
    $R_{\nu_k}(s) \to \mathbb{1}_{\{A_E = A'_E\}} - 2\nu_k(A_E \mid s) + \|\nu_k(\cdot \mid s)\|^2$ where $A_E, A'_E \sim \nu_E(\cdot \mid s)$;
    $z_k = \mathrm{UCBVI}(T, P_{\nu_k}, R_{\nu_k})$
    **Update policies:**
    Sample $S_k^\mu \sim d^{\mu_k, y_k}, A_k^\mu \sim \mu_E(\cdot \mid S_k^\mu), S_k^\nu \sim d^{z_k, \nu_k}, A_k^\nu \sim \nu_E(\cdot \mid S_k^\nu)$.
    $g_k^\mu(s, a) = \mu_k(a \mid S_k^\mu)\mathbb{1}_{S_k^\mu = s} - \mathbb{1}_{A_k^\mu = a}$
    $g_k^\nu(s, a) = \nu_k(a \mid S_k^\nu)\mathbb{1}_{S_k^\nu = s} - \mathbb{1}_{A_k^\nu = a}$
    $\mu_{k+1}(a \mid s) \propto \mu_k(a \mid s) \exp\left(-\eta g_k^\mu(s, a)\right)$;
    $\nu_{k+1}(b \mid s) \propto \nu_k(b \mid s) \exp\left(-\eta g_k^\nu(s, a)\right)$

**end**
**return** $\mu_{\hat{k}}, \nu_{\hat{k}}$ for $\hat{k} \sim \mathrm{Unif}([K])$

---

If the best response $\nu_k^\star \in \mathrm{br}(\mu_k)$ cannot be computed, we can majorize the above quantity by the policy $y_k$ such that $y_k \in \mathrm{argmax}_{\nu \in \Pi} \frac{|\mathcal{A}_{\max}|}{K(1-\gamma)^2}\mathbb{E}_{s \sim d^{\mu_k, \nu}}\left[\|\mu_E(\cdot \mid s) - \mu_k(\cdot \mid s)\|^2\right]$. In words, $y_k$ is the policy that solves a single-agent MDP with reward $\|\mu_E(\cdot \mid s) - \mu_k(\cdot \mid s)\|^2$ where the opponent keeps the strategy $\mu_k$ fixed and the player with strategy $\nu$ seeks to maximize the probability of visiting *uncertain* states where the uncertainty is captured by $\|\mu_E(\cdot \mid s) - \mu_k(\cdot \mid s)\|^2$. This intuition motivates the name *maximum uncertainty response*.

At this point, since both the policies $\mu_k$ and $y_k$ are known, it is possible to roll out such policy pair in the environment and collect data to control $\|\mu_E(\cdot \mid s) - \mu_k(\cdot \mid s)\|^2$ for states $s$ in the support of $d^{\mu_k, y_k}$. Of course, exact computation of $y_k$ is not possible because we know neither the transition dynamics nor $\mu_E$ (which enters the reward function) exactly. However, an approximate solution can be computed, for example, via UCBVI[3] adapted to handle the stochastic nature of the reward and the discounted setting considered in this work.

The following result states the theoretical guarantees for Algorithm 2. A proof can be found in Appendix E.

**Theorem 4.2.** *Let us run Algorithm 2 for* $K = \mathcal{O}\left(\frac{|\mathcal{S}||\mathcal{A}_{\max}|^2 \log|\mathcal{A}_{\max}| \log(1/\varepsilon)}{(1-\gamma)^4 \varepsilon^4}\right)$ *outer iterations and* $T = \mathcal{O}\left(\frac{|\mathcal{S}|^3 |\mathcal{A}_{\max}|^3 \log(1/\varepsilon)}{(1-\gamma)^8 \varepsilon^4}\right)$ *inner iterations with learning rate* $\eta = \frac{2|\mathcal{S}| \log|\mathcal{A}_{\max}|}{K}$. *Then, for a certain* $\hat{k} \sim \mathrm{Unif}([K])$ *it holds that* $\mathbb{E}\left[\mathrm{Nash-Gap}(\mu_{\hat{k}}, \nu_{\hat{k}})\right] \leq \varepsilon$.

It is easy to see that since the total number of expert queries is of order $\mathcal{O}(K \cdot T)$, the total number of expert queries to achieve an $\varepsilon$-approximate Nash equilibrium in expectation is $\widetilde{\mathcal{O}}\left(\frac{|\mathcal{S}|^4 |\mathcal{A}_{\max}|^5}{(1-\gamma)^{12} \varepsilon^8}\right)$.

Again, notice that there is no concentrability requirement in the upper bound and that the result is achieved without the need to call a best response oracle. This comes at the cost of a worse sample complexity bound but is applicable to a larger class of games, even in those where a best response oracle is not available.

**Remark 4.1.** *Note that our algorithms scale with* $\mathrm{poly}(|\mathcal{A}_{\max}|)$. *While this may appear suboptimal in the two-player zero-sum setting, it is important to emphasize that the underlying algorithms support decentralized execution. In particular, in Algorithm 1, the dependence on* $\mathcal{A}_{\max}^2$ *does not stem from the two-player structure, but rather from the reformulation of the objective necessary to obtain an unbiased estimator for the gradient update. Similarly, the* $|\mathcal{A}_{\max}|^5$ *dependence in Algorithm 2 arises from the squared objective and the RL inner loop. Crucially, in this inner loop, each agent solves a single-agent MDP, ensuring that the algorithm remains fully decentralized. Altogether, these observations indicate that our algorithms scale linearly with* $|\mathcal{A}_{\max}|$ *and* **do not suffer from the curse of multi-agents** *in the* $n$*-player setting. A sketched version for* $n$*-player general-sum games can be found in Appendix H.*

## 5 Numerical Validation

In this section, we provide a numerical evaluation of our proposed algorithms in the Markov Game considered in the lower bound construction (Fig. 1) as this environment allows us to control $\mathcal{C}(\mu^{\mathrm{E}}, \nu^{\mathrm{E}})$ by considering different convex combinations of the two pure Nash equilibria profiles (i.e., the black and the blue path in Figure 1). This environment serves as a proof of concept to demonstrate the practical feasibility of our methods. In particular, we aim to highlight that the performance of BC depends on the concentrability coefficient $\mathcal{C}(\mu^{\mathrm{E}}, \nu^{\mathrm{E}})$ even when it is bounded, and completely fails when $\mathcal{C}(\mu^{\mathrm{E}}, \nu^{\mathrm{E}}) = \infty$. Note that in all considered cases, we have that $\beta = 0$ and therefore the BC bound proven in Tang et al. [2024] would always be vacuous, while Theorem 3.1 remains valid. The code used for the experiments is available at https://github.com/tfreihaut/Murmail.

We evaluate Multi-Agent BC and MURMAIL (Algorithm 2) in the considered environment and measure the exploitability of the resulting policies with respect to the number of expert queries (for MURMAIL) and dataset size (for BC). The results are presented in Fig. 2.

As predicted by our theoretical analysis, Multi-Agent BC fails in settings with $\mathcal{C}(\mu^{\mathrm{E}}, \nu^{\mathrm{E}}) = \infty$, whereas MURMAIL still succeeds in minimizing the Nash gap. However, in environments where $\mathcal{C}(\mu^{\mathrm{E}}, \nu^{\mathrm{E}}) < \infty$, BC can outperform MURMAIL in terms of efficiency $\varepsilon^{-2}$ compared to $\varepsilon^{-8}$. Nevertheless, one should also consider that the performance of MURMAIL is independent of $\mathcal{C}(\mu^{\mathrm{E}}, \nu^{\mathrm{E}})$ and therefore MURMAIL can outperform BC in cases where $\mathcal{C}(\mu^{\mathrm{E}}, \nu^{\mathrm{E}})$ is bounded but large. This highlights the importance of algorithm selection based on the underlying environment. Additional details, experiments in another environment, and practical insights for improving MURMAIL's performance are discussed in Appendix J.

---

[3]or any other algorithm for solving a single agent discounted tabular Markov decision process.

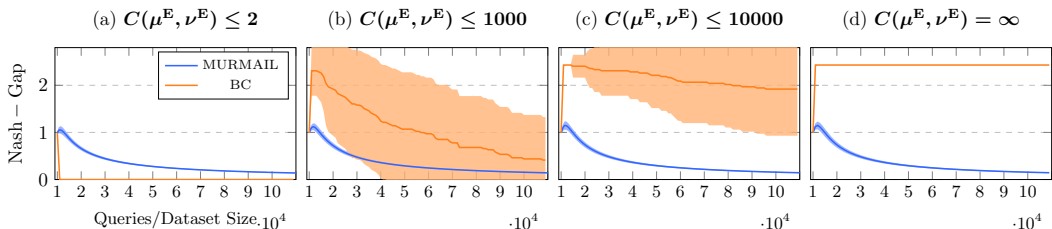

Figure 2: Empirical evaluation for environments with different $\mathcal{C}(\mu^{\mathrm{E}}, \nu^{\mathrm{E}})$.

## 6  Conclusion and Future Directions

This paper provides the first sample complexity analysis of behavioral cloning in the multi-agent setting. The provided upper bound depends on the *all policy deviation concentrability* coefficient. Additionally, which is shown to be unavoidable in general. Unfortunately, it is quite easy to come up with MGs where the concentrability coefficient is unbounded. In particular, we introduce the smaller quantity $\mathcal{C}(\mu^{\mathrm{E}}, \nu^{\mathrm{E}})$ is unbounded and where no non-interactive MAIL algorithm can learn an equilibrium from data. In this situation, we resort to expert queries and we introduce novel algorithms dubbed MAIL-BRO and MURMAIL, which achieve an $\varepsilon$-approximate Nash equilibrium with a polynomial number of expert queries and computational cost polynomial in all problem parameters.

We outline a few interesting future directions and research questions left open by our work.

**Lower Bound.** This work has provided a hardness result, showing that if $\mathcal{C}(\mu^{\mathrm{E}}, \nu^{\mathrm{E}})$ is unbounded, there exists no non-interactive MAIL algorithm that can learn. However, it holds true that $\mathcal{C}_{\max} \geq \mathcal{C}(\mu^{\mathrm{E}}, \nu^{\mathrm{E}})$ and it remains an open question if there exists a Markov Game, where $\mathcal{C}(\mu^{\mathrm{E}}, \nu^{\mathrm{E}})$ is bounded but $\mathcal{C}_{\max} = \infty$ and no non-interactive MAIL algorithm can recover an equilibrium. Answering this would close the gap between the provided upper bound of BC and the lower bound. In particular, this would answer what the fundamental quantity of non-interactive MAIL is. Additionally, it remains an open question if one can prove a statistical lower bound featuring $\mathcal{C}(\mu^{\mathrm{E}}, \nu^{\mathrm{E}})$ or $\mathcal{C}_{\max}$ and whether BC is rate optimal in $\varepsilon$.

**Improving the theoretical bounds in $\varepsilon$ and problem dependent parameters.** The focus of this work was to show the first sample complexity bound for a computationally efficient algorithm in the queriable expert setting. For the sake of simplicity, we did not try to optimize the dependence of the upper bound in the accuracy parameters $\varepsilon$, effective horizon $(1 - \gamma)^{-1}$, states and actions cardinality $|\mathcal{S}|$ and $|\mathcal{A}|$. A possible direction of improvement is to derive a tighter analysis of the outer loop using faster rates for the regret of the squared loss (see for example [Cesa-Bianchi and Lugosi, 2006, Chapter 3]).

Moreover, the upper bound could be improved by removing the need of the RL inner loop in MURMAIL. In general, replacing it with a no regret learner that minimizes the regret in an MDP with changing reward function and transitions is not possible because of the negative result by Abbasi-Yadkori et al. [2013]. On the positive side, it is known from the game theoretic literature that no regret learning would be possible under these conditions if the state space is tree structured as in an extensive form game [Osborne and Rubinstein, 1994]. We leave the study of this improvement, which can be relevant for several games such as Poker, for future works.

**Extension to deep imitation learning.** The current analysis is limited to tabular Markov Games. However, the main conceptual ideas easily carry on to deep imitation learning experiments. The largest theory-practice gap would be in the inner loop where `UCBVI` would need to be replaced by a Deep RL algorithm such as `DQN` Mnih et al. [2015] or Soft Actor Critic Haarnoja et al. [2018], just to name a few.

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

## Contents of Appendix

This appendix provides supplementary material to support the main findings of the paper. It begins with an overview of all the relevant notations used throughout the work, followed by a review of related work. We then present the complete proofs for key results (Appendix C to Appendix G), which were omitted from the main text. Appendix H outlines how our framework can be extended to n-player general-sum games. Further details on our experimental setup, results, and practical application considerations are provided in Appendix J; this section also features a comparison with the lower bound from Tang et al. [2024]. Finally, the appendix compiles a list of useful results, along with their proofs, that are referenced throughout this work. For a better overview we provide a table of contents.

# A   Notation

| Notation | Description |
|---|---|
| $\mathcal{G}$ | Two-Player Zero-Sum Markov Game |
| $\mathcal{S}$ | Finite (joint-)state space |
| $\mathcal{A}$ | Player 1's finite action space |
| $\mathcal{B}$ | Player 2's finite action space |
| $\mathcal{A}_{max}$ | Max action space size, $max(|\mathcal{A}|, |\mathcal{B}|)$ or $max_i(|\mathcal{A}_i|)$ |
| $P$ | Transition function |
| $r$ | Reward vector |
| $\gamma$ | Discount factor |
| $d_0$ | Initial state distribution |
| $\Delta_{\mathcal{A}}, \Delta_{\mathcal{B}}$ | Probability simplex over action spaces $\mathcal{A}, \mathcal{B}$ |
| $\mu$ | $\mathcal{S} \to \Delta_{\mathcal{A}}$, Policy of player 1 |
| $\nu$ | $\mathcal{S} \to \Delta_{\mathcal{B}}$, Policy of player 2 |
| $V^{\mu,\nu}(s)$ | State value function for policy pair $(\mu, \nu)$ at state $s$ |
| $Q^{\mu,\nu}(s, a, b)$ | State-action value for $(\mu, \nu)$ at $(s, a, b)$ |
| $d^{\mu,\nu}(s')$ | State visitation probability for policy pair $(\mu, \nu)$ |
| $\mathrm{Nash} - \mathrm{Gap}(\mu, \nu)$ | Gap to Nash Equilibrium (NE) for strategy pair $(\mu, \nu)$ |
| $br(\cdot)$ | Best response set |
| $\mu^* \in \mathrm{br}(\nu)$ | Best response strategy for player 1 to $\nu$ |
| $\nu^* \in \mathrm{br}(\mu)$ | Best response strategy for player 2 to $\mu$ |
| $\varepsilon$-NE | $\varepsilon$-approximate Nash Equilibrium |
| $P_\nu(s'|s, a)$ | Induced transition to $s'$ from $(s, a)$ with fixed $\nu$ |
| $\Pi$ | Set of all possible policies |
| $\mathcal{D}$ | Dataset of N trajectories |
| $N$ | Number of trajectories in dataset |
| $\tau_k$ | k-th trajectory in dataset |
| $H$ | Trajectory length, $H \sim Geo(1 - \gamma)$ |
| Alg | Algorithm outputting a policy pair |
| $(\hat{\mu}, \hat{\nu})$ | Output/Behavior Cloning policy pair |
| $\mathcal{C}(\mu, \nu)$ | Single policy deviation concentrability of $(\mu, \nu)$ |
| $\mathbb{P}(\tau; \mu, \nu)$ | Probability of trajectory $\tau$ given policies $(\mu, \nu)$ |
| $N(s, a), N(s, b), N(s)$ | Counts of $(s, a), (s, b), s$ in $\mathcal{D}$ |
| $\delta$ | Probability threshold (confidence bounds) |
| $Error(\hat{\nu})$ | Error term for player 2's estimated policy |
| $Error(\hat{\mu})$ | Error term for player 1's estimated policy |
| $S_{xplt1}, S_{xplt2}, S_{copy}$ | State sets in constructed Markov Game |
| $g_k^{\mu}, g_k^{\nu}$ | Stochastic unbiased gradient estimates |
| $\eta$ | Learning rate |
| $y_k, z_k$ | Policies from UCBVI in MURMAIL |
| $R_{\mu_k}(s), R_{\nu_k}(s)$ | Single agent stochastic reward in MURMAIL |
| $\varepsilon_{opt}$ | Optimality gap for RL inner loop |
| $\pi_i, \pi_{-i}$ | Policy of player $i$ and others in n-player games |
| $TV(\cdot, \cdot)$ | Total Variation distance |

# B    Related Work

Here, we present the most related work to our results.

**Single-Agent Imitation Learning.**    There has been significant progress in the theoretical analysis of single-agent imitation learning. In the fully offline setting, Behavior Cloning (BC) [Pomerleau, 1991] has recently been revisited by Foster et al. [2024] using the log loss as a supervised learning notion to be minimized between the imitator and the expert policy. Foster et al. [2024] shows an expert sample complexity bound independent of the horizon parameter under deterministic stationary policies and sparse reward function. Therefore, a dependence on the horizon appears only if the reward function is dense or if the class containing the expert policy is non stationary. Moreover, they prove that, without further assumptions, interactive imitation learning cannot outperform BC in a worst case sense.

This last finding is surprising given the seminal results showing that interactive imitation learning algorithms such as Dagger [Ross et al., 2011], Logger [Li and Zhang, 2022] and On-Q or reward moments matching [Swamy et al., 2021] outperform BC with the 0/1 or total variation loss in terms of error propagation analysis. Alternatively, better error propagation analysis properties can be derived if resetting to states sampled from the expert state occupancy measure is allowed [Swamy et al., 2023].

Some benefits over BC in the single-agent setting can be instead obtained with known transitions and initial distributions. Along this line Mimic-MD [Rajaraman et al., 2020] shows that the expert sample complexity can be improved by a factor $\sqrt{H}$ where $H$ is the finite horizon of the problem. Moreover, this is the best possible improvement without further assumptions given the lower bound of Rajaraman et al. [2021b] for $N \geq 6H$. Swamy et al. [2022] improve further the upper bound in the small data regime $N \leq H$. Later, MB-TAIL [Xu et al., 2023] achieves the optimal sample complexity for the large sample regime just under trajectory access to the environment (without requiring perfect knowledge of dynamics and initial state distribution).

Moreover, given trajectory access to the MDP, imitation learning in the single-agent setting is possible without observing the expert actions. For example, it is possible to imitate observing only the states visited by the expert [Sun et al., 2019, Kidambi et al., 2021, Viel et al., 2025] or from reward features in Linear MDPs [Moulin et al., 2025, Viano et al., 2024, 2022]. To summarize, we have seen that in the single agent setting, interactive expert does not give an advantage while knowledge of the transition or sampling access to the environment comes with two main advantages (improved horizon dependence in the tabular setting and possibility of imitation without seeing expert actions).

Strikingly, the scenario is completely swapped in the multi-agent setting. Our negative result Theorem 3.2 shows that given knowledge of the transition no significant improvements over BC can be expected. On the other side, our positive result Theorem 4.2 shows that in the interactive setting a consistent improvement is expected over BC if $\mathcal{C}(\mu^{\mathrm{E}}, \nu^{\mathrm{E}})$ is large.

**Multi-Agent Imitation Learning.**    Theoretical work in multi-agent imitation learning is limited. Existing studies mainly focus on empirical results in cooperative [Bui et al., 2024, Le et al., 2017] and adversarial [Yu et al., 2019, Song et al., 2018] settings, typically optimizing a value-gap objective, which does not capture the strategic component of multi-agent interactions. This is in contrast with the usual objective in most forward multi-agent methods which instead minimize Nash or regret gaps to measure deviations. To cite few examples, Bai and Jin [2020] learn Nash equilibria in zero-sum games with online access, Xie et al. [2020] extends the result to linear turn-based Markov Games, Liu et al. [2021], Jin et al. [2024] learn $\varepsilon$-CE, $\varepsilon$-CCE and $\varepsilon$-NE with or without suffering the curse of multi-agent respectively, Cui and Du [2022a] learn Nash equilibria in an offline manner and, finally, Bai et al. [2021] with bandit feedback and online interactions with the environment.

In imitation learning, the first to adopt the Nash gap in Normal Form Games are Waugh et al. [2011]. Recently, the Nash gap has also been considered in Imitation Learning for mean-field games [Ramponi et al., 2023]. The authors provide an upper bound for BC and adversarial Imitation Learning that is exponential in the horizon in case the dynamics and the rewards depend on the population distribution. To overcome this exponential dependency, the authors introduce a proxy to the Nash imitation gap, based on a mean field control formulation, that allows to construct an upper bound that is quadratic in the horizon. Overall, they focus on finding metrics that, if minimized they imply a small Nash gap in virtue of the above error propagation analysis. However, their work left open how the Nash Gap can

be minimized algorithmically. In this work, we take an alternative approach that works directly on upper-bounding the Nash Gap and focuses on developing an algorithmic rather than a general error propagation analysis. The closest to our work is Tang et al. [2024], extending this to finite-horizon Markov games where the observed expert data are sampled from a correlated equilibrium profile. They show that when the transition dynamics are unknown, the regret scales linearly with the horizon, even if behavior cloning successfully recovers the expert policy within the support of the expert's state distribution. To address this issue, they propose two algorithms: BLADES, which explicitly queries all single-policy deviations from the current strategy, incurring an exponential dependence on the size of the state space, and MALICE, which assumes full state coverage in the offline dataset and still incurs in a computational cost exponential in the number of states. While Tang et al. [2024] present an error propagation analysis, neither MALICE or BLADES are accompanied by a formal sample complexity analysis, leaving open questions about their statistical efficiency in practical settings.

Our work proposes an algorithm MURMAIL which is provably statistically and computationally efficient, marking a significant step forward with respect to the current literature on the topic. Moreover, on the lower bound side, we extend the construction by Tang et al. [2024] to hold even if the learner knows the transition dynamics of the game.

**Offline Zero-Sum Games.** In offline zero-sum Markov games, Cui and Du [2022a] and Zhong et al. [2022] show that learning is impossible if the dataset only covers Nash equilibrium strategies. Instead they show that a unilateral concentration is required to recover Nash equilibrium strategies. This result highlights a fundamental gap between offline learning in multi-agent versus single-agent settings. Their lower bound is constructed by considering two distinct Normal Form Games that differ only in the reward of a single joint action, resulting in different Nash equilibria. The dataset includes actions from the equilibrium and suboptimal strategies but omits data corresponding to deviations from the observed equilibrium. As a result, the two games become indistinguishable under the available data, as the missing deviations preclude disambiguation. However, this argument does not extend to the imitation learning setting, where the dataset is restricted to the (deterministic) expert policy, resulting in a perfect recoverability for their considered Normal Form Game. In this case, establishing hardness requires a more nuanced analysis that leverages the multiplicity of equilibria. Furthermore, it is important to note that their deviation coefficient is defined with respect to the maximum over all possible policies, whereas in imitation learning, it is defined only relative to the estimated Nash equilibrium strategy.

## C Proofs on BC Upper bound

In this section, we give the omitted proofs for Theorem 3.1. In the first step we state the error decomposition of the Nash Gap

$$
\begin{aligned}
V^{\mu^\star,\widehat{\nu}}(s_0) - V^{\widehat{\mu},\nu^\star} &= V^{\mu^\star,\widehat{\nu}}(s_0) - V^{\mu^{\mathrm{E}},\nu^{\mathrm{E}}}(s_0) + V^{\mu^{\mathrm{E}},\nu^{\mathrm{E}}}(s_0) - V^{\widehat{\mu},\nu^\star} \\
&\leq \underbrace{V^{\mu^\star,\widehat{\nu}}(s_0) - V^{\mu^\star,\nu^{\mathrm{E}}}(s_0)}_{:=\mathrm{Error}(\widehat{\nu})} + \underbrace{V^{\mu^{\mathrm{E}},\nu^\star}(s_0) - V^{\widehat{\mu},\nu^\star}}_{:=\mathrm{Error}(\widehat{\mu})},
\end{aligned}
$$

where $\mu^\star \in \mathrm{br}(\widehat{\nu})$ and $\nu^\star \in \mathrm{br}(\widehat{\mu})$. We can see that we can split the error into an error for the policy recovered for player 1 depending on the estimation of player 2's policy ($\mathrm{Error}(\widehat{\nu})$) and for player 1's policy respectively ($\mathrm{Error}(\widehat{\mu})$). In the following, we will only give the proofs for player 1, as the proofs for player 2 follow analogously. Next, we give a useful lemma that upper-bound the value difference in a two-player game, when one player's policy is fixed in both value functions, by the total variation. We give the general result and then apply it to our case.

**Lemma C.1.** *For any policy $\mu$ of the max-player, we have that*

$$
\left| V^{\mu,\nu}(s_0) - V^{\mu,\nu'}(s_0) \right| \leq \frac{2}{1-\gamma} \mathbb{E}_{\mu,\nu} \left[ \sum_{t=0}^{\infty} \gamma^t \mathrm{TV}\left(\nu(\cdot \mid s), \nu'(\cdot \mid s)\right) \right].
$$

*Similarly, for any policy $\nu$ of the min-player, we have that*

$$
\left| V^{\mu,\nu}(s_0) - V^{\mu',\nu}(s_0) \right| \leq \frac{2}{1-\gamma} \mathbb{E}_{\mu,\nu} \left[ \sum_{t=0}^{\infty} \gamma^t \mathrm{TV}\left(\mu(\cdot \mid s), \mu'(\cdot \mid s)\right) \right].
$$

*Proof.* Here, we only prove the first statement. The second statement can be proved by the same idea. By K.1, we have that

$$V^{\mu,\nu}(s_0) - V^{\mu,\nu'}(s_0) = \mathbb{E}_{\mu,\nu}\left[\sum_{t=0}^{\infty}\gamma^t\left(\mathbb{E}_{(a,b)\sim(\mu,\nu)}\left[Q^{\mu,\nu'}(s,a,b)\right] - \mathbb{E}_{(a,b)\sim(\mu,\nu')}\left[Q^{\mu,\nu'}(s,a,b)\right]\right)\right].$$

Applying the triangle inequality leads to

$$\left|V^{\mu,\nu}(s_0) - V^{\mu,\nu'}(s_0)\right| \le \mathbb{E}_{\mu,\nu}\left[\sum_{t=0}^{\infty}\gamma^t\left|\mathbb{E}_{(a,b)\sim(\mu,\nu)}\left[Q^{\mu,\nu'}(s,a,b)\right] - \mathbb{E}_{(a,b)\sim(\mu,\nu')}\left[Q^{\mu,\nu'}(s,a,b)\right]\right|\right].$$

For the term $\left|\mathbb{E}_{(a,b)\sim(\mu,\nu)}\left[Q^{\mu,\nu'}(s,a,b)\right] - \mathbb{E}_{(a,b)\sim(\mu,\nu')}\left[Q^{\mu,\nu'}(s,a,b)\right]\right|$, we have that

$$\left|\mathbb{E}_{(a,b)\sim(\mu,\nu)}\left[Q^{\mu,\nu'}(s,a,b)\right] - \mathbb{E}_{(a,b)\sim(\mu,\nu')}\left[Q^{\mu,\nu'}(s,a,b)\right]\right|$$
$$= \left|\sum_{a\in\mathcal{A}}\sum_{b\in\mathcal{B}}\mu(a\mid s)\left(\nu(b\mid s) - \nu'(b\mid s)\right)Q^{\mu,\nu'}(s,a,b)\right|$$
$$\le \frac{2}{1-\gamma}\sum_{a\in\mathcal{A}}\mu(a\mid s)\sum_{b\in\mathcal{B}}|\nu(b\mid s) - \nu'(b\mid s)|$$
$$= \frac{2}{1-\gamma}\text{TV}\left(\nu(\cdot\mid s),\nu'(\cdot\mid s)\right),$$

where we again applied the triangle inequality and the fact that the rewards are bounded by 1. Additionally, in the last equality we used the definition of the total variation and the fact that $\mu(\cdot\mid s)$ is a probability distribution.

Combining the obtained results we get

$$\left|V^{\mu,\nu}(s_0) - V^{\mu,\nu'}(s_0)\right| \le \frac{2}{1-\gamma}\mathbb{E}_{\mu,\nu}\left[\sum_{t=0}^{\infty}\gamma^t\text{TV}\left(\nu(\cdot\mid s),\nu'(\cdot\mid s)\right)\right],$$

which completes the proof of the first statement. $\qquad\square$

Applying the result to the BC setting and noting that by definition of the best response we have $V^{\mu^\star,\widehat{\nu}}(s_0) - V^{\mu^\star,\nu^{\text{E}}}(s_0) \ge 0$ for $\mu^\star \in \text{br}(\widehat{\nu})$ and that the result needs to hold true $\forall\mu \in \text{br}(\nu)$, we obtain

$$\text{Error}(\widehat{\nu}) = V^{\mu^\star,\widehat{\nu}}(s_0) - V^{\mu^\star,\nu^{\text{E}}}(s_0) \le \frac{2}{1-\gamma}\max_{\mu\in\text{br}(\widehat{\nu})}\mathbb{E}_{\mu,\nu^{\text{E}}}\left[\sum_{t=0}^{\infty}\gamma^t\text{TV}\left(\nu^{\text{E}}(\cdot\mid s),\widehat{\nu}(\cdot\mid s)\right)\right].$$

Similarly, for any policy $\nu$ of the min-player, we have that

$$\text{Error}(\widehat{\mu}) = V^{\mu^{\text{E}},\nu^\star}(s_0) - V^{\widehat{\mu},\nu^\star}(s_0) \le \frac{2}{1-\gamma}\max_{\nu\in\text{br}(\widehat{\mu})}\mathbb{E}_{\mu^{\text{E}},\nu}\left[\sum_{t=0}^{\infty}\gamma^t\text{TV}\left(\mu^{\text{E}}(\cdot\mid s),\widehat{\mu}(\cdot\mid s)\right)\right].$$

The reason for the additional max (and min) is that the best response map for a given policy is generally not unique, so we need to pick a distribution from that set. To make the bound apply to all possible best responses, we pick the maximum (or minimum) from the best response set.

Using the definition of the expectation and doing a change of measure, we get

$$V^{\mu^\star,\widehat{\nu}}(s_0) - V^{\mu^\star,\nu^{\text{E}}}(s_0) \le \frac{2}{1-\gamma}\max_{\mu\in\text{br}(\widehat{\nu})}\mathbb{E}_{\mu,\nu^{\text{E}}}\left[\sum_{t=0}^{\infty}\gamma^t\text{TV}\left(\nu^{\text{E}}(\cdot\mid s),\widehat{\nu}(\cdot\mid s)\right)\right]$$
$$= \frac{2}{1-\gamma}\max_{\mu\in\text{br}(\widehat{\nu})}\sum_{t=0}^{\infty}\gamma^t\sum_{s\in\mathcal{S}}\frac{d^{\mu,\nu^{\text{E}}}(s)}{d^{\mu^{\text{E}},\nu^{\text{E}}}(s)}d^{\mu^{\text{E}},\nu^{\text{E}}}(s)\text{TV}\left(\nu^{\text{E}}(\cdot\mid s),\widehat{\nu}(\cdot\mid s)\right).$$
$$\le \frac{2}{1-\gamma}\max_{\mu\in\text{br}(\widehat{\nu})}\left\|\frac{d^{\mu,\nu^{\text{E}}}}{d^{\mu^{\text{E}},\nu^{\text{E}}}}\right\|_{\infty}\mathbb{E}_{\mu^{\text{E}},\nu^{\text{E}}}\left[\sum_{t=0}^{\infty}\gamma^t\text{TV}\left(\nu^{\text{E}}(\cdot\mid s),\widehat{\nu}(\cdot\mid s)\right)\right].$$

Note that we can do the change of measure also with respect to a more general offline distribution $\rho$. Here, we use $\rho = d^{\mu^{\mathrm{E}}, \nu^{\mathrm{E}}}$. If we would use the general offline sampling distribution, we would have

$$
\begin{aligned}
V^{\mu^{\star}, \widehat{\nu}}(s_0) - V^{\mu^{\star}, \nu^{\mathrm{E}}}(s_0) &= \frac{2}{1 - \gamma} \max_{\mu \in \mathrm{br}(\widehat{\nu})} \sum_{t=0}^{\infty} \gamma^t \sum_{s \in \mathcal{S}} \frac{d^{\mu, \nu^{\mathrm{E}}}(s)}{\rho(s)} \rho(s) \mathrm{TV}\left( \nu^{\mathrm{E}}(\cdot \mid s), \widehat{\nu}(\cdot \mid s) \right) . \\
&\leq \frac{2}{1 - \gamma} \max_{\mu \in \mathrm{br}(\widehat{\nu})} \left\| \frac{d^{\mu, \nu^{\mathrm{E}}}}{\rho} \right\|_{\infty} \mathbb{E}_{\rho}\left[ \sum_{t=0}^{\infty} \gamma^t \mathrm{TV}\left( \nu^{\mathrm{E}}(\cdot \mid s), \widehat{\nu}(\cdot \mid s) \right) \right] .
\end{aligned}
$$

Next note that $\mu \in \mathrm{br}(\hat{\nu})$ depends on the estimation of the policy $\nu^{\mathrm{E}}$ and therefore is a random variable. To avoid such a term in the upper bound of the expression we make use of the following upper bound:

$$
\begin{aligned}
&\frac{2}{1 - \gamma} \max_{\mu \in \mathrm{br}(\widehat{\nu})} \left\| \frac{d^{\mu, \nu^{\mathrm{E}}}}{d^{\mu^{\mathrm{E}}, \nu^{\mathrm{E}}}} \right\|_{\infty} \mathbb{E}_{\mu^{\mathrm{E}}, \nu^{\mathrm{E}}}\left[ \sum_{t=0}^{\infty} \gamma^t \mathrm{TV}\left( \nu^{\mathrm{E}}(\cdot \mid s), \widehat{\nu}(\cdot \mid s) \right) \right] \\
&\leq \frac{2}{1 - \gamma} \max_{\nu \in \Pi} \max_{\mu \in \mathrm{br}(\nu)} \left\| \frac{d^{\mu, \nu^{\mathrm{E}}}}{d^{\mu^{\mathrm{E}}, \nu^{\mathrm{E}}}} \right\|_{\infty} \mathbb{E}_{\mu^{\mathrm{E}}, \nu^{\mathrm{E}}}\left[ \sum_{t=0}^{\infty} \gamma^t \mathrm{TV}\left( \nu^{\mathrm{E}}(\cdot \mid s), \widehat{\nu}(\cdot \mid s) \right) \right],
\end{aligned}
$$

where $\nu \in \Pi$ is any admissible policy for the second player. Taking the maximum over the best responses to this more general policy class forms an upper bound.

Additionally, note that the expectation is now over the expert policy, therefore we can use the dataset to get an estimate. Therefore, we apply the standard concentration argument to upper-bound term $\mathbb{E}_{\mu^{\mathrm{E}}, \nu^{\mathrm{E}}}\left[ \sum_{t=0}^{\infty} \gamma^t \mathrm{TV}\left( \nu^{\mathrm{E}}(\cdot \mid s), \widehat{\nu}(\cdot \mid s) \right) \right]$. By K.2 and union bound, with probability at least $1 - \delta/2$, $\forall s \in \mathcal{S}$

$$
\mathrm{TV}\left( \nu^{\mathrm{E}}(\cdot | s), \widehat{\nu}(\cdot | s) \right) \leq \sqrt{\frac{2|\mathcal{B}| \log(2|\mathcal{S}|/\delta)}{\max\{N(s), 1\}}}.
$$

Then, we can have that

$$
\begin{aligned}
\mathbb{E}_{\mu^{\mathrm{E}}, \nu^{\mathrm{E}}}\left[ \sum_{t=0}^{\infty} \gamma^t \mathrm{TV}\left( \nu^{\mathrm{E}}(\cdot | s), \widehat{\nu}(\cdot | s) \right) \right] &\leq \frac{1}{(1 - \gamma)} \sum_{s \in \mathcal{S}} d^{\mu^{\mathrm{E}}, \nu^{\mathrm{E}}}(s) \sqrt{\frac{2|\mathcal{B}| \log(2|\mathcal{S}|/\delta)}{\max\{N(s), 1\}}} \\
&= \frac{1}{(1 - \gamma)} \sum_{s \in \mathcal{S}} \sqrt{d^{\mu^{\mathrm{E}}, \nu^{\mathrm{E}}}(s)} \sqrt{\frac{2|\mathcal{B}| d^{\mu^{\mathrm{E}}, \nu^{\mathrm{E}}}(s) \log(2|\mathcal{S}|/\delta)}{\max\{N(s), 1\}}} \\
&\overset{(\mathrm{i})}{\leq} \frac{1}{(1 - \gamma)} \sqrt{\sum_{s \in \mathcal{S}} \frac{2|\mathcal{B}| d^{\mu^{\mathrm{E}}, \nu^{\mathrm{E}}}(s) \log(2|\mathcal{S}|/\delta)}{\max\{N(s), 1\}}} \\
&\overset{(\mathrm{ii})}{\leq} \frac{1}{(1 - \gamma)} \sqrt{\sum_{s \in \mathcal{S}} \frac{16|\mathcal{B}| \log^2(2|\mathcal{S}|/\delta)}{N}} \\
&= \frac{4}{(1 - \gamma)} \sqrt{\frac{|\mathcal{S}||\mathcal{B}| \log^2(2|\mathcal{S}|/\delta)}{N}},
\end{aligned}
$$

where in (i) we applied Cauchy Schwarz and in (ii) we applied Lemma K.3 and we denoted $N$ as the size of the dataset.

Finally, we obtain the policy value bound.

$$
V^{\mu^{\star}, \widehat{\nu}}(s_0) - V^{\mu^{\star}, \nu^{\mathrm{E}}}(s_0) \leq \frac{8}{(1 - \gamma)^2} \max_{\nu \in \Pi} \max_{\mu \in \mathrm{br}(\nu)} \left\| \frac{d^{\mu, \nu^{\mathrm{E}}}}{d^{\mu^{\mathrm{E}}, \nu^{\mathrm{E}}}} \right\|_{\infty} \sqrt{\frac{|\mathcal{S}| |\mathcal{B}| \log^2(2|\mathcal{S}|/\delta)}{N}}
$$

and doing the same analysis for player 2 we get

$$V^{\mu^{\mathrm{E}},\nu^\star}(s_0) - V^{\widehat{\mu},\nu^\star}(s_0) \leq \frac{8}{(1-\gamma)^2} \max_{\mu \in \Pi} \max_{\nu \in \mathrm{br}(\mu)} \left\| \frac{d^{\mu^{\mathrm{E}},\nu}}{d^{\mu^{\mathrm{E}},\nu^{\mathrm{E}}}} \right\|_\infty \sqrt{\frac{|\mathcal{S}|\,|\mathcal{A}|\log^2(2\,|\mathcal{S}|/\delta)}{N}}.$$

Finally, by using the error decomposition derived in the first step, defining $\mathcal{C}_{\max} :=$ $\max\left\{ \max_{\nu \in \Pi} \max_{\mu \in \mathrm{br}(\nu)} \left\| \frac{d^{\mu,\nu^{\mathrm{E}}}}{\rho} \right\|_\infty, \; \max_{\mu \in \Pi} \max_{\nu \in \mathrm{br}(\mu)} \left\| \frac{d^{\mu^{\mathrm{E}},\nu}}{\rho} \right\|_\infty \right\}$ with $\rho = d^{\mu^{\mathrm{E}},\nu^{\mathrm{E}}}$ and use that $|\mathcal{A}_{\max}| = \max\{|\mathcal{A}|,|\mathcal{B}|\}$ we obtain with probability of at least $1 - \delta$

$$V^{\mu^\star,\widehat{\nu}}(s_0) - V^{\widehat{\mu},\nu^\star}(s_0) \leq \mathcal{C}_{\max}\frac{8}{(1-\gamma)^2} \sqrt{\frac{|\mathcal{S}|\,|\mathcal{A}_{\max}|\log^2(2\,|\mathcal{S}|/\delta)}{N}}$$

completing the proof of Theorem 3.1.

## D  Proof for necessity of $\mathcal{C}(\mu^{\mathrm{E}},\nu^{\mathrm{E}})$

In this section we give the proof of Theorem 3.2, showing the necessity of a bounded $\mathcal{C}(\mu^{\mathrm{E}},\nu^{\mathrm{E}})$ for non-interactive imitation learning even if the learner is fully-aware of the transition model. For a better understanding of the following proof it is essential to remind ourselves of a (simplified) Markov Game hardness construction introduced in Section 3.1 and illustrated in Fig. 1.

*Proof of Theorem 3.2.* Let us consider the following family of Zero-Sum Markov Games $\mathcal{G}_\infty = \{\mathcal{G}_i\}_{i=1}^{|\mathcal{A}|}$ with action spaces of the same cardinality $|\mathcal{A}| \geq 3$ given by $\mathcal{A} = \{a_1, a_2 \ldots, a_{i-1}, a_E, a_{i+1}, \ldots, a_{|\mathcal{A}|}\}$, $\mathcal{B} = \{b_1, b_2 \ldots, b_{i-1}, b_E, b_{i+1}, \ldots, b_{|\mathcal{A}|}\}$ and a shared state space for both agents $\mathcal{S} = \{s_0, s_1, s_2, s_3, s'_{3_1}, \ldots, s'_{3_{2|\mathcal{A}|-1}}, s_{\mathrm{xplt1}_1}, \ldots, s_{\mathrm{xplt1}_{(|\mathcal{A}|-1)^2/2}}, s_{\mathrm{xplt2}_1}, \ldots, s_{\mathrm{xplt2}_{(|\mathcal{A}|-1)^2/2}}\}$. From now on we can divide the state space into 5 parts. We have $S_{\mathrm{expert}} := \{s_0, s_2, s_3\}$ the states visited by the expert, $s_1$ the gating state, $S_{\mathrm{xplt1}} := \{s_{\mathrm{xplt1}_1}, \ldots, s_{\mathrm{xplt1}_{(|\mathcal{A}|-1)^2/2}}\}$ the states where player 1 can be exploited, $S_{\mathrm{xplt2}} := \{s_{\mathrm{xplt2}_1}, \ldots, s_{\mathrm{xplt2}_{(|\mathcal{A}|-1)^2/2}}\}$ the states where player 2 can be exploited and $S_{\mathrm{copy}} := \{s'_{3_1}, \ldots, s'_{3_{2|\mathcal{A}|-1}}\}$ that are copies of the final states visited by the expert in the sense that they are sharing the same reward. Additionally, consider a dirac on state $s_0$ as an initial state distribution, i.e. $d_0(s) = \delta_{s_0}$. The transition model is deterministic and in all states except $s_0$ and $s_1$, simply a transition to one neighboring state. Therefore, we only give a detailed description for these two states. And for the gating state $s_1$, we only consider the next potential set of states, as states inside these sets share the same reward function and each action pair leads to a unique state inside these sets, i.e. every action combination leads to a different state. In particular, we have

$$P_{\mathcal{G}_i}(\cdot \mid s_0, a, b) = \begin{cases} s_2 & \text{if } (a,b) = a_i b_i, \\ s_1 & \text{otherwise.} \end{cases}$$

$$P_{\mathcal{G}_i}(\cdot \mid s_1, a, b) = \begin{cases} S_{\mathrm{copy}}, & \text{if } (a,b) \in \{(a_i,b_i)\} \cup \{(a_i,b_j),(a_j,b_i) \mid \forall j \neq i\}, \\ S_{\mathrm{xplt1}}, & \text{if } (a,b) \in \mathcal{E}_{p,q} \\ S_{\mathrm{xplt2}}, & \text{otherwise,} \end{cases}$$

where $\mathcal{E}_{p,q} := \{(a_j,b_j) \mid \forall j \neq i\} \cup \{(a_{j_p},b_{j_q}) \mid 1 \leq p < q \leq n-1,\}$ and $j_p, j_q \in \{1,\ldots,|\mathcal{A}|\} \setminus \{i\}$. The state-only reward, which equals across all games inside the sets $\mathcal{G}_i \in \mathcal{G}_{\mathrm{conc}}^E$ is given by

$$R_1(s) = \begin{cases} 1 & \text{if } s \in S_{\mathrm{explt2}}, \\ -1 & \text{if } s \in S_{\mathrm{explt1}}, \\ 0 & \text{otherwise.} \end{cases}$$

As the considered Markov Game is a Zero-Sum Game, it holds that $R_2(s) = -R_1(s)$. While the transition dynamic looks complicated, the two important things to keep in mind that, once an agent differs from action $a_i$ and $b_i$ respectively, an action can be chosen in such a way, that the following state is inside $S_{\mathrm{xplt1}}$ or $S_{\mathrm{xplt2}}$ respectively and all action combinations has a unique follow up state, i.e. $|S_{\mathrm{copy}}| \cup |S_{\mathrm{xplt1}}| \cup |S_{\mathrm{xplt2}}| = |\mathcal{A}|\,|\mathcal{B}|$.

It follows that the actions taken by the agents only matter in the states $s_0$ and $s_1$. For these states we consider the following Nash equilibrium expert policy $\mu^{\mathrm{E}}(a_i \mid s_0) = \nu^{\mathrm{E}}(b_i \mid s_0) = 1$ and $\mu^{\mathrm{E}}(a_i \mid s_1) = \nu^{\mathrm{E}}(b_i \mid s_1) = 1$. It follows immediately, that the given policy is indeed a Nash equilibrium as no single agent benefits by deviating. Additionally, note that for all actions $a_j \forall j \neq i$, each player is exploitable in $s_1$.

An illustration of the described Markov Game for $|\mathcal{A}| = |\mathcal{B}| = 3$ can be found in Fig. 1, where $a_i = a_3$ and $b_i = b_3$.

Next, note that for all $\mathcal{G}_i \in \mathcal{G}_\infty$ it indeed holds that $\mathcal{C}(\mu^{\mathrm{E}}, \nu^{\mathrm{E}}) = \infty$ as fixing for example policy $\nu^{\mathrm{E}}$ another best response for player, i.e. $\mu_2^{\mathrm{E}} \in \mathrm{br}(\nu^{\mathrm{E}})$ is given by $\mu_2^{\mathrm{E}}(a_j \mid s_0) = 1$, for $j \neq i$ and $\mu_2^{\mathrm{E}}(a_i \mid s_1) = \mu^{\mathrm{E}}(a_i \mid s_1) = 1$, meaning that the only states visited by the policy pair $(\mu_2^{\mathrm{E}}, \nu^{\mathrm{E}})$ are $s_0, s_1, s_4'$.

Now we show that for the family of Markov Games $\mathcal{G}_\infty$ where for each $\mathcal{G}_i \in \mathcal{G}_\infty$ it holds that $\mathcal{C}(\mu^{\mathrm{E}}, \nu^{\mathrm{E}}) = \infty$, any learning algorithm Alg has a Nash Gap of the order $(1 - \gamma)^{-1}$. For that let $(\widehat{\mu}, \widehat{\nu})$ be the output of any non-interactive imitation learning algorithm Alg with data from $(\mu^{\mathrm{E}}, \nu^{\mathrm{E}})$. It is important to observe that since all games in $\mathcal{G}_\infty$ are identical in $s_0$ and they differs only in transition dynamics from $s_1$. However, no information about $s_1$ is available in $\mathcal{D}$. Therefore, the learner has no mean to distinguish which game she is facing. For this reason $\widehat{\mu}, \widehat{\nu}$ do not depend on the game index $i$. Then denoting by $A_{\mathrm{Alg}}$ and $B_{\mathrm{Alg}}$ the action played by the learner in the state $s_1$, it holds true that

$$\max_{\mathcal{G}_i \in \mathcal{G}_\infty} V_{\mathcal{G}_i}^{\mu^\star, \widehat{\nu}}(s_0) - V_{\mathcal{G}_i}^{\widehat{\mu}, \nu^\star}(s_0)$$

$$\geq \frac{\sum_{i=1}^{\mathcal{A}} V_{\mathcal{G}_i}^{\mu^\star, \widehat{\nu}}(s_0) - V_{\mathcal{G}_i}^{\widehat{\mu}, \nu^\star}(s_0)}{|\mathcal{A}|}$$

$$\overset{(i)}{=} \frac{1}{(1 - \gamma)} \left( \frac{\sum_{i=1}^{|\mathcal{A}|} \mathbb{P}(B_{\mathrm{Alg}} \neq b_i)}{|\mathcal{A}|} + \frac{\sum_{i=1}^{|\mathcal{A}|} \mathbb{P}(A_{\mathrm{Alg}} \neq a_i)}{|\mathcal{A}|} \right)$$

$$= \frac{1}{(1 - \gamma)} \left( \frac{\sum_{i=1}^{|\mathcal{A}|} (1 - \widehat{\nu}(b_i \mid s_1))}{|\mathcal{A}|} + \frac{\sum_{i=1}^{|\mathcal{A}|} (1 - \widehat{\mu}(a_i \mid s_1))}{|\mathcal{A}|} \right)$$

$$= \frac{1}{(1 - \gamma)} \left( \frac{|\mathcal{A}| - 1}{|\mathcal{A}|} + \frac{|\mathcal{A}| - 1}{|\mathcal{A}|} \right)$$

$$\geq \frac{1}{1 - \gamma},$$

where $(i)$ follows from the construction of $\mathcal{G}_i \in \mathcal{C}(\mu^{\mathrm{E}}, \nu^{\mathrm{E}})$, as all actions, but actions $a_i, b_i$ are exploitable by the opponent in the game $\mathcal{G}_i$.

Additionally, note that even if the learner has access to the transition dynamics, the learner can not differentiate the actions from $s_1$ as all actions lead to different states. Therefore, she cannot use this knowledge to recover an action that would lead to $s \in S_{\mathrm{copy}}$, which would avoid a regret of the order $(1 - \gamma)^{-1}$. This completes the proof.

$\square$

# E    Analysis of BR Oracle Algorithm

This section presents the analysis of Algorithm 1 which provides a sample complexity guarantee without requiring single deviation concentrability under the assumption of a Best Response Oracle (Definition 4.1). The difference to the later presented algorithm MURMAIL Algorithm 2 is that here we can query the best response oracle to sample from the expectation that contains the best response of the current policy $\mu_k$ and $\nu_k$ respectively. In particular, we sample a state from the induced discounted state distributions. However, as noted in our discussion around Eq. (3) we first have to transform the optimization problem into a convex one as the original objective of our optimization problem is non-convex. This results in Eq. (4), from where we can obtain a Martingale difference

sequence and a regret term, which then can be minimized as we now can construct an unbiased gradient estimator and use a version of online mirror descent to construct an update of our policies.

Now, we restate the theorem and give the complete proof.

**Theorem 4.1.** *Let us run Algorithm 1 for* $K = \mathcal{O}\left(\frac{|\mathcal{S}||\mathcal{A}_{\max}|^2 \log|\mathcal{A}_{\max}| \log(1/\delta)}{(1-\gamma)^4 \varepsilon^4}\right)$ *iterations with learning rate* $\eta = \frac{2|\mathcal{S}| \log|\mathcal{A}_{\max}|}{K}$. *Then, the sequence of policies* $\{\mu_k, \nu_k\}_{k=1}^K$ *satisfies with probability at least* $1 - 5\delta$ *that* $\frac{1}{K}\sum_{k=1}^K \max_{\mu \in \Pi} \langle d_0, V^{\mu,\nu_k}\rangle - \min_{\nu \in \Pi} \langle d_0, V^{\mu_k,\nu}\rangle \leq \mathcal{O}(\varepsilon)$. *Therefore, setting* $\delta = \mathcal{O}(\varepsilon)$ *ensures that for a certain* $\hat{k} \sim \mathrm{Unif}([K])$ *it holds that* $\mathbb{E}\left[\mathrm{Nash-Gap}(\mu_{\hat{k}}, \nu_{\hat{k}})\right] \leq \varepsilon$. *That is,* $\mu_{\hat{k}}, \nu_{\hat{k}}$ *is an* $\varepsilon$-Nash equilibrium in expectation.

*Proof.* In the first step, we square the optimization problem and decompose into a part that considers policy $\mu_k$ for player 1 and $\nu_k$ for player 2.

$$
\left(\frac{1}{K}\sum_{k=1}^K \max_{\mu \in \Pi} \langle d_0, V^{\mu,\nu_k}\rangle - \min_{\nu \in \Pi} \langle d_0, V^{\mu_k,\nu}\rangle\right)^2 = \left(\frac{1}{K}\sum_{k=1}^K \left\langle d_0, V^{\mu_k^\star,\nu_k} - V^{\mu_k,\nu_k^\star}\right\rangle\right)^2
$$

$$
= \left(\frac{1}{K}\sum_{k=1}^K \left\langle d_0, V^{\mu_k^\star,\nu_k} - V^{\mu_E,\nu_E} + V^{\mu_E,\nu_E} - V^{\mu_k,\nu_k^\star}\right\rangle\right)^2
$$

$$
\leq \left(\frac{1}{K}\sum_{k=1}^K \left\langle d_0, V^{\mu_k^\star,\nu_k} - V^{\mu_k^\star,\nu_E} + V^{\mu_E,\nu_k^\star} - V^{\mu_k,\nu_k^\star}\right\rangle\right)^2
$$

$$
\leq 2\left(\frac{1}{K}\sum_{k=1}^K \left\langle d_0, V^{\mu_k^\star,\nu_k} - V^{\mu_k^\star,\nu_E}\right\rangle\right)^2 + 2\left(\frac{1}{K}\sum_{k=1}^K \left\langle d_0, V^{\mu_E,\nu_k^\star} - V^{\mu_k,\nu_k^\star}\right\rangle\right)^2, \tag{5}
$$

where $\mu_k^\star \in \mathrm{br}(\nu_k)$ and $\nu_k^\star \in \mathrm{br}(\mu_k)$. At this point, dividing by $K$ squaring and applying the performance difference Lemma K.1 that allows to decompose the global regret into a weighted some of regrets at each state, we obtain

$$
\left(\frac{1}{K}\sum_{k=1}^K \left\langle d_0, V^{\mu_E,\nu_k^\star} - V^{\mu_k,\nu_k^\star}\right\rangle\right)^2
$$

$$
= \left(\frac{1}{K}\sum_{k=1}^K \mathbb{E}_{s \sim d^{\mu_k,\nu_k^\star}}\left[\left\langle \mathbb{E}_{b \sim \nu_k^\star(\cdot|s)} Q^{\mu_E,\nu_k^\star}(s,\cdot,b), \mu_E(\cdot|s) - \mu_k(\cdot|s)\right\rangle\right]\right)^2
$$

$$
\leq \left(\frac{1}{K}\sum_{k=1}^K \mathbb{E}_{s \sim d^{\mu_k,\nu_k^\star}}\left[\|\mu_E(\cdot|s) - \mu_k(\cdot|s)\| \sqrt{|\mathcal{A}_{\max}|}(1-\gamma)^{-1}\right]\right)^2
$$

$$
\leq \frac{|\mathcal{A}_{\max}|}{K(1-\gamma)^2}\sum_{k=1}^K \mathbb{E}_{s \sim d^{\mu_k,\nu_k^\star}}\left[\|\mu_E(\cdot|s) - \mu_k(\cdot|s)\|^2\right] \tag{6}
$$

where the second last step used the Cauchy-Schwarz inequality and the last step used the Jensen's inequality and the concavity of the square root. Analogous steps give

$$
\left(\frac{1}{K}\sum_{k=1}^K \left\langle d_0, V^{\mu_k^\star,\nu_k} - V^{\mu_k^\star,\nu_E}\right\rangle\right)^2 \leq \frac{|\mathcal{A}_{\max}|}{K(1-\gamma)^2}\sum_{k=1}^K \mathbb{E}_{s \sim d^{\mu_k^\star,\nu_k}}\left[\|\nu_E(\cdot|s) - \nu_k(\cdot|s)\|^2\right]. \tag{7}
$$

At this point, we see that the expectation is over the best response and the current policies of iteration $k$. Therefore, we now make use of the Best Response Oracle to sample $S_k^\mu \sim d^{\mu_k,\nu_k^\star}$ and $S_k^\nu \sim d^{\mu_k^\star,\nu_k}$. Next, we can add and subtract the terms $\|\mu_E(\cdot|S_k^\mu) - \mu_k(\cdot|S_k^\mu)\|^2$ in (6) and $\|\nu_E(\cdot|S_k^\nu) - \nu_k(\cdot|S_k^\nu)\|^2$

in (7), and we obtain that

$$\left(\frac{1}{K}\sum_{k=1}^{K}\left\langle d_0, V^{\mu_E,\nu_k^\star} - V^{\mu_k,\nu_k^\star}\right\rangle\right)^2$$

$$\leq \frac{|\mathcal{A}_{\max}|}{K(1-\gamma)^2}\sum_{k=1}^{K}\left(\mathbb{E}_{s\sim d^{\mu_k},y_k}\left[\|\mu_E(\cdot|s) - \mu_k(\cdot|s)\|^2\right] - \|\mu_E(\cdot|S_k^\mu) - \mu_k(\cdot|S_k^\mu)\|^2\right)$$

(Martingale)

$$+ \frac{|\mathcal{A}_{\max}|}{K(1-\gamma)^2}\sum_{k=1}^{K}\|\mu_E(\cdot|S_k^\mu) - \mu_k(\cdot|S_k^\mu)\|^2.$$

(Regret)

We have that (Martingale) can be bounded as follows via Azuma-Hoeffding inequality (see e.g. Theorem 7.2.1 in [Alon and Spencer, 2004]). In particular, define $X_k = \mathbb{E}_{s\sim d^{\mu_k},y_k}\left[\|\mu_E(\cdot|s) - \mu_k(\cdot|s)\|^2\right] - \|\mu_E(\cdot|S_k^\mu) - \mu_k(\cdot|S_k^\mu)\|^2$, notice that $\{X_k\}_{k=1}^{K}$ is a martingale difference sequence almost surely bounded by 2. Therefore, it holds with probability at least $1-\delta$ that

$$\sum_{k=1}^{K}\left(\mathbb{E}_{s\sim d^{\mu_k},y_k}\left[\|\mu_E(\cdot|s) - \mu_k(\cdot|s)\|^2\right] - \|\mu_E(\cdot|S_k^\mu) - \mu_k(\cdot|S_k^\mu)\|^2\right) \leq \sqrt{K\log(1/\delta)}$$

Finally, for (Regret) let us define the loss $\ell_k(\mu) = \|\mu_E(\cdot|S_k^\mu) - \mu(\cdot|S_k^\mu)\|^2$ and notice that $\ell_k(\mu_E) = 0$. Therefore, by convexity of $\ell_k$,

$$(\text{Regret}) = \sum_{k=1}^{K}\ell_k(\mu_k) - \ell_k(\mu_E) \leq \sum_{k=1}^{K}\langle\nabla_\mu\ell_k(\mu_k), \mu_k - \mu_E\rangle$$

where we have that $\nabla_\mu\ell_k(\mu_k) = \left[\nabla_{\mu(\cdot|s_1)}\ell_k(\mu_k)^T, \ldots, \nabla_{\mu(\cdot|s_{|\mathcal{S}|})}\ell_k(\mu_k)^T\right]^T$ and the gradients with respect to a policy evaluated at a particular state are given as

$$\nabla_{\mu(\cdot|s)}\ell_k(\mu_k) = \begin{cases} \mu_k(\cdot|s) - \mu_E(\cdot|s) & \text{if } s = S_k^\mu \\ 0 & \text{otherwise} \end{cases}.$$

Since we do not have complete knowledge of the expert policy but only sampling access to it, we need to introduce the stochastic gradient estimator $g_k^\mu$. To this end, we sample an action $A_k^\mu \sim \mu_E(\cdot|S_k^\mu)$ and we define the following gradient estimator

$$g_k^\mu = \begin{cases} \mu_k(\cdot|s) - \mathbf{e}_{A_k^\mu} & \text{if } s = S_k^\mu \\ 0 & \text{otherwise} \end{cases}.$$

Notice that $g_k^\mu$ is unbiased and we have that $\left\|g_k^\mu - \nabla_{\mu(\cdot|s)}\ell_k(\mu_k)\right\| \leq \left\|\mathbf{e}_{A_k^\mu} - \mu_E(\cdot|S_k^\mu)\right\| \leq \sqrt{2}$. Therefore, the sequence $\{Y_k\}_{k=1}^{K}$ where $Y_k = \langle\nabla_\mu\ell_k(\mu_k) - g_k^\mu, \mu_k - \mu_E\rangle$ is a martingale difference sequence adapted to the filtration $\mathcal{F}_t$ which includes all the algorithmic randomness up to the generation of $\mu_k$. Indeed we have that $\mathbb{E}[Y_t|\mathcal{F}_t] = 0$ and

$$\mathbb{E}\left[Y_t^2|\mathcal{F}_t\right] \leq \mathbb{E}\left[\left\|\mathbf{e}_{A_k^\mu} - \mu_E(\cdot|S_k^\mu)\right\|^2\|\mu_k - \mu_E\|^2\Big|\mathcal{F}_t\right] \leq 4|\mathcal{S}|.$$

Therefore, thanks to an application of the Azuma-Hoeffding inequality, with probability at least $1-\delta$

$$\sum_{k=1}^{K}\langle\nabla_\mu\ell_k(\mu_k) - g_k^\mu, \mu_k - \mu_E\rangle \leq \sqrt{2K|\mathcal{S}|\log(1/\delta)}.$$

Therefore, we can bound the regret as follows

$$\sum_{k=1}^{K} \langle \nabla_\mu \ell_k(\mu_k), \mu_k - \mu_E \rangle = \sum_{k=1}^{K} \langle g_k^\mu, \mu_k - \mu_E \rangle + \sum_{k=1}^{K} \langle \nabla_\mu \ell_k(\mu_k) - g_k^\mu, \mu_k - \mu_E \rangle$$

$$\leq \frac{|\mathcal{S}| \log \mathcal{A}}{\eta} + \frac{\eta}{2} \sum_{k=1}^{K} \left\| \mu(\cdot | S_k^\mu) - \mathbf{e}_{A_k^\mu} \right\|_\infty + \sqrt{2K |\mathcal{S}| \log(1/\delta)}$$

$$\leq \frac{|\mathcal{S}| \log |\mathcal{A}_{\max}|}{\eta} + \frac{\eta K}{2} + \sqrt{2K |\mathcal{S}| \log(1/\delta)}$$

$$\leq \sqrt{\frac{K |\mathcal{S}| \log |\mathcal{A}_{\max}|}{2}} + \sqrt{2K |\mathcal{S}| \log(1/\delta)},$$

where for the first term, we recognized that the policies updates in Algorithm 1 can be seen as mirror descent updates (see Lemma E.1) and we used the standard regret bound for online mirror descent (see for example Orabona [2023]) instantiated with the following Bregman divergence $\sum_{s \in \mathcal{S}} D_{KL}(\mu(\cdot|s), \mu'(\cdot|s))$ and with learning rate $\eta = \frac{2|\mathcal{S}| \log |\mathcal{A}_{\max}|}{K}$ as done in Algorithm 1. All in all, we obtain via a union bound that with probability at least $1 - 4\delta$

$$\left( \frac{1}{K} \sum_{k=1}^{K} \left\langle d_0, V^{\mu_E, \nu_k^\star} - V^{\mu_k, \nu_k^\star} \right\rangle \right)^2$$

$$\leq \frac{|\mathcal{A}_{\max}|}{(1-\gamma)^2} \left( \sqrt{\frac{(2|\mathcal{S}| + 1) \log(1/\delta)}{K}} + \sqrt{\frac{|\mathcal{S}| \log |\mathcal{A}_{\max}|}{2K}} \right).$$

Moreover, analogous calculations give

$$\left( \frac{1}{K} \sum_{k=1}^{K} \left\langle d_0, V^{\mu_k^\star, \nu_k} - V^{\mu_k^\star, \nu_E} \right\rangle \right)^2$$

$$\leq \frac{|\mathcal{A}_{\max}|}{(1-\gamma)^2} \left( \sqrt{\frac{(2|\mathcal{S}| + 1) \log(1/\delta)}{K}} + \sqrt{\frac{|\mathcal{S}| \log |\mathcal{A}_{\max}|}{2K}} \right).$$

Then, plugging into (5), using a union bound and taking square root on both sides, we obtain via another union bound that with probability at least $1 - 5\delta$

$$\frac{1}{K} \sum_{k=1}^{K} \max_{\mu \in \Pi} \langle d_0, V^{\mu, \nu_k} \rangle - \min_{\nu \in \Pi} \langle d_0, V^{\mu_k, \nu} \rangle$$

$$\leq \sqrt{\frac{4|\mathcal{A}_{\max}|}{(1-\gamma)^2} \left( \sqrt{\frac{(2|\mathcal{S}| + 1) \log(1/\delta)}{K}} + \sqrt{\frac{|\mathcal{S}| \log |\mathcal{A}_{\max}|}{2K}} \right)}.$$

At this point, setting $K = \mathcal{O}\left( \frac{|\mathcal{S}| |\mathcal{A}_{\max}|^2 \log |\mathcal{A}_{\max}| \log(1/\delta)}{(1-\gamma)^4 \varepsilon^4} \right)$ ensures that with probability at least $1 - 5\delta$

$$\frac{1}{K} \sum_{k=1}^{K} \max_{\mu \in \Pi} \langle d_0, V^{\mu, \nu_k} \rangle - \min_{\nu \in \Pi} \langle d_0, V^{\mu_k, \nu} \rangle \leq \mathcal{O}(\varepsilon).$$

Therefore, the total number of expert queries is $\mathcal{O}(K) = \mathcal{O}\left( \frac{|\mathcal{S}| |\mathcal{A}_{\max}|^2 \log |\mathcal{A}_{\max}| \log(1/\delta)}{(1-\gamma)^4 \varepsilon^4} \right)$.  $\square$

The next results shows that the policies updates used in Algorithm 1 and Algorithm 2 are mirror descent updates for an appropriately chosen Bregman divergence. To this end for any $p, d \in \Delta_{\mathcal{A}}$ we define the KL divergence as $KL(p, q) = \sum_{a \in \mathcal{A}} p(a) \log(p(a)/q(a))$ with the convention that $KL(p, q) = 0$ if there exists an action $a$ such that $p(a) = 0$ and $q(a) > 0$.

**Lemma E.1.** *The updates used in Algorithm 1 and Algorithm 2, that is*

$$\mu_{k+1}(a \mid s) \propto \mu_k(a \mid s) \exp\left(-\eta g_k^\mu(s,a)\right) \quad \nu_{k+1}(b \mid s) \propto \nu_k(b \mid s) \exp\left(-\eta g_k^\nu(s,a)\right)$$

*are equivalent to mirror descent updates for the Bregman divergence $\sum_{s\in\mathcal{S}} KL(\mu(\cdot|s), \mu_k(\cdot|s))$. That is, the updates can be equivalently rewritten as*

$$\mu_{k+1} = \operatorname*{argmin}_{\mu\in\Pi} \langle g_k^\mu, \mu\rangle + \frac{1}{\eta}\sum_{s\in\mathcal{S}} KL(\mu(\cdot|s), \mu_k(\cdot|s))$$

*and*

$$\nu_{k+1} = \operatorname*{argmin}_{\nu\in\Pi} \langle g_k^\nu, \nu\rangle + \frac{1}{\eta}\sum_{s\in\mathcal{S}} KL(\nu(\cdot|s), \nu_k(\cdot|s))$$

*Proof.* We prove the result for one player ( the $\mu$ player ). The result for the other player would follow exactly the same steps. Let us consider the proximal update

$$\mu_{k+1} = \operatorname*{argmin}_{\mu\in\Pi} \langle g_k^\mu, \mu\rangle + \frac{1}{\eta}\sum_{s\in\mathcal{S}} KL(\mu(\cdot|s), \mu_k(\cdot|s))$$

The Bregman divergence chosen is induced by the function $\psi(\mu) = \sum_{s\in\mathcal{S}}\sum_{a\in\mathcal{A}} \mu(a|s)\log\mu(a|s)$ sum of the negative entropy over the state space. Notice that the gradient norm tends to infinite as $\mu$ approaches the border of the policy space (i.e. some entries $\mu(a|s)$ tends to zero), that is $\lim_{\mu\to\delta\Pi}\|\nabla\psi(\mu)\|$. This means that the first order optimality condition implies that the derivative $F(\mu) = \langle g_k^\mu, \mu\rangle + \frac{1}{\eta}\sum_{s\in\mathcal{S}} KL(\mu(\cdot|s), \mu_k(\cdot|s))$ equals zero at the minimizing policy which is $\mu_{k+1}$ by definition. Therefore, in the following we use this fact to derive the exponential weight updates used in Algorithm 1 and 2.

$$\nabla F(\mu_{k+1}) = 0 \implies g_k^\mu(s,a) + \frac{1}{\eta}\log\left(\frac{\mu_{k+1}(a|s)}{\mu_k(a|s)}\right) = c$$

for some $c \in \mathbb{R}$ which ensures normalization of $\mu_{k+1}$. Therefore, inverting the last expression, we get

$$\mu_{k+1}(a|s) = \mu_k(a|s)\exp\left(\eta(c - g_k^\mu(s,a))\right)$$

Choosing $c \in \mathbb{R}$ to ensure that $\forall s \in \mathcal{S}$ it holds that $\sum_{a\in\mathcal{A}} \mu_{k+1}(a|s) = 1$ concludes the proof. $\qquad\square$

# F  Analysis of Algorithm 2

This section presents the analysis of Algorithm 2 which provides a sample complexity guarantee without requiring neither concentrability or best response oracle.

**Theorem 4.2.** *Let us run Algorithm 2 for $K = \mathcal{O}\left(\frac{|\mathcal{S}||\mathcal{A}_{\max}|^2 \log|\mathcal{A}_{\max}|\log(1/\varepsilon)}{(1-\gamma)^4\varepsilon^4}\right)$ outer iterations and $T = \mathcal{O}\left(\frac{|\mathcal{S}|^3|\mathcal{A}_{\max}|^3\log(1/\varepsilon)}{(1-\gamma)^8\varepsilon^4}\right)$ inner iterations with learning rate $\eta = \frac{2|\mathcal{S}|\log|\mathcal{A}_{\max}|}{K}$. Then, for a certain $\hat{k} \sim \mathrm{Unif}([K])$ it holds that $\mathbb{E}\left[\mathrm{Nash-Gap}(\mu_{\hat{k}}, \nu_{\hat{k}})\right] \leq \varepsilon$.*

*Proof.* The proof follows similar to the one of Theorem 4.1 with the addition of an RL inner loop. In a first step, we first derive the same decomposition as in (5). Again, dividing by $K$ squaring, applying the performance difference, the Cauchy-Schwartz inequality leads to (6) and using Jensen's inequality and the concavity of the square root we get

$$\left(\frac{1}{K}\sum_{k=1}^K \left\langle d_0, V^{\mu_E, \nu_k^\star} - V^{\mu_k, \nu_k^\star}\right\rangle\right)^2$$

$$\leq \frac{|\mathcal{A}_{\max}|}{K(1-\gamma)^2}\sum_{k=1}^K \mathbb{E}_{s\sim d^{\mu_k, \nu_k^\star}}\left[\|\mu_E(\cdot|s) - \mu_k(\cdot|s)\|^2\right], \tag{8}$$

where $\nu_k^\star \in \mathrm{br}(\mu_k)$ and analogously for $\nu_k$

$$\left(\frac{1}{K}\sum_{k=1}^K \left\langle d_0, V^{\mu_k^\star, \nu_k} - V^{\mu_k^\star, \nu_E}\right\rangle\right)^2 \leq \frac{|\mathcal{A}_{\max}|}{K(1-\gamma)^2}\sum_{k=1}^K \mathbb{E}_{s\sim d^{\mu_k^\star, \nu_k}}\left[\|\nu_E(\cdot|s) - \nu_k(\cdot|s)\|^2\right], \tag{9}$$

where $\mu_k^\star \in \mathrm{br}(\nu_k)$. At this point, as we do not assume to have access to a Best Response oracle, let us introduce the sequence $\{z_k\}_{k=1}^K$ and $\{y_k\}_{k=1}^K$ produced by UCB-VI in the inner loop of Algorithm 2. Since the stochastic reward used in the inner loop is unbiased and almost surely bounded by 2 by Lemma G.7, Lemma G.6 run for a number of inner iterations $T = \mathcal{O}\left(\frac{|\mathcal{S}|^3|\mathcal{A}_{\max}|\log(1/\delta)}{(1-\gamma)^4\varepsilon_{\mathrm{opt}}^2}\right)$ ensures that with probability $1 - 3\delta$

$$\mathbb{E}_{s\sim d^{\mu_k^\star,\nu_k}}\left[\|\nu_E(\cdot|s) - \nu_k(\cdot|s)\|^2\right] \le \mathbb{E}_{s\sim d^{z_k,\nu_k}}\left[\|\nu_E(\cdot|s) - \nu_k(\cdot|s)\|^2\right] + \varepsilon_{\mathrm{opt}} \tag{10}$$

and

$$\mathbb{E}_{s\sim d^{\mu_k,\nu_k^\star}}\left[\|\mu_E(\cdot|s) - \mu_k(\cdot|s)\|^2\right] \le \mathbb{E}_{s\sim d^{\mu_k,y_k}}\left[\|\mu_E(\cdot|s) - \mu_k(\cdot|s)\|^2\right] + \varepsilon_{\mathrm{opt}} \tag{11}$$

Note that now we can sample $S_k^\mu \sim d^{\mu_k,y_k}$ and $S_k^\nu \sim d^{z_k,\nu_k}$. Therefore, we can again add and subtract the terms $\|\mu_E(\cdot|S_k^\mu) - \mu_k(\cdot|S_k^\mu)\|^2$ in (8) and $\|\nu_E(\cdot|S_k^\nu) - \nu_k(\cdot|S_k^\nu)\|^2$ in (9) we obtain that

$$\left(\frac{1}{K}\sum_{k=1}^K\left\langle d_0, V^{\mu_E,\nu_k^\star} - V^{\mu_k,\nu_k^\star}\right\rangle\right)^2$$

$$\le \frac{|\mathcal{A}_{\max}|}{K(1-\gamma)^2}\sum_{k=1}^K\left(\mathbb{E}_{s\sim d^{\mu_k,y_k}}\left[\|\mu_E(\cdot|s) - \mu_k(\cdot|s)\|^2\right] - \|\mu_E(\cdot|S_k^\mu) - \mu_k(\cdot|S_k^\mu)\|^2\right) \tag{12}$$

$$+ \frac{|\mathcal{A}_{\max}|}{K(1-\gamma)^2}\sum_{k=1}^K\|\mu_E(\cdot|S_k^\mu) - \mu_k(\cdot|S_k^\mu)\|^2 \tag{Regret}$$

$$+ \frac{|\mathcal{A}_{\max}|}{(1-\gamma)^2}\varepsilon_{\mathrm{opt}} \tag{Inner RL Loop Error}$$

We have that (12) can be bounded analogously as in Eq. (Martingale) with $S_k^\mu \sim d^{\mu_k,y_k}$.

Again, as done in the proof of Theorem 4.1, we recognize that the policies updates performed by Algorithm 2 are instances of online mirror descent ( see Lemma E.1 ). Therefore, we can bound the regret term as follows

$$\sum_{k=1}^K\langle\nabla_\mu\ell_k(\mu_k), \mu_k - \mu_E\rangle \le \sqrt{\frac{K|\mathcal{S}|\log|\mathcal{A}_{\max}|}{2}} + \sqrt{2K|\mathcal{S}|\log(1/\delta)}.$$

All in all, we obtain via a union bound that with probability at least $1 - 4\delta$

$$\left(\frac{1}{K}\sum_{k=1}^K\left\langle d_0, V^{\mu_E,\nu_k^\star} - V^{\mu_k,\nu_k^\star}\right\rangle\right)^2$$

$$\le \frac{|\mathcal{A}_{\max}|}{(1-\gamma)^2}\left(\sqrt{\frac{(2|\mathcal{S}|+1)\log(1/\delta)}{K}} + \sqrt{\frac{|\mathcal{S}|\log|\mathcal{A}_{\max}|}{2K}} + \varepsilon_{\mathrm{opt}}\right).$$

Moreover, analogous calculations give

$$\left(\frac{1}{K}\sum_{k=1}^K\left\langle d_0, V^{\mu_k^\star,\nu_k} - V^{\mu_k^\star,\nu_E}\right\rangle\right)^2$$

$$\le \frac{|\mathcal{A}_{\max}|}{(1-\gamma)^2}\left(\sqrt{\frac{(2|\mathcal{S}|+1)\log(1/\delta)}{K}} + \sqrt{\frac{|\mathcal{S}|\log|\mathcal{A}_{\max}|}{2K}} + \varepsilon_{\mathrm{opt}}\right).$$

Then, using the same decomposition presented in (5), using a union bound and taking square root on both sides, we obtain via another union bound that with probability at least $1 - 5\delta$

$$\frac{1}{K}\sum_{k=1}^K\max_{\mu\in\Pi}\langle d_0, V^{\mu,\nu_k}\rangle - \min_{\nu\in\Pi}\langle d_0, V^{\mu_k,\nu}\rangle$$

$$\le \sqrt{\frac{4|\mathcal{A}_{\max}|}{(1-\gamma)^2}\left(\sqrt{\frac{(2|\mathcal{S}|+1)\log(1/\delta)}{K}} + \sqrt{\frac{|\mathcal{S}|\log|\mathcal{A}_{\max}|}{2K}} + \varepsilon_{\mathrm{opt}}\right)}.$$

At this point, setting $K = \mathcal{O}\left(\frac{|\mathcal{S}||\mathcal{A}_{\max}|^2 \log|\mathcal{A}_{\max}| \log(1/\delta)}{(1-\gamma)^4 \varepsilon^4}\right)$ ensures that with probability at least $1 - 5\delta$

$$\frac{1}{K}\sum_{k=1}^{K}\max_{\mu\in\Pi}\langle d_0, V^{\mu,\nu_k}\rangle - \min_{\nu\in\Pi}\langle d_0, V^{\mu_k,\nu}\rangle \leq \mathcal{O}(\varepsilon) + 2\sqrt{|\mathcal{A}_{\max}|(1-\gamma)^{-2}\varepsilon_{\mathrm{opt}}}.$$

Finally, setting $\varepsilon_{\mathrm{opt}} = |\mathcal{A}_{\max}|^{-1}(1-\gamma)^2\varepsilon^2$ that is $T = \mathcal{O}\left(\frac{|\mathcal{S}|^3|\mathcal{A}_{\max}|^3 \log(1/\delta)}{(1-\gamma)^8 \varepsilon^4}\right)$ ensures that with probability $1 - 5\delta$, we have that $\frac{1}{K}\sum_{k=1}^{K}\max_{\mu\in\Pi}\langle d_0, V^{\mu,\nu_k}\rangle - \min_{\nu\in\Pi}\langle d_0, V^{\mu_k,\nu}\rangle \leq \mathcal{O}(\varepsilon)$. Therefore, the total number of expert queries in $\mathcal{O}(K \cdot T) = \mathcal{O}\left(\frac{|\mathcal{S}||\mathcal{A}_{\max}|^2 \log|\mathcal{A}_{\max}| \log(1/\delta)}{(1-\gamma)^4 \varepsilon^4} \cdot \frac{|\mathcal{S}|^3|\mathcal{A}_{\max}|^3 \log(1/\delta)}{(1-\gamma)^8 \varepsilon^4}\right)$. $\qquad\square$

## G  Analysis for the RL inner loop

For the RL inner loop we analyze UCBVI for stochastic rewards in the discounted setting with a random reward. In particular we have that each time a state-action pair is visited we observe a stochastic reward which is unbiased and with almost surely bounded noise. Compared to the standard analysis in Azar et al. [2017], we handle the discounted infinite horizon setting. In principle, MURMAIL can be used replacing UCBVI with other RL algorithms in the inner loop.

---

**Algorithm 3: UCBVI**

---

**Input:** iteration budget $T$, transition dynamics $P$, unbiased reward function sampler $\mathcal{R}$
**Initialize** $Q_1(s,a) = (1-\gamma)^{-1}$ and $V_1(s) = (1-\gamma)^{-1}$ for all $s, a \in \mathcal{S}\times\mathcal{A}$.
**for** $t = 1$ *to* $T$ **do**
$\quad \pi_t(s) = \mathrm{argmax}_{a\in\mathcal{A}} Q_t(s,a)$
$\quad$ Sample $S_t, A_t \sim d^{\pi_t}$, $S_t' \sim P(\cdot|S_t, A_t)$.
$\quad$ Generate stochastic reward function $R_t \sim \mathcal{R}(S_t)$.
$\quad$ Update counts $N_t(s,a) = N_t(s,a) + \mathbb{1}_{\{S_t, A_t = s, a\}}$,
$\quad\ N_t(s,a,s') = N_t(s,a) + \mathbb{1}_{\{S_t, A_t, S_t' = s, a, s'\}}$.
$\quad$ Estimate transitions and reward

$$\widehat{P}_t(s'|s,a) = \frac{N_t(s,a,s')}{N_t(s,a)+1} \quad \widehat{r}_t(s,a) = \frac{\sum_{t=1}^{T} R_t \mathbb{1}_{\{S_t, A_t = s, a\}}}{N_t(s,a)+1}$$

$\quad$ Set bonuses

$$b_t(s,a) = \frac{4|\mathcal{S}|}{1-\gamma}\sqrt{\frac{\log(2T(T+1)|\mathcal{S}|/\delta)}{N_t(s,a)+1}}$$

$\quad$ Update state action value functions

$$Q_{t+1} = \left[\widehat{r}_t + \gamma\widehat{P}_t V_t + b_t\right]_0^{Q_t}$$

$$V_{t+1}(s) = \max_{a\in\mathcal{A}} Q_{t+1}(s,a)$$

**end**
**return** $\pi_{\mathrm{out}}$ *such that* $d^{\pi_{\mathrm{out}}} = T^{-1}\sum_{t=1}^{T} d^{\pi_t}$

---

The first step of our analysis is to invoke a standard extended performance difference lemma in the infinite horizon setting.

We first introduce some Lemmas which will be useful in the rest of the analysis

**Lemma G.1.** *Consider the MDP* $M = (\mathcal{S}, \mathcal{A}, \gamma, P, r, d_0)$ *and two policies* $\pi, \pi' : \mathcal{S} \to \Delta_{\mathcal{A}}$. *Then consider for any* $\widehat{Q} \in \mathbb{R}^{|\mathcal{S}||\mathcal{A}|}$ *and* $\widehat{V}^{\pi}(s) = \left\langle\pi(\cdot|s), \widehat{Q}(s,\cdot)\right\rangle$ *and* $Q^{\pi'}, V^{\pi'}$ *be respectively the state-action and state value function of the policy* $\pi$ *in MDP* $M$. *Then, it holds that*

$(1-\gamma)\left\langle d_0, \widehat{V}^\pi - V^{\pi'}\right\rangle$ *equals*

$$\left\langle d^{\pi'}, \widehat{Q} - r - \gamma P\widehat{V}^\pi\right\rangle + \mathbb{E}_{s\sim d^{\pi'}}\left[\left\langle \widehat{Q}(s,\cdot,), \pi(\cdot|s) - \pi'(\cdot|s)\right\rangle\right].$$

*Proof.* A proof can be found in Viel et al. [2025]. $\square$

We assume for the moment to have valid bonuses, that is functions $b_k$ such that they guarantees for all $t \in [T]$ and for all $s, a \in \mathcal{S} \times \mathcal{A}$

$$\left|\gamma\widehat{P}_t V_t(s,a) - \gamma P V_t(s,a) + \widehat{r}_t(s,a) - r(s,a)\right| \le b_t(s,a)$$

and we prove that under the above conditions pointwise optimism hold. This point is made precise in the next Lemma.

**Lemma G.2.** *Given a sequence $b_t : \mathcal{S} \times \mathcal{A} \to \mathbb{R}$ such that*

$$\left|\gamma\widehat{P}_t V_t(s,a) - \gamma P V_t(s,a) + \widehat{r}_t(s,a) - r(s,a)\right| \le b_t(s,a)$$

*for all $t \in [T]$, and $s, a \in \mathcal{S} \times \mathcal{A}$ it holds that*

$$V_t \ge V^{\pi^\star} \quad Q_t \ge Q^{\pi^\star} \quad \forall \quad t \in [T]$$

*Proof.* First, let us proof the base case. This is easy since $Q_1(s,a) = \frac{1}{1-\gamma}$ and $V_1(s) = \frac{1}{1-\gamma}$ for all $s, a \in \mathcal{S} \times \mathcal{A}$ and it holds that $Q^{\pi^\star}(s,a) \le \frac{1}{1-\gamma}$ and $V^{\pi^\star}(s) \le \frac{1}{1-\gamma}$ for all $s, a \in \mathcal{S} \times \mathcal{A}$. For the inductive step, let us set as inductive hypothesis that $Q_t - Q^{\pi^\star} \ge 0$ and $V_t - V^{\pi^\star} \ge 0$. Then, recall the update for $Q_{t+1}$,

$$Q_{t+1} = \left[\widehat{r}_t + \gamma\widehat{P}_t V_t + b_t\right]_0^{Q_t}$$

In case the upper truncation is triggered in a generic state action pair $s, a$, we have that $Q_{t+1}(s,a) - Q^{\pi^\star}(s,a) = Q_t(s,a) - Q^{\pi^\star}(s,a) \ge 0$ by the inductive hypothesis. For the state action pairs, where the upper transitions is not triggered we have that

$$\begin{aligned}
Q_{t+1}(s,a) - Q^{\pi^\star}(s,a) &\ge \widehat{r}_t(s,a) + \gamma\widehat{P}_t V_t(s,a) + b_t(s,a) - Q^{\pi^\star}(s,a) \\
&= \widehat{r}_t(s,a) + \gamma\widehat{P}_t V_t(s,a) + b_t(s,a) - r(s,a) - \gamma P V^{\pi^\star}(s,a) \\
&= \widehat{r}_t(s,a) + \gamma\widehat{P}_t V_t(s,a) - \gamma P V_t(s,a) + b_t(s,a) - r(s,a) - \gamma P V^{\pi^\star}(s,a) + \gamma P V_t(s,a) \\
&\ge \gamma P V_t(s,a) - \gamma P V^{\pi^\star}(s,a) \\
&\ge 0.
\end{aligned}$$

Notice that the second last step follows from the validity of the bonuses and the last one follows from the monotonicity of the operator $P$ and by the inductive hypothesis $V_t - V^{\pi^\star} \ge 0$.

At this point we have proven that $Q_{t+1} - Q^\star \ge 0$. For proving the optimism of the estimated state value functions we proceed as follows. Let $a^\star = \text{argmax}_{a\in\mathcal{A}} Q^{\pi^\star}(s,a)$,

$$\begin{aligned}
V_{t+1}(s) - V^{\pi^\star}(s) &= \max_{a\in\mathcal{A}} Q_{t+1}(s,a) - \max_{a\in\mathcal{A}} Q^{\pi^\star}(s,a) \\
&= \max_{a\in\mathcal{A}} Q_{t+1}(s,a) - Q^{\pi^\star}(s,a^\star) \\
&\ge Q_{t+1}(s,a^\star) - Q^{\pi^\star}(s,a^\star) \ge 0.
\end{aligned}$$

$\square$

The next lemma bounds the regret of `UCBVI` (Algorithm 3) with a sequence of valid bonuses with the sum of expected on policy bonuses.

**Lemma G.3.** *Let us consider* UCBVI *run for $T$ iteration with a sequence of valid bonuses $\{b_t\}_{t=1}^T$, then it holds that*

$$\frac{1}{T}\sum_{t=1}^T \left\langle d_0, V^{\pi^\star} - V^{\pi_t}\right\rangle \leq \frac{2}{T(1-\gamma)}\sum_{t=1}^T \langle d^{\pi_t}, b_t\rangle + \frac{|\mathcal{S}||\mathcal{A}|}{(1-\gamma)^2 T}.$$

*Proof.* Using the point wise optimism in Lemma G.2 and the decomposition in Lemma G.1 we have the following decomposition on the regret of UCBVI

$$\frac{1-\gamma}{T}\sum_{t=1}^T \left\langle d_0, V^{\pi^\star} - V^{\pi_t}\right\rangle$$

$$\leq \frac{1-\gamma}{T}\sum_{t=1}^T \langle d_0, V_t - V^{\pi_t}\rangle$$

$$= \frac{1}{T}\sum_{t=1}^T \langle d^{\pi_t}, Q_{t+1} - r + \gamma P V_t\rangle + \frac{1}{T}\sum_{t=1}^T \langle d^{\pi_t}, Q_t - Q_{t+1}\rangle$$

$$\leq \frac{1}{T}\sum_{t=1}^T \left\langle d^{\pi_t}, \widehat{r}_t + \gamma \widehat{P}_t V_t + b_t - r + \gamma P V_t\right\rangle + \frac{1}{T}\sum_{t=1}^T \langle d^{\pi_t}, Q_t - Q_{t+1}\rangle$$

$$\leq \frac{2}{T}\sum_{t=1}^T \langle d^{\pi_t}, b_t\rangle + \frac{1}{T}\sum_{t=1}^T \langle d^{\pi_t}, Q_t - Q_{t+1}\rangle$$

where last inequality holds thanks to the validity of the bonuses. For the second term, we can get the following bound which crucially use in the first inequality the fact that the sequence $\{Q_t\}_{t=1}^T$ is decreasing.

$$\frac{1}{T}\sum_{t=1}^T \langle d^{\pi_t}, Q_t - Q_{t+1}\rangle \leq \frac{1}{T}\sum_{t=1}^T \sum_{s,a} Q_t(s,a) - Q_{t+1}(s,a)$$

$$= \frac{1}{T}\sum_{s,a}\sum_{t=1}^T Q_t(s,a) - Q_{t+1}(s,a)$$

$$= \frac{1}{T}\sum_{s,a} Q_1(s,a)$$

$$= \frac{|\mathcal{S}||\mathcal{A}|}{(1-\gamma)T}.$$

$\square$

## G.1 Showing validity of the bonuses

We show in this section how to design a valid sequence of bonuses.

**Lemma G.4.** *Let us consider run* UCBVI *for $T$ for a stochastic reward almost surely bounded by 2, i.e. $R_{\max} \leq 2$ iterations and consider the following transition and reward estimators*

$$\widehat{P}_t(s'|s,a) = \frac{N_t(s,a,s')}{N_t(s,a)+1} \quad \widehat{r}_t(s,a) = \frac{\sum_{t=1}^T R_t \mathbb{1}_{\{S_t, A_t = s,a\}}}{N_t(s,a)+1}$$

*then the bonus sequence defined as*

$$b_t(s,a) = \frac{4|\mathcal{S}|}{1-\gamma}\sqrt{\frac{\log(2T(T+1)|\mathcal{S}|/\delta)}{N_t(s,a)+1}}$$

*satisfies*

$$\mathbb{P}\left[\left|\widehat{r}_t(s,a) + \gamma\widehat{P}_t V_t(s,a) - r(s,a) - \gamma P V_t(s,a)\right| \leq b_t(s,a) \quad \forall t \in [T]\right] \geq 1 - 2\delta.$$

*Proof.* For all $t \in [T]$ simultaneously, we have the following high probability upper bound

$$\widehat{P}_t(s'|s,a) - P(s'|s,a)$$

$$= \frac{\sum_{\tau=1}^{T} \mathbb{1}_{\{S_\tau, A_\tau = s, a\}} \mathbb{1}_{\{S'_\tau = s'\}}}{N_t(s,a) + 1} - \frac{N_t(s,a) + 1}{N_t(s,a)} \frac{\sum_{\tau=1}^{T} \mathbb{1}_{\{S_\tau, A_\tau = s, a\}} P(s'|s,a)}{N_t(s,a) + 1}$$

$$= \frac{\sum_{\tau=1}^{T} \mathbb{1}_{\{S_\tau, A_\tau = s, a\}} \left( \mathbb{1}_{\{S'_\tau = s'\}} - P(s'|s,a) \right)}{N_t(s,a) + 1} - \frac{\sum_{\tau=1}^{T} \mathbb{1}_{\{S_\tau, A_\tau = s, a\}} P(s'|s,a)}{N_t(s,a)(N_t(s,a) + 1)}$$

$$= \frac{\sum_{\tau=1:S_\tau, A_\tau = s, a}^{T} \left( \mathbb{1}_{\{S'_\tau = s'\}} - P(s'|s,a) \right)}{N_t(s,a) + 1} - \frac{\sum_{\tau=1}^{T} \mathbb{1}_{\{S_\tau, A_\tau = s, a\}} P(s'|s,a)}{N_t(s,a)(N_t(s,a) + 1)}$$

$$\leq \frac{\sqrt{N_t(s,a) \log(N_t(s,a)(N_t(s,a) + 1)/\delta)}}{N_t(s,a) + 1} - \frac{\sum_{\tau=1}^{T} \mathbb{1}_{\{S_\tau, A_\tau = s, a\}} P(s'|s,a)}{N_t(s,a)(N_t(s,a) + 1)}$$

$$\leq \sqrt{\frac{\log(T(T+1)/\delta)}{N_t(s,a) + 1}} - \frac{\sum_{\tau=1}^{T} \mathbb{1}_{\{S_\tau, A_\tau = s, a\}} P(s'|s,a)}{N_t(s,a)(N_t(s,a) + 1)}$$

where the last inequality follows with probability $1 - \delta$ from an application of the Azuma Hoeffding inequality making special care of the fact that the total number of visits $N_t(s,a)$ is not an independent random variable with respect to the random variables of which we are computing the mean, that is $\left\{ \mathbb{1}_{\{S_\tau, A_\tau = s, a\}} \right\}_{t=1}^{T}$. For this reason we pay the factor $\log(N_t(s,a)(N_t(s,a) + 1))$ in the upper bound. We refer the reader to [Lattimore and Szepesvári, 2020, Exercise 7.1 ] for details. Therefore, by triangular inequality and a union bound over the state space.

$$\left\| \widehat{P}_t(\cdot|s,a) - P(\cdot|s,a) \right\|_\infty \leq \sqrt{\frac{\log(2T(T+1)|\mathcal{S}|/\delta)}{N_t(s,a) + 1}} + \frac{\sum_{\tau=1}^{T} \mathbb{1}_{\{S_\tau, A_\tau = s, a\}} P(s'|s,a)}{N_t(s,a)(N_t(s,a) + 1)}$$

$$\leq \sqrt{\frac{\log(2T(T+1)|\mathcal{S}|/\delta)}{N_t(s,a) + 1}} + \frac{1}{(N_t(s,a) + 1)}$$

For the reward concentration we have that

$$|r(s,a) - \widehat{r}_k(s,a)| = \left| \frac{\sum_{t=1}^{T} \mathbb{1}_{\{S_t, A_t = s, a\}}(R_t - r(s,a))}{N_t(s,a) + 1} - \frac{\sum_{t=1}^{T} \mathbb{1}_{\{S_t, A_t = s, a\}} r(s,a)}{N_t(s,a)(N_t(s,a) + 1)} \right|$$

$$\leq \left| \frac{\sum_{t=1}^{T} \mathbb{1}_{\{S_t, A_t = s, a\}}(R_t - r(s,a))}{N_t(s,a) + 1} \right| + \left| \frac{\sum_{t=1}^{T} \mathbb{1}_{\{S_t, A_t = s, a\}} r(s,a)}{N_t(s,a)(N_t(s,a) + 1)} \right|$$

$$\leq \sqrt{\frac{R_{\max} \log(2T(T+1)/\delta)}{N_t(s,a) + 1}} + \frac{R_{\max}}{(N_t(s,a) + 1)}$$

where the last inequality holds with probability $1 - \delta$ thanks to the double sided Azuma-Hoeffding inequality.

For the second part of the statement consider that each possible element of the sequence $\{V_t\}_{t=1}^{T}$ generated by UCBVI satisfies $\|V_t\|_1 \leq \frac{|\mathcal{S}|}{1-\gamma}$. Therefore, it holds that

$$\left| \widehat{P}_t V_t(s,a) - P V_t(s,a) \right| \leq \left\| \widehat{P}_t(\cdot|s,a) - P(\cdot|s,a) \right\|_\infty \|V_t\|_1$$

$$\leq \frac{|\mathcal{S}|}{1-\gamma} \left\| \widehat{P}_t(\cdot|s,a) - P(\cdot|s,a) \right\|_\infty$$

$$\leq \frac{|\mathcal{S}|}{1-\gamma} \left( \sqrt{\frac{\log(2T(T+1)|\mathcal{S}|/\delta)}{N_t(s,a) + 1}} + \frac{1}{(N_t(s,a) + 1)} \right)$$

where the last inequality holds with probability $1 - \delta$. Therefore, it follows that for all $t \in [T]$ simultaneously, with probability at least $1 - 2\delta$

$$\left| \widehat{r}_t(s,a) + \gamma \widehat{P}_t V_t(s,a) - r(s,a) - \gamma P V_t(s,a) \right| \leq \frac{R_{\max} + |\mathcal{S}|}{1 - \gamma}$$
$$\cdot \left( \sqrt{\frac{\log(2T(T+1)\,|\mathcal{S}|\,/\delta)}{N_t(s,a) + 1}} + \frac{1}{(N_t(s,a) + 1)} \right)$$
$$\leq b_t(s,a),$$

where the final upper bound by the bonus uses that $|\mathcal{S}| \geq 2$ and $\sqrt{\frac{\log(2T(T+1)|\mathcal{S}|/\delta)}{N_t(s,a)+1}} \geq \frac{1}{(N_t(s,a)+1)}$. $\qquad \square$

## G.2 Bound the bonus sum

**Lemma G.5.** *The expected on policy bonus sum is bounded as follows with probability $1 - \delta$*

$$\sum_{t=1}^{T} \langle d^{\pi_t}, b_t \rangle \leq \frac{4\,|\mathcal{S}|\,\sqrt{\log(2T(T+1)\,|\mathcal{S}|\,/\delta)}}{1 - \gamma} \sqrt{2\,|\mathcal{S}|\,|\mathcal{A}|\,T \log(T)}$$
$$+ \frac{16\,|\mathcal{S}|}{1 - \gamma} \sqrt{\log(2T(T+1)\,|\mathcal{S}|\,/\delta)} \log\left(\frac{2T}{\delta}\right)$$
$$\leq \widetilde{\mathcal{O}}\left( \frac{\sqrt{|\mathcal{S}|^3\,|\mathcal{A}|\,T \log(1/\delta)}}{1 - \gamma} \right).$$

*Proof.* We apply [Rosenberg et al., 2020, Lemma D.4] to conclude that with probability at least $1 - \delta$

$$\sum_{t=1}^{T} \langle d^{\pi_t}, b_t \rangle = 2 \sum_{t=1}^{T} b_t(S_t, A_t) + \frac{16\,|\mathcal{S}|}{1 - \gamma} \sqrt{\log(2T(T+1)\,|\mathcal{S}|\,/\delta)} \log\left(\frac{2T}{\delta}\right)$$

Then, we have that

$$\sum_{t=1}^{T} b_t(S_t, A_t) = \sum_{t=1}^{T} \frac{4\,|\mathcal{S}|}{1 - \gamma} \sqrt{\frac{\log(2T(T+1)\,|\mathcal{S}|\,/\delta)}{N_t(S_t, A_t) + 1}}$$
$$\leq \frac{4\,|\mathcal{S}|\,\sqrt{\log(2T(T+1)\,|\mathcal{S}|\,/\delta)}}{1 - \gamma} \sqrt{T \sum_{t=1}^{T} \frac{1}{N_t(S_t, A_t) + 1}}$$
$$\leq \frac{4\,|\mathcal{S}|\,\sqrt{\log(2T(T+1)\,|\mathcal{S}|\,/\delta)}}{1 - \gamma} \sqrt{T \sum_{t=1}^{T} \frac{1}{N_t(S_t, A_t) + 1}}$$

Finally, it holds that

$$\sum_{t=1}^{T} \frac{1}{N_t(S_t, A_t) + 1} = \sum_{s,a} \sum_{t=1}^{T} \frac{\mathbb{1}_{\{S_t, A_t = s, a\}}}{1 + \sum_{\tau=1}^{t} \mathbb{1}_{\{S_\tau, A_\tau = s, a\}}}$$
$$\leq |\mathcal{S}|\,|\mathcal{A}| \log(T)$$

where the last inequality follows applying [Orabona, 2023, Lemma 4.13] for $f(x) = x^{-1}$. Putting everything together concludes the proof. $\qquad \square$

### G.3 Final `UCBVI` bound

**Lemma G.6.** *Let us consider `UCBVI` (Algorithm 3) in an environment with a stochastic unbiased reward almost surely bounded by 2 run for $T$ iteration with a sequence of valid bonuses $\{b_t\}_{t=1}^T$ specified in the statement of Lemma G.4, then it holds that with probability at least $1 - 3\delta$*

$$\frac{1}{T}\sum_{t=1}^T \left\langle d_0, V^{\pi^\star} - V^{\pi_t}\right\rangle \leq \widetilde{\mathcal{O}}\left(\sqrt{\frac{|\mathcal{S}|^3 |\mathcal{A}| \log(1/\delta)}{(1-\gamma)^4 T}}\right) + \frac{|\mathcal{S}| |\mathcal{A}|}{(1-\gamma)^2 T}.$$

*Therefore, for the mixture policy $\pi_{\text{out}}$ such that $\frac{1}{T}\sum_{t=1}^T d^{\pi_t} = d^{\pi_{\text{out}}}$ it holds that with probability $1 - 3\delta$*

$$\left\langle d_0, V^{\pi^\star} - V^{\pi_{\text{out}}}\right\rangle \leq \varepsilon_{\text{opt}}$$

*for $T = \widetilde{\mathcal{O}}\left(\frac{|\mathcal{S}|^3 |\mathcal{A}| \log(1/\delta)}{(1-\gamma)^4 \varepsilon_{\text{opt}}^2}\right)$.*

*Proof.* The proof follows trivially from the combination of Lemma G.5 and Lemma G.3 and a union bound over the event that the bonus are valid and the event under which the bound in Lemma G.5 holds . $\square$

### G.4 Properties of the reward estimate

**Lemma G.7.** *For any policy $\pi \in \Pi$ and expert policy $\pi_E \in \Pi$ consider a particular state $s \in \mathcal{S}$ and sampling $A_E \sim \pi_E(\cdot|s)$, $A'_E \sim \pi_E(\cdot|s)$ . Then, the following facts hold true*

$$\mathbb{E}\left[\mathbb{1}\left\{A_E = A'_E\right\} - 2\pi(A_E|s) + \|\pi(\cdot|s)\|^2\right] = \|\pi_E(\cdot|s) - \pi(\cdot|s)\|^2$$

*and*

$$\mathbb{1}\left\{A_E = A'_E\right\} - 2\pi(A_E|s) + \|\pi(\cdot|s)\|^2 \leq 2 \quad \text{almost surely}$$

*Proof.* First note that

$$\|\pi_E - \pi\|^2 = \|\pi_E(\cdot|s)\|^2 - 2\left\langle\pi_E(\cdot|s), x^k(\cdot|s)\right\rangle + \|\pi(\cdot|s)\|^2$$
$$= \sum_a \pi_E(a \mid s)^2 + \sum_a \pi(a \mid s)^2 - 2\sum_a \pi_E(a \mid s)\pi(a \mid s)$$
$$= \sum_a \pi_E(a \mid s)^2 + \sum_a \pi(a \mid s)^2 - 2\mathbb{E}_{A\sim\pi_E(\cdot|s)}[\pi(a \mid s)].$$

Now, note that for a given $a \in \mathcal{A}$, we get that

$$\pi_E^2(a \mid s) = \mathbb{P}(A_E = a)^2 = \mathbb{P}(A_E = a)\mathbb{P}(A'_E = a) = \mathbb{P}(A_E = A'_E) = \mathbb{E}[\mathbb{1}_{\{A_E = A'_E\}}],$$

where $A_E, A'_E$ are independent samples from $\pi_E(\cdot \mid s)$. Therefore, we can conclude that

$$\mathbb{1}\left\{A_E = A'_E\right\} - 2\pi(A_E|s) + \|\pi(\cdot|s)\|^2.$$

is an unbiased estimator of $\|\pi_E - \pi\|^2$. The second statement is easy to show

$$\mathbb{1}\left\{A_E = A'_E\right\} - 2\pi(A_E|s) + \|\pi(\cdot|s)\|^2 \leq \mathbb{1}\left\{A_E = A'_E\right\} + \|\pi(\cdot|s)\|^2 \leq 2.$$

$\square$

## H  Extension to $n$-player general-sum games

In this section, we sketch the analysis for the $n$-player general sum extension. The goal of this section is to show that the algorithm design is decentralized, meaning that it can avoid **the curse of multi-agents** in $n$-player general-sum Games as stated in Remark 4.1. The idea is that the introduced algorithms keep the other players fixed, in the RL inner-loop and the BR oracle calls respectively. This results in a decentralized execution. This section starts with the introduction of $n$-player general-sum

Games and all the necessary notations. Then, we show how the objective varies slightly from the one in Zero-Sum Games. Last, we give the proof sketch for the $n$-player general-sum case.

First note that, an infinite-horizon general-sum Markov game is defined by $\mathcal{G} = (n, \mathcal{S}, \mathcal{A}, P, r, \gamma, d_0)$, where $n$ is the number of players, $\mathcal{S}$ is the finite (joint-)state space, $\mathcal{A} := \mathcal{A}_1 \times \ldots \times \mathcal{A}_n$ is the finite (joint-)action space, where $\mathcal{A}_i$ is the action space of player $i \in \{1, \ldots, n\}$, $P \in \mathbb{R}^{|\mathcal{S}||\mathcal{A}| \times |\mathcal{A}|}$ is the (unknown) transition function, $r \in \mathbb{R}^{|\mathcal{S}||\mathcal{A}|}$ the reward vector, a discount factor $\gamma \in [0, 1)$ and $d_0$ a distribution over the state space from which the starting distribution is sampled. In general-sum games there is no additional restriction on the reward function. A policy of a player $i$ is defined as $\pi_i : \mathcal{S} \to \Delta_{\mathcal{A}_i}$ and we denote the joint policy as $\boldsymbol{\pi} = (\pi_1, \ldots, \pi_n) = (\pi_i, \pi_{-i})$, where $\pi_{-i}$ denotes the policy of all players except player $i$. We also use $\pi_{-(i,j)}$ to denote all players but players $i, j$. Additionally, we denote a joint action as $\mathbf{a} = (a_1, \ldots, a_n)$. The value function and state-action value function for any player $i$ for a given state $s \in \mathcal{S}$, and any state-action pair $(s, \mathbf{a}) \in \mathcal{S} \times \mathcal{A}$ is given by

$$V_i^{\boldsymbol{\pi}}(s) := \mathbb{E}_{\boldsymbol{\pi}}\left[\sum_{t=0}^{\infty} r_i(S, \boldsymbol{A}) \mid S_0 = s\right]$$

$$Q_i^{\boldsymbol{\pi}}(s, \mathbf{a}) := \mathbb{E}_{\boldsymbol{\pi}}\left[\sum_{t=0}^{\infty} r_i(S, \boldsymbol{A}) \mid S_0 = s, A_0 = \mathbf{a}\right].$$

All other expressions are defined as in Section 2 with the extension that the joint actions are now given by $\mathbf{a}$ and one fixes the policies of all players expect player $i$, i.e. $\pi_{-i}$, for the induced Games.

The important difference for our analysis is in the change of the objective. In the two player Zero-sum case the objective of the the Nash Gap (1) is already a simplified form. In general, the Nash Gap is defined as the sum of exploitabilities of each player, i.e.

$$\text{Nash-Gap}(\boldsymbol{\pi}) := \sum_{i=1}^{n} \max_{\pi_i'} V_i^{\pi_i', \pi_{-i}}(s_0) - V_i^{\pi_i, \pi_{-i}}(s_0). \tag{13}$$

One can easily see the structure of the 2 player zero-sum Game leads to

$$\text{Nash-Gap}(\boldsymbol{\pi}) := \sum_{i=1}^{2} \max_{\pi_i'} V_i^{\pi_i', \pi_{-i}}(s_0) - V_i^{\pi_i, \pi_{-i}}(s_0)$$

$$= \max_{\pi_1'} V_1^{\pi_1', \pi_2}(s_0) - V_1^{\pi_1, \pi_2}(s_0) + \max_{\pi_2'} V_2^{\pi_1, \pi_2'}(s_0) - V_2^{\pi_1, \pi_2}(s_0)$$

$$= \max_{\pi_1'} V_1^{\pi_1', \pi_2}(s_0) - V_1^{\pi_1, \pi_2}(s_0) - \min_{\pi_2'} V_1^{\pi_2', \pi_1}(s_0) + V_1^{\pi_1, \pi_2}(s_0)$$

$$= \max_{\pi_1'} V_1^{\pi_1', \pi_2}(s_0) - \min_{\pi_2'} V_1^{\pi_1, \pi_2'}(s_0),$$

where in the third equality, we used the assumption on the reward for two player zero-sum games, i.e. $r^1(s, a, b) = -r^2(s, a, b)$. Noting that $\boldsymbol{\pi} = (\pi_1, \pi_2) = (\mu, \nu)$ this is exactly the definition of (1).

In the following, we will show the implication of the change of the objective on the Multi-agent Imitation Learning setting. We start again by rewriting the objective with the expert policies

$$\text{Nash-Gap}(\boldsymbol{\pi}) = \sum_{i=1}^{n} \max_{\pi_i'} V_i^{\pi_i', \pi_{-i}}(s_0) - V_i^{\pi_i, \pi_{-i}}(s_0)$$

$$= \sum_{i=1}^{n} V_i^{\pi_i^\star, \pi_{-i}}(s_0) - V_i^{\pi_{E_i}, \pi_{E_{-i}}}(s_0) + V_i^{\pi_{E_i}, \pi_{E_{-i}}}(s_0) - V_i^{\pi_i, \pi_{-i}}(s_0)$$

$$\leq \sum_{i=1}^{n} \left( \underbrace{V_i^{\pi_i^\star, \pi_{-i}}(s_0) - V_i^{\pi_i^\star, \pi_{E_{-i}}}(s_0)}_{\text{Exploit}-\text{Gap}_i} + \underbrace{V_i^{\pi_{E_i}, \pi_{E_{-i}}}(s_0) - V_i^{\pi_i, \pi_{-i}}(s_0)}_{\text{Value}-\text{Gap}} \right).$$

The *Exploit-Gap* is similar to the objective analyzed in the zero-sum case with the difference that now that the policies of the other players are varying in $n - 1$ cases. The *Value-Gap* is new and does not appear in the zero-sum case as one can again use the structure of the reward in that case. However,

the latter is easy to analyze as it can be seen as a single-agent MDP with the joint policy $\boldsymbol{\pi}$ and the joint expert $\boldsymbol{\pi}_E$ respectively.

We can now compute the analysis for Behavior Cloning and start by bounding $\mathrm{Exploit} - \mathrm{Gap}$

$$
\begin{aligned}
\mathrm{Exploit} - \mathrm{Gap}_i &\le \frac{2}{1-\gamma} \max_{\pi_i \in \mathrm{br}(\pi_{-i})} \mathbb{E}_{\pi_i, \pi_{E-i}} \left[ \sum_{t=0}^{\infty} \gamma^t \mathrm{TV}\left(\pi_{E_{-i}}(\cdot \mid s), \widehat{\pi}_{-i}(\cdot \mid s)\right) \right] \\
&\le \frac{2}{1-\gamma} \max_{\pi_i \in \mathrm{br}(\pi_{-i})} \mathbb{E}_{\pi_i, \pi_{E-i}} \left[ \sum_{t=0}^{\infty} \gamma^t \mathrm{TV}\left(\pi_E(\cdot \mid s), \widehat{\pi}(\cdot \mid s)\right) \right] \\
&\le \frac{2}{1-\gamma} \max_{\pi_i \in \mathrm{br}(\pi_{-i})} \left\| \frac{d^{\pi_i, \pi_{E-i}}}{d^{\pi_{E_i}, \pi_{E-i}}} \right\|_{\infty} \mathbb{E}_{\boldsymbol{\pi}_E} \left[ \sum_{t=0}^{\infty} \gamma^t \mathrm{TV}\left(\pi_E(\cdot \mid s), \widehat{\pi}(\cdot \mid s)\right) \right]
\end{aligned}
$$

Next, we can use that the policies are all conditionally independent in the state as we assumed to have a NE expert. Therefore, it holds true that

$$
\mathrm{TV}\left(\pi_E(\cdot \mid s), \widehat{\pi}(\cdot \mid s)\right) \le \sum_{i=1}^{n} \mathrm{TV}\left(\pi_{E_i}(\cdot \mid s), \widehat{\pi}_i(\cdot \mid s)\right).
$$

Similar arguments can be used to minimize the *Value-Gap* without the change of measure to obtain

$$
\mathrm{Value} - \mathrm{Gap} \le \frac{2}{1-\gamma} \left( \sum_{i=1}^{n} \mathrm{TV}(\pi_{E_i}, \widehat{\pi}_i) \right) \tag{14}
$$

Using this for each player we obtain

$$
\mathrm{Nash\text{-}Gap}(\boldsymbol{\pi}) \le \frac{8n}{(1-\gamma)^2} \max_i \max_{\pi_i \in \mathrm{BR}(\pi_{-i})} \left\| \frac{d^{\pi_i, \pi_{E-i}}}{d^{\pi_{E_i}, \pi_{E-i}}} \right\|_{\infty} \sqrt{\frac{|\mathcal{S}| \left(\sum_i \mathcal{A}_i\right) \log^2\left(n |\mathcal{S}| / \delta\right)}{N}}
$$

Next, we want to sketch the extension of Algorithm 2 for $n$-player general-sum games. We only give the extension for this algorithm as the ideas translate analogously to Algorithm 1. In a first step we

have to adjust the decomposition in (5). We get

$$\left( \frac{1}{K} \sum_{k=1}^{K} \sum_{i=1}^{n} \max_{\pi_i \in \Pi} \left\langle d_0, V_i^{\pi_i, \pi_{-i}^k} \right\rangle - \left\langle d_0, V_i^{\pi_i^k, \pi_{-i}^k} \right\rangle \right)^2$$

$$= \left( \frac{1}{K} \sum_{k=1}^{K} \sum_{i=1}^{n} \left\langle d_0, V_i^{\pi_i^{\star,k}, \pi_{-i}^k} - V_i^{\pi_i^k, \pi_{-i}^k} \right\rangle \right)^2$$

$$= \left( \frac{1}{K} \sum_{k=1}^{K} \sum_{i=1}^{n} \left\langle d_0, V_i^{\pi_i^{\star,k}, \pi_{-i}^k} - V_i^{\pi_{E_i}, \pi_{E_{-i}}} + V_i^{\pi_{E_i}, \pi_{E_{-i}}} - V_i^{\pi_i^k, \pi_{-i}^k} \right\rangle \right)^2$$

$$\leq \left( \frac{1}{K} \sum_{k=1}^{K} \sum_{i=1}^{n} \left\langle d_0, V_i^{\pi_i^{\star,k}, \pi_{-i}^k} - V_i^{\pi_i^{\star,k}, \pi_{E_{-i}}} + V_i^{\pi_{E_i}, \pi_{E_{-i}}} - V_i^{\pi_i^k, \pi_{-i}^k} \right\rangle \right)^2$$

$$\leq 2 \left( \underbrace{\frac{1}{K} \sum_{k=1}^{K} \sum_{i=1}^{n} \left\langle d_0, V_i^{\pi_i^{\star,k}, \pi_{-i}^k} - V_i^{\pi_i^{\star,k}, \pi_{E_{-i}}} \right\rangle}_{(i) \text{Exploit} - \text{Gap}} \right)^2$$

$$+ 2 \left( \underbrace{\frac{1}{K} \sum_{k=1}^{K} \sum_{i=1}^{n} \left\langle d_0, V_i^{\pi_{E_i}, \pi_{E_{-i}}} - V_i^{\pi_i^k, \pi_{-i}^k} \right\rangle}_{(ii) \text{Value} - \text{Gap}} \right)^2$$

$$\leq 2n \left( \left( \frac{1}{K} \sum_{k=1}^{K} \left\langle d_0, V_1^{\text{br}(\pi_{-1}^k), \pi_{-1}^k} - V_1^{\text{br}(\pi_{-1}^k), \pi_{E_{-1}}} \right\rangle \right)^2 \right.$$

$$+ \ldots + \left. \left( \frac{1}{K} \sum_{k=1}^{K} \left\langle d_0, V_n^{\text{br}(\pi_{-n}^k), \pi_{-n}^k} - V_n^{\text{br}(\pi_{-n}^k), \pi_{E_{-n}}} \right\rangle \right)^2 \right)$$

$$+ 2 \left( \frac{1}{K} \sum_{k=1}^{K} \sum_{i=1}^{n} \left\langle d_0, V_i^{\pi_{E_i}, \pi_{E_{-i}}} - V_i^{\pi_i^k, \pi_{-i}^k} \right\rangle \right)^2,$$

where $\pi_i^{\star,k} \in \text{br}(\pi_{-i}^k)$. We will focus on (i) first. Then, we tackle the Value Gap. In particular, we will focus on the composition for any player $i \in \{1, \ldots, n\}$. For $(i)$, we cannot continue directly as done in proof of Theorem 4.2 as now the policies inside differ in $n-1$ other policies and therefore

we cannot directly apply the performance difference lemma. Instead, we first have to do the following

$$(\mathrm{i}) = \left( \frac{1}{K} \sum_{k=1}^{K} \left\langle d_0, V_i^{\pi_i^{\star,k}, \pi_{-i}^k} - V_i^{\pi_i^{\star,k}, \pi_{E-i}} \right\rangle \right)^2$$

$$= \left( \frac{1}{K} \sum_{k=1}^{K} \left\langle d_0, V_i^{\pi_i^{\star,k}, (\pi_1^k, \dots, \pi_{i-1}^k, \pi_{i+1}^k, \dots \pi_n^k)} - V_i^{\pi_i^{\star,k}, (\pi_{E_1}, \dots, \pi_{E_{i-1}}, \pi_{E_{i+1}}, \dots, \pi_{E_n})} \right\rangle \right)^2$$

$$= \left( \frac{1}{K} \sum_{k=1}^{K} \left\langle d_0, V_i^{\pi_i^{\star,k}, (\pi_1^k, \dots, \pi_{i-1}^k, \pi_{i+1}^k, \dots \pi_n^k)} - V_i^{\pi_i^{\star,k}, (\pi_1^k, \dots, \pi_{i-1}^k, \pi_{i+1}^k, \dots, \pi_{n-1}^k, \pi_{En})} \right. \right.$$

$$\left. + V_i^{\pi_i^{\star,k}, (\pi_1^k, \dots, \pi_{i-1}^k, \pi_{i+1}^k, \pi_{n-1}^k, \dots \pi_{En})} - V_i^{\pi_i^{\star,k}, (\pi_1^k, \dots, \pi_{i-1}^k, \pi_{i+1}^k, \dots, \pi_{E_{n-1}}, \pi_{En})} \right.$$

$$\vdots$$

$$\left. \left. + V_i^{\pi_i^{\star,k}, (\pi_1^k, \dots, \pi_{E_{i-1}}, \pi_{E_{i+1}}, \dots, \pi_{En})} - V_i^{\pi_i^{\star,k}, (\pi_{E_1}, \dots, \pi_{E_{i-1}}, \pi_{E_{i+1}}, \dots, \pi_{En})} \right\rangle \right)^2$$

$$\leq (n-1) \left( \left( \frac{1}{K} \sum_{k=1}^{K} \left\langle d_0, V_i^{\pi_i^{\star,k}, (\pi_1^k, \dots, \pi_{i-1}^k, \pi_{i+1}^k, \dots \pi_n^k)} - V_i^{\pi_i^{\star,k}, (\pi_1^k, \dots, \pi_{i-1}^k, \pi_{i+1}^k, \dots, \pi_{n-1}^k, \pi_{En})} \right\rangle \right)^2 \right.$$

$$\vdots$$

$$\left. + \left( \frac{1}{K} \sum_{k=1}^{K} \left\langle d_0, V_i^{\pi_i^{\star,k}, (\pi_1^k, \dots, \pi_{E_{i-1}}, \pi_{E_{i+1}}, \dots, \pi_{En})} - V_i^{\pi_i^{\star,k}, (\pi_{E_1}, \dots, \pi_{E_{i-1}}, \pi_{E_{i+1}}, \dots, \pi_{En})} \right\rangle \right)^2 \right)$$

Note that by the telescopic sum construction, we now have that each difference of value function only differs in one policy and last we applied $(a+b)^2 \leq 2(a^2 + b^2)$. We will now focus on one term for the exploit Gap for any $i$. Therefore, we can proceed similar as in proof Theorem 4.2 and by dividing out $K^2$, applying the performance difference lemma ($(n-1)$ times, for every player but player $i$), Cauchy Schwarz and Jensen we get

$$(n-1) \left( \left( \frac{1}{K} \sum_{k=1}^{K} \left\langle d_0, V_i^{\pi_i^{\star,k}, (\pi_1^k, \dots, \pi_{i-1}^k, \pi_{i+1}^k, \dots \pi_n^k)} - V_i^{\pi_i^{\star,k}, (\pi_1^k, \dots, \pi_{i-1}^k, \pi_{i+1}^k, \dots, \pi_{n-1}^k, \pi_{En})} \right\rangle \right)^2 \right.$$

$$\vdots$$

$$\left. + \left( \frac{1}{K} \sum_{k=1}^{K} \left\langle d_0, V_i^{\pi_i^{\star,k}, (\pi_1^k, \dots, \pi_{E_{i-1}}, \pi_{E_{i+1}}, \dots, \pi_{En})} - V_i^{\pi_i^{\star,k}, (\pi_{E_1}, \dots, \pi_{E_{i-1}}, \pi_{E_{i+1}}, \dots, \pi_{En})} \right\rangle \right)^2 \right)$$

$$\leq \frac{(n-1)|A_{\max}|}{(1-\gamma)^2 K} \left( \sum_{k=1}^{K} \mathbb{E}_{s \sim d^{y_{i,1}^{\star,k}, \pi_{-i}^k}} \left[ \left\| \pi_n^k(\cdot \mid s) - \pi_{E_n}(\cdot \mid s) \right\|^2 \right] + \varepsilon_{\mathrm{opt}}^{i,1} \right.$$

$$\vdots$$

$$\left. + \sum_{k=1}^{K} \mathbb{E}_{s \sim d^{y_{i,n}^{\star,k}, (\pi_1^k, \pi_{E-(i,1)})}} \left[ \left\| \pi_1^k(\cdot \mid s) - \pi_{E_1}(\cdot \mid s) \right\|^2 \right] + \varepsilon_{\mathrm{opt}}^{i,n} \right)$$

$$= \frac{(n-1)|A_{\max}|}{(1-\gamma)^2 K} \sum_{k=1}^{K} \sum_{j \neq i}^{n} \mathbb{E}_{s \sim d^{y_{ij}^{\star,k}, (\pi_{j:n}^k \pi_{\mathrm{E},1:j})-i}} \left[ \left\| \pi_{n-j+1}^k(s) - \pi_{\mathrm{E},n-j+1}(s) \right\|^2 \right] + \varepsilon_{\mathrm{opt}}^{i,j}$$

where we introduced $n-1$ RL inner loops for the player $i$ to approximate the maximum uncertainty response policy denoted by $y_{ij}^{\star,k}$ for $j \in [n]$. More precisely, we have

$$\mathbb{E}_{s \sim d^{\pi_i^{\star,k}, (\pi_{j:n}^k \pi_{E,1:j})_{-i}}} \left[ \left\| \pi_{n-j+1}^k(s) - \pi_{E,n-j+1}(s) \right\|^2 \right]$$
$$\leq \mathbb{E}_{s \sim d^{y_{ij}^{\star,k}, (\pi_{j:n}^k \pi_{E,1:j})_{-i}}} \left[ \left\| \pi_{n-j+1}^k(s) - \pi_{E,n-j+1}(s) \right\|^2 \right] + \varepsilon_{\text{opt}}^{ij}$$

For the Value Gap, we obtain the following:

$$\left( \frac{1}{K} \sum_{k=1}^{K} \sum_{i=1}^{n} \left\langle d_0, V_i^{\pi_{E_i}, \pi_{E-i}} - V_i^{\pi_i^k, \pi_{-i}^k} \right\rangle \right)^2 \tag{15}$$

$$\leq \left( \frac{2n}{1-\gamma} \frac{1}{K} \sum_{k=1}^{K} \mathbb{E}_{s \sim d^{\pi^k}} \left[ \text{TV}\left( \pi_E(\cdot \mid s), \pi^k(\cdot \mid s) \right) \right] \right)^2 \tag{16}$$

$$\leq \frac{4n^2}{(1-\gamma)^2} \frac{1}{K} \sum_{k=1}^{K} \mathbb{E}_{s \sim d^{\pi^k}} \left[ \text{TV}^2\left( \pi_E(\cdot \mid s), \pi^k(\cdot \mid s) \right) \right] \tag{17}$$

$$= \frac{4n^2}{(1-\gamma)^2} \frac{1}{K} \sum_{k=1}^{K} \sum_{i=1}^{n} \mathbb{E}_{s \sim d^{\pi^k}} \left[ \text{TV}^2\left( \pi_{E_i}(\cdot \mid s), \pi_i^k(\cdot \mid s) \right) \right] \tag{18}$$

$$\tag{19}$$

where for the first inequality we have applied the performance difference lemma to the joint policies $\pi^E$ and $\pi^k$ as well as Hölder with $\|\cdot\|_1$ and $\|\cdot\|_\infty$ and bound the value function with its maximum value $\frac{1}{1-\gamma}$. For the second inequality we used Cauchy Schwarz/Jensen's inequality and in the last step that the policies are all conditional independent on $s$. Therefore, the whole objective can be upper bounded by

$$\frac{4n^2}{(1-\gamma)^2} \frac{1}{K} \sum_{k=1}^{K} \sum_{i=1}^{n} \mathbb{E}_{s \sim d^{\pi^k}} \left[ \text{TV}^2\left( \pi_{E_i}(\cdot \mid s), \pi_i^k(\cdot \mid s) \right) \right]$$

$$+ \frac{(n-1)|A_{\max}|}{(1-\gamma)^2 K} \sum_{k=1}^{K} \sum_{i=1}^{n} \sum_{j \neq i} \mathbb{E}_{s \sim d^{y_{ij}^{\star,k}, (\pi_{j:n}^k \pi_{E,1:j})_{-i}}} \left[ \left\| \pi_{n-j+1}^k(s) - \pi_{E,n-j+1}(s) \right\|^2 \right] + \varepsilon_{\text{opt}}^{i,j}$$

$$= \frac{1}{K} \sum_{k=1}^{K} \sum_{i=1}^{n} \left( \frac{4n^2}{(1-\gamma)^2} \mathbb{E}_{s \sim d^{\pi^k}} \left[ \text{TV}^2\left( \pi_{E_i}(\cdot \mid s), \pi_i^k(\cdot \mid s) \right) \right] \right.$$

$$\left. + \frac{(n-1)|A_{\max}|}{(1-\gamma)^2} \sum_{j \neq i} \mathbb{E}_{s \sim d^{y_{ij}^{\star,k}, (\pi_{j:n}^k \pi_{E,1:j})_{-i}}} \left[ \left\| \pi_{n-j+1}^k(s) - \pi_{E,n-j+1}(s) \right\|^2 \right] + \varepsilon_{\text{opt}}^{i,j} \right)$$

$$\leq \frac{n^2 |A_{\max}|}{(1-\gamma)^2} \frac{1}{K} \sum_{k=1}^{K} \sum_{i=1}^{n} \left( \mathbb{E}_{s \sim d^{\pi^k}} \left[ \left\| \pi_{E_i}(s) - \pi_i^k(s) \right\|^2 \right] \right.$$

$$\left. + \sum_{j \neq i} \mathbb{E}_{s \sim d^{y_{ij}^{\star,k}, (\pi_{j:n}^k \pi_{E,1:j})_{-i}}} \left[ \left\| \pi_{n-j+1}^k(s) - \pi_{E,n-j+1}(s) \right\|^2 \right] + \varepsilon_{\text{opt}}^{i,j} \right),$$

where we used that $\text{TV}^2(\pi_{E_i}, \pi_i) \leq \frac{A_i}{4} \|\pi_{E_i} - \pi_i\|^2$.

Now note that we can sample according to all state occupancy measures. In particular, we have $S_k^{i,j} \sim d^{y_{ij}^{\star,k}, (\pi_{j:n}^k \pi_{E,1:j})_{-i}}$ and $S_k^{\pi^k} \sim d^{\pi^k}$. Again adding and subtracting $\left\| \pi_{E_i}(\cdot \mid S_k^{\pi^k}) - \pi^k(\cdot \mid S_k^{\pi^k}) \right\|^2$ and $\left\| \pi_{E_i}(\cdot \mid S_k^{i,j}) - \pi^k(\cdot \mid S_k^{i,j}) \right\|^2$ we get

$$\frac{n^2 \left|A_{\max}\right|}{(1-\gamma)^2} \frac{1}{K} \sum_{i=k}^{K} \sum_{i=1}^{n}$$

$$\left( \sum_{j\neq i}^{n} \mathbb{E}_{s \sim d^{y_{ij}^{\star,k}, (\pi_{j:n}^k \pi_{\mathrm{E},1:j})_{-i}}} \left[ \left\| \pi_{n-j+1}^k(\cdot \mid s) - \pi_{\mathrm{E},n-j+1}(\cdot \mid s) \right\|^2 \right] \right.$$

$$- \left\| \pi_{E,n-j+1}(\cdot \mid S_k^{i,j}) - \pi_{n-j+1}^k(\cdot \mid S_k^{i,j}) \right\|^2$$

$$+ \mathbb{E}_{s \sim d^{\pi^k}} \left[ \left\| \pi_{E_i}(\cdot \mid s) - \pi_i^k(\cdot \mid s) \right\|^2 \right] - \left\| \pi_{E_i}(\cdot \mid S_k^{\pi^k}) - \pi^k(\cdot \mid S_k^{\pi^k}) \right\|^2 \qquad \text{(Martingale)}$$

$$+ \sum_{j\neq i}^{n} \left\| \pi_{\mathrm{E},n-j+1}(\cdot \mid S_k^{i,j}) - \pi_{n-j+1}^k(\cdot \mid S_k^{i,j}) \right\|^2$$

$$\left. + \left\| \pi_{E_i}(\cdot \mid S_k^{\pi^k}) - \pi^k(\cdot \mid S_k^{\pi^k}) \right\|^2 \right) \qquad \text{(Regret)}$$

$$+ \frac{n^3 \left|A_{\max}\right|}{(1-\gamma)^2} \varepsilon_{\mathrm{opt}}$$

Note that now again, we have analogously to (12) a martingale term here and a regret term that we can control by updating the policies with online mirror descent. In particular, we can rearrange the regret term as follows

$$\text{(Regret)} = \sum_{i=1}^{n} \left( \sum_{j\neq n-i-1} \left\| \pi_{\mathrm{E}_i}(\cdot|S_k^{j,n-i+1}) - \pi_i^k(\cdot|S_k^{j,n-i+1}) \right\|^2 + \left\| \pi_{\mathrm{E}_i}(\cdot \mid S_k^{\pi^k}) - \pi^k(\cdot \mid S_k^{\pi^k}) \right\|^2 \right).$$

At this point, following an analogous analysis as done in proof of Theorem 4.2, we get a total bound in the order of $\mathcal{O}\big(\frac{\mathrm{poly}(n,|\mathcal{S}|,|A_{\max}|,(1-\gamma)^{-1})}{\varepsilon^8}\big)$. Similar steps can also be done for Algorithm 1 to obtain $\mathcal{O}\big(\frac{\mathrm{poly}(n,|\mathcal{S}|,|A_{\max}|,(1-\gamma)^{-1})}{\varepsilon^4}\big)$. This shows that the algorithm design indeed allows to avoid the **curse of multi-agents** and instead scales polynomial in the number of agents $n$.

## I  Comparison to Lower Bound in Tang et al.

In this section, we compare our result Theorem 3.2 to Theorem 4.3 in Tang et al. [2024]. In particular, we emphasize how their construction allows to avoid a linear regret in the case of a **fully known transition model**. Note that they consider a finite horizon setting and general-sum games with a correlated equilibrium expert. For a better readability we first restate their Theorem in a infinite horizon setting.

**Theorem I.1** (Theorem 4.3 in Tang et al. [2024]). *There exists a Markov Game, an expert policy pair* $(\mu^{\mathrm{E}}, \nu^{\mathrm{E}})$ *and a learner policy* $(\mu, \nu)$*, such that even when the state visitation distribution of* $(\mu, \nu)$ *exactly matches* $(\mu^{\mathrm{E}}, \nu^{\mathrm{E}})$*, the Nash gap satisfies*

$$\text{Nash-Gap}(\mu, \nu) \geq \Omega\left((1-\gamma)^{-1}\right).$$

The Markov Game that they construct is given in Fig. 3.

The Markov Game consists of the action space $\mathcal{A} = \{a_1, a_2, a_3\}$ and the state space $\mathcal{S} = \{s_0, s_1, s_2, s_3, s_4\}$. For the transition model, which is unknown to the learner, it holds true that

$$P(\cdot|s_0, a, b) = \begin{cases} s_1 & \text{if } (a,b) = a_2 a_1, \\ s_4 & \text{otherwise.} \end{cases}$$

and for all other states transition to one neighboring state with probability one, independent of the chosen action. The state-only reward of the cooperative Markov Game is given by

$$R_1(s) = \begin{cases} 1 & \text{if } s = s_3, \\ 0 & \text{otherwise.} \end{cases}$$

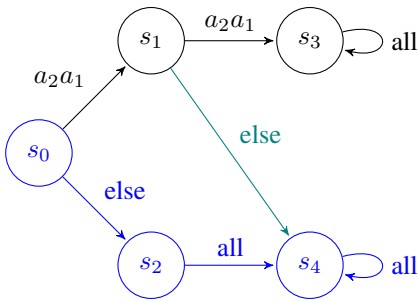

Figure 3: Cooperative Markov Game with Linear Regret in case of unknown transitions

It follows immediately, that an expert with a Nash-gap of $0$, i.e. an NE is given by the following policy pair

$$\mu^{\mathrm{E}}(a_1 \mid s_0) = \nu^{\mathrm{E}}(a_1 \mid s_0) = 1, \mu^{\mathrm{E}}(a_3 \mid s_1) = \nu^{\mathrm{E}}(a_3 \mid s_1) = 1,$$

and in the other states any action $a \in \mathcal{A}$ can be chosen. As the expert data is not covering data for state $s_1$, it only covers the blue path in Fig. 3, Tang et al. [2024] argue that any policy can be chosen for the learner in state $s_1$ and choosing $\mu(a_1 \mid s_1) = \nu(a_1 \mid s_1) = 1$. However, if we know the transition model, the learner can be steered to choose a *robust* action such that the learner will be taken back to states known from the expert data, i.e. state $s_4$, even when one agent would deviate from the current policy. This is highlighted in green in Fig. 3. The only action that is considered *robust* is action $a_3$ for both agents, exactly the action chosen from the expert. This implies that the learning policy in the case of a fully known transition model has

$$\mathrm{Nash\text{-}Gap}(\mu, \nu) = 0.$$

This in contrast to our construction in Fig. 1, where even under a known transition there is no way to steer the learner to known paths. Therefore, Theorem 3.2 shows the necessity of the expert agent deviation coefficient and separates MAIL form SAIL, where effective learning is possible under known transitions Rajaraman et al. [2020]. Additionally, we consider a Zero-Sum Markov Game, not considered in Tang et al. Tang et al. [2024].

## J   Experiments

In this section, we give a detailed description of the underlying environments used for the numerical validation of MURAIL and describe the setup in general. Additionally, we give some practical insights that could speed up convergence.

### J.1   Environments

We consider two different environments for our numerical validation, one that has $\mathcal{C}(\mu^{\mathrm{E}}, \nu^{\mathrm{E}}) < \infty$, and the lower bound construction Fig. 1 with different NE experts to control $\mathcal{C}(\mu^{\mathrm{E}}, \nu^{\mathrm{E}})$. In particular we have multiple with $\mathcal{C}(\mu^{\mathrm{E}}, \nu^{\mathrm{E}}) < \infty$ and the same NE expert as in Theorem 3.2 to get $\mathcal{C}(\mu^{\mathrm{E}}, \nu^{\mathrm{E}}) = \infty$.

**Environments with $\mathcal{C}(\mu^{\mathrm{E}}, \nu^{\mathrm{E}}) < \infty$.**   For this we consider two environments. For the first environment, we generate a random Zero-Sum Markov Game with $|\mathcal{S}| = 10, |\mathcal{A}| = |\mathcal{B}| = 3$ and a reward between $-1$ and $1$. To ensure that the expert covers all states we use a uniform initial state distribution, i.e $d_0(s_0) := \mathrm{Unif}(\mathcal{S})$. We set the discount factor to $0.9$.

Additionally, we choose the Markov Game from the Lower bound construction and use that the set of Nash equilibria is convex for Zero-sum Games. This way we take a mixture of Nash equilibria that chooses the $S_{\mathrm{copy}}$ path and the blue path, for a detailed description see Appendix J.2.

**Environment with $\mathcal{C}(\mu^{\mathrm{E}}, \nu^{\mathrm{E}}) = \infty$.**   For $\mathcal{C}(\mu^{\mathrm{E}}, \nu^{\mathrm{E}}) = \infty$, we use the Zero-Sum Markov Game given in Fig. 1 with the simplification that $|S_{\mathrm{xplt1}}| = |S_{\mathrm{xplt1}}| = |S_{\mathrm{copy}}| = 1$ as our goal here is only to verify the theoretical insights, but not to prove that also the transition model cannot be used for non-interactive Imitation Learning algorithms. This means that we have $|\mathcal{S}| = 7$. Additionally, we

have $|\mathcal{A}| = |\mathcal{B}| = 3$ and $d_0(s_0) = \delta_{s_0}$. We set the discount factor to $0.9$. The reward is given by $R(S_{\text{xplt2}})$, $R(S_{\text{xplt1}}) = -0.1$ and $0$ otherwise.

**Exploitability.**  To calculate the exploitability, we fix the current policies of one player iteratively and then run a standard Value Iteration for single-agent MDPs under the true underlying reward function.

## J.2 Experimental Setup

We run the experiments for each environments 1000 times over different seeds and average the results. For both environments we compute the optimal learning rate $\eta$. For simplicity, we use UCBVI algorithm for a state only reward as the RL inner loop of MURMAIL. Note, that this can be replaced by any other no regret algorithm.

**Expert distributions for different $\mathcal{C}(\mu^{\text{E}}, \nu^{\text{E}})$.**  To get control over $\mathcal{C}(\mu^{\text{E}}, \nu^{\text{E}})$, note that the set of Nash equilibria is convex for Two Player Zero-Sum Games. Therefore, consider the Lower bound example illustrated in Fig. 1. Note that we have a pure NE that chooses action $a_3 b_3$ to get on the blue path. Choosing a different pure action, let us assume $a_2, b_2$ will lead the agent to choose the path that goes on $s_1, S_{\text{copy}}$. Now, as the set of NE is convex, we can also mix these equilibria. To choose the minimal $\mathcal{C}(\mu^{\text{E}}, \nu^{\text{E}})$, we pick for $(a)$ the Nash equilibrium such that $\mu^{\text{E}}(a_2 \mid s_0) = \nu^{\text{E}}(a_2 \mid s_0) = \mu^{\text{E}}(a_3 \mid s_0) = \nu^{\text{E}}(a_3 \mid s_0) = 0.5$. To increase $\mathcal{C}(\mu^{\text{E}}, \nu^{\text{E}})$, we have to increase the probability of the experts to take action $a_3$ and $b_3$ respectively. We choose for $(c)$ $\mu^{\text{E}}(a_3 \mid s_0) = \nu^{\text{E}}(a_3 \mid s_0) = 0.999$, for $(c)$ we pick $\mu^{\text{E}}(a_3 \mid s_0) = \nu^{\text{E}}(a_3 \mid s_0) = 0.9999$. For $(d)$, we use the same expert policy as in the lower bound construction, i.e. $\mu^{\text{E}}(a_3 \mid s_0) = \nu^{\text{E}}(a_3 \mid s_0) = 1$.

To generate the expert distributions, we use a Value Iteration algorithm for Two Player Zero-Sum Games as e.g. described in Perolat et al. [2015] for the randomly generated Markov Game.

## J.3 Practical considerations

Next, we list practical considerations for our algorithms, that could speed up the performance. First, note that while solving the RL inner loop in Algorithm 2 can be computationally expensive, the objective between successive iterations changes only through the updates of the policies $\mu_k$ and $\nu_k$. Consequently, if these policies change only slightly between iterations, the optimal solutions for $y_k$ and $z_k$ may also vary only marginally. This observation suggests that initializing the optimization with the solution from the previous iteration, a common technique known as *warm-start optimization*, can significantly accelerate convergence.

Second, although the samples generated in the RL inner loop cannot formally be reused for the outer loop policy updates due to measurability issues of the resulting Martingale sequence, in practice it is often beneficial to recycle these samples. Doing so can reduce the total number of required samples without noticeably affecting empirical performance.

Last, note that we assumed for our analysis that there is no initial dataset for interactive Imitation Learning Section 2. However, in general it is possible to consider an initial dataset $\mathcal{D}$, from which we can learn initial policies with a non-interactive Imitation Learning algorithm like BC. This can speed up the convergence of our proposed algorithms as the maximum uncertainty exploration will mainly focus on states out of the distribution from the initial dataset. We give the algorithm of MURMAIL with an initial dataset in Algorithm 4. Similarly, one can adjust Algorithm 1.

## J.4 Additional plots

In this section, we list an additional plot for the second more involved environment that has $\mathcal{C}(\mu^{\text{E}}, \nu^{\text{E}}) < \infty$. Here we can observe similarly to case $(a)$ of Fig. 2 that the speed of convergence from BC is higher compared to MURMAIL. It indicates that the chosen algorithm has a small concentrability coefficient $\mathcal{C}(\mu^{\text{E}}, \nu^{\text{E}})$ and again highlights the importance of algorithm selection depending on the underlying environment.

# K Useful Results

In this section, we list useful theorems and lemmas used to prove the main results.

---

**Algorithm 4:** MURMAIL with initial dataset

---

**Input:** number of iterations $K$, learning rates $\eta$, inner iteration budget $T$, dataset $\mathcal{D}$,
      non-interactive Imitation Learning algorithm Alg

**Output:** $\varepsilon$-Nash equilibrium $(\hat{\mu}, \hat{\nu})$

% Run non-interactive Imitation Learning algorithm to initialize policies

$(\mu_1, \nu_1) = \text{Alg}(\mathcal{D})$

**for** $k = 1$ *to* $K$ **do**

    **Inner Single-Agent RL Updates:**

    % Maximum uncertainty response to $\mu$-player update

    Define single agent transition $P_{\mu_k}(s' \mid s, b) = \sum_{a \in \mathcal{A}} \mu_k(a \mid s) P(s' \mid s, a, b)$;

    Define single agent stochastic reward $R_{\mu_k}(s) \to \mathbb{1}_{\{A_E = A'_E\}} - 2\mu_k(A_E \mid s) + \|\mu_k(\cdot|s)\|^2$

      where $A_E, A'_E \sim \mu_E(\cdot \mid s)$;

    $y_k = \text{UCBVI}(T, P_{\mu_k}, R_{\mu_k})$;

    % Maximum uncertainty response to $\nu$-player update

    $P_{\nu_k}(s'|s, a) = \sum_{b \in \mathcal{B}} \nu_k(b|s) P(s' \mid s, a, b)$;

    $R_{\nu_k}(s) \to \mathbb{1}_{\{A_E = A'_E\}} - 2\nu_k(A_E \mid s) + \|\nu_k(\cdot \mid s)\|^2$ where $A_E, A'_E \sim \nu_E(\cdot \mid s)$;

    $z_k = \text{UCBVI}(T, P_{\nu_k}, R_{\nu_k})$

    **Update policies:**

    Sample $S_k^\mu \sim d^{\mu_k, y_k}$, $A_k^\mu \sim \mu_E(\cdot \mid S_k^\mu)$, $S_k^\nu \sim d^{z_k, \nu_k}$, $A_k^\nu \sim \nu_E(\cdot \mid S_k^\nu)$.

    $g_k^\mu(s, a) = \mu_k(a \mid S_k^\mu) \mathbb{1}_{S_k^\mu = s} - \mathbb{1}_{A_k^\mu = a}$

    $g_k^\nu(s, a) = \nu_k(a \mid S_k^\nu) \mathbb{1}_{S_k^\nu = s} - \mathbb{1}_{A_k^\nu = a}$

    $\mu_{k+1}(a \mid s) \propto \mu_k(a \mid s) \exp\left(-\eta g_k^\mu(s, a)\right)$;

    $\nu_{k+1}(b \mid s) \propto \nu_k(b \mid s) \exp\left(-\eta g_k^\nu(s, a)\right)$

**end**

**return** $\mu_{\hat{k}}$, $\nu_{\hat{k}}$ *for* $\hat{k} \sim \text{Unif}([K])$

---

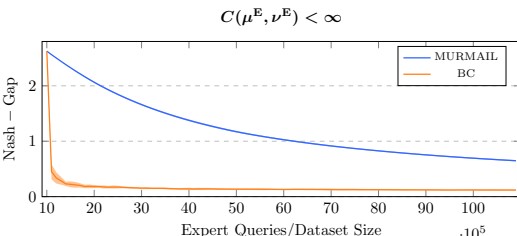

Figure 4: Nash Gap for MURMAIL and BC

**Lemma K.1** (see e.g. Lemma IX.5 by Alatur et al. [2024]). *For any policy of the max-player $\mu$ and two policies of the min-player $\nu$ and $\nu'$, we have*

$$V_1^{\mu, \nu}(s_0) - V_1^{\mu, \nu'}(s_0)$$
$$= \mathbb{E}_{\mu, \nu}\left[\sum_{t=0}^\infty \gamma^t \mathbb{E}_{(a,b) \sim (\mu, \nu)}\left[Q^{\mu, \nu'}(s, a, b)\right] - \mathbb{E}_{(a,b) \sim (\mu, \nu')}\left[Q^{\mu, \nu'}(s, a, b)\right]\right].$$

*Similarly, for any two policies of the max-player $\mu$ and $\hat{\mu}$ and policy of the min-player $\nu$, we have*

$$V_1^{\mu, \nu}(s_0) - V_1^{\hat{\mu}, \nu}(s_0)$$
$$= \mathbb{E}_{\mu, \nu}\left[\sum_{t=0}^\infty \gamma^t \mathbb{E}_{(a,b) \sim (\mu, \nu)}\left[Q^{\mu, \nu'}(s, a, b)\right] - \mathbb{E}_{(a,b) \sim (\mu, \nu')}\left[Q^{\mu, \nu'}(s, a, b)\right]\right].$$

*Proof.* The proof can seen as the two player case of the standard simulation lemma for MDPs as one player remains fixed. For completeness reasons we the first statement, the second one follows analogously. By the Bellman equation it holds true that

$$V_1^{\mu, \nu}(s) = \mathbb{E}_{a \sim \mu, b \sim \nu}\left[r(s, a, b) + \gamma \mathbb{E}_{s' \sim P}[V^{\mu, \nu}(s')]\right].$$

Applying this to the difference of the value functions yields

$$
\begin{aligned}
& V_1^{\mu,\nu}(s) - V_1^{\mu,\nu'}(s) \\
&= \mathbb{E}_{a\sim\mu,b\sim\nu}\left[r(s,a,b) + \gamma\mathbb{E}_{s'\sim P}[V^{\mu,\nu}(s')]\right] - \mathbb{E}_{a\sim\mu,b\sim\nu'}\left[r(s,a,b) + \gamma\mathbb{E}_{s'\sim P}[V^{\mu,\nu'}(s')]\right] \\
&= \left(\mathbb{E}_{a\sim\mu,b\sim\nu}\left[r(s,a,b) + \gamma\mathbb{E}_{s'\sim P}[V^{\mu,\nu}(s')] - \mathbb{E}_{a\sim\mu,b\sim\nu}[r(s,a,b) + \gamma\mathbb{E}_{s'\sim P}[V^{\mu,\nu'}(s')]]\right]\right) \\
&\quad + \left(\mathbb{E}_{a\sim\mu,b\sim\nu}[r(s,a,b) + \gamma\mathbb{E}_{s'\sim P}[V^{\mu,\nu'}(s')] - \mathbb{E}_{a\sim\mu,b\sim\nu'}\left[r(s,a,b) + \gamma\mathbb{E}_{s'\sim P}[V^{\mu,\nu'}(s')]\right]\right) \\
&= \gamma\mathbb{E}_{a\sim\mu,b\sim\nu}\left[\mathbb{E}_{s'\sim P}[V^{\mu,\nu}(s')] - \mathbb{E}_{s'\sim P}[V^{\mu,\nu'}(s')]\right] \\
&\quad + \left(\mathbb{E}_{a\sim\mu,b\sim\nu}[Q^{\mu,\nu'}(s,a,b)] - \mathbb{E}_{a\sim\mu,b\sim\nu'}\left[Q^{\mu,\nu'}(s,a,b))\right]\right),
\end{aligned}
$$

where we used that the immediate reward cancels out for $a\sim\mu, b\sim\nu$. Applying the same argument inductively for $s = s_0$ completes the proof. $\qquad\square$

**Lemma K.2** (Concentration Inequality for Total Variation Distance, see e.g. Thm 2.1 by Berend and Kontorovich [2012]). *Let $\mathcal{X} = \{1, 2, \cdots, |\mathcal{X}|\}$ be a finite set. Let $P$ be a distribution on $\mathcal{X}$. Furthermore, let $\widehat{P}$ be the empirical distribution given $m$ i.i.d. samples $x_1, x_2, \cdots, x_n$ from $P$, i.e.,*

$$
\widehat{P}(j) = \frac{1}{n}\sum_{i=1}^{n}\mathbb{I}\{x_i = j\}.
$$

*Then, with probability at least $1 - \delta$, we have that*

$$
\left\|P - \widehat{P}\right\|_1 := \sum_{x\in\mathcal{X}}\left|P(x) - \widehat{P}(x)\right| \leq \sqrt{\frac{2|\mathcal{X}|\log(1/\delta)}{n}}.
$$

*Proof.* Define the function $f(x_1, \ldots, x_n) = \sum_{x\in\mathcal{X}}|\widehat{P}(x) - P(x)|$, where $\widehat{P}$ is the empirical distribution. Replacing one sample $x_i$ can change $f$ by at most $2/n$, since the empirical frequencies change by at most $1/n$ per coordinate and total variation sums these differences.

By McDiarmid's inequality, we have for any $\varepsilon > 0$,

$$
\Pr\left(f - \mathbb{E}[f] \geq \varepsilon\right) \leq \exp\left(-\frac{n\varepsilon^2}{2}\right).
$$

Berend and Kontorovich (2013) show that $\mathbb{E}[f] \leq \sqrt{\frac{|\mathcal{X}|}{n}}$. Setting the failure probability to $\delta$, we solve

$$
\exp\left(-\frac{n\varepsilon^2}{2}\right) = \delta \quad\Longrightarrow\quad \varepsilon = \sqrt{\frac{2\log(1/\delta)}{n}}.
$$

Therefore, with probability at least $1 - \delta$,

$$
\left\|P - \widehat{P}\right\|_1 \leq \sqrt{\frac{|\mathcal{X}|}{n}} + \sqrt{\frac{2\log(1/\delta)}{n}} \leq \sqrt{\frac{2|\mathcal{X}|\log(1/\delta)}{n}},
$$

$\qquad\square$

**Lemma K.3** (Binomial concentration, see e.g. Lemma A.1 by Xie et al. [2021]). *Suppose $N \sim \text{Bin}(n, p)$ where $n \geq 1$ and $p \in [0, 1]$. Then with probability at least $1 - \delta$, we have*

$$
\frac{p}{N \vee 1} \leq \frac{8\log(1/\delta)}{n},
$$

*where $N \vee 1 := \max\{1, N\}$.*

*Proof.* We consider two cases. Case 1: $p \leq \frac{8 \log(1/\delta)}{n}$. As $N \vee 1 \geq 1$, we have $\frac{p}{N \vee 1} \leq p \leq \frac{8 \log(1/\delta)}{n}$ almost surely. Case 2: $p > \frac{8 \log(1/\delta)}{n}$. Note, that then $\mathbb{E}[N] = np > 8 \log(1/\delta)$ and by the multiplicative Chernoff bound, for any $0 < \varepsilon < 1$ it holds true that

$$\mathbb{P}\left(N < (1 - \varepsilon)np\right) \leq \exp\left(-\frac{\varepsilon^2}{2}np\right).$$

Now, with $\varepsilon = \frac{1}{2}$ we have

$$\mathbb{P}\left(N < (1 - \varepsilon)np\right) \leq \exp\left(-\frac{np}{8}\right) \leq \delta.$$

Therefore, with probability of at least $1 - \delta$ it holds $N \geq \frac{np}{2}$ and therefore on this event also $\frac{p}{n \vee 1} \leq \frac{2}{n}$. In total we get $\frac{p}{N \vee 1} \leq \frac{8 \log(1/\delta)}{n}$. Combining both cases completes the proof. □

