# OpenReview forum: "Learning Equilibria from Data: Provably Efficient Multi-Agent Imitation Learning"
_NeurIPS.cc/2025/Conference — NeurIPS 2025 poster_

### Official Review · Reviewer_JkAh · 2025-06-09

**Clarity:** 3
**Significance:** 3
**Originality:** 3
**Rating:** 5
**Confidence:** 2

**Summary:**

The paper deals with the setting of multi-agent RL where the agent tries to learn in a Markov game. In particular, the authors investigate the possibility of doing imitation learning in this setting.

One one hand, the paper derives a fundamental lower bound for the Behavioral Cloning algoritm. This shows that, even in the ideal scenario of inifite data and access to a generative model, there are environments where the learned policy can be exploited by the adversary. Therefore, it is not possible to learn an $\varepsilon-$Nash equilibrium.

Also the authors design two alsorithms, one that is oracle-efficient and the other that gets rid of the oracle at a cost of a much higher sample complexity. the oracel here is a Best Response Oracle (BRO), who is assumed to output the best response for a given strategy.

**Questions:**

See weaknesses

**Ethical Concerns:**

["NO or VERY MINOR ethics concerns only"]

**Final Justification:**

I confirm my initial opinion that this was a good paper.

**Limitations:**

Unfortunately, my very limited knowledge of the topic prevents me to understanding how groundbreaking this paper is w.r.t. the literature.

**Quality:**

3

**Strengths And Weaknesses:**

**strenghts**

The paper defines the Concentrability Coefficient $C$ and shows that this is a fundamental quantity to control the error of BC in this setting. To do this, the authors construct a Markov game in which no non-interactive algorithm, even with infinite data and full transition knowledge, can learn a near-Nash equilibrium, when $C$ is infinity. This result establishes a fundamental impossibility in offline MAIL, highlighting the need for interaction.

If I understand correctly, this is the main take-home message of the paper: in order to learn $\varepsilon-$Nash equilibria for arbitrarily small $\varepsilon$ from expert demonstrations, one must be able to interact with the expert to query demonstrations at specific points.

The negative result is completed by two positive result saying that 1) MAIL-BRO, which requires a best response oracle, achieves $\varepsilon-$Nash with O($\varepsilon^{-4}$) queries in Polynomial-time, with no dependence on concentrability 2) MURMAIL does the same without the need of an oracle at a price of an higher sample complexity.

**weaknesses**

The only weakness that I see is the bad dependence of the upper bounds for the sample complexity in both $\varepsilon$ and the constants. It is not clear if these results are optimal.

---

> ### Author Rebuttal · Authors · 2025-07-31
>
> We thank the reviewer for their time and thoughtful and positive feedback. We appreciate that the reviewer recognized the significance of our results, especially the impossibility of offline MAIL without interaction and the corresponding positive results for the interactive setting.
> Below, we address the reviewer's question regarding sample complexity.
>
>
> **Sample complexity**
>
> The reviewer correctly points out that the current upper bounds for sample complexity leave room for improvement, which we also acknowledge in our paper *“the focus of this work was to show the first sample complexity bound for a computationally efficient algorithm in the queriable expert setting.”*
> Below we would like to point out why it is non-trivial to improve the dependency on $\epsilon$ for MAIL-BRO and MURMAIL.
>
> *Non-convexity of objective*
>
> - The original problem is non-convex, therefore, we first have to change the objective to make it convex, which involves squaring it. This leads to $\epsilon^{-4}$ dependency for MAIL-BRO. Currently, we do not see a way to resolve this.
>
> *Missing knowledge of reward function*
>
>  - Removing the assumption of a best response oracle and introducing MURMAIL brings up new challenges. The key one is how to upper bound the expectation that involves the *best response policy*, despite not having access to the reward function. This rules out the use of standard optimistic estimators. To overcome this, we introduce the *maximum uncertainty response policy*. This new principle leads to a further degradation in sample complexity to $\epsilon^{-8}$.
>
> We believe this technique is novel and key to enable our guarantees in the context of Multi-Agent Imitation Learning, where no reward knowledge is assumed. That said, we agree with the reviewer that improved bounds might be achievable with new techniques, and we view this as a promising direction for future research.
>
> Once again, we thank the reviewer for their time and positive assessment of our paper. We hope this response clarifies the origin of the sample complexity bounds. We would be happy to address any further questions.
>
> Best,
>
> Authors

---

### Official Review · Reviewer_KjGo · 2025-07-02

**Clarity:** 4
**Significance:** 3
**Originality:** 3
**Rating:** 5
**Confidence:** 4

**Summary:**

The paper studies the problem of learning an approximate Nash Equilibrium (aNE) in two player, zero sum Markov games. The authors first study the non-interactive variation where the learners have access to samples from the game, but cannot simulate it. They prove that by using Behavior Cloning, they can compute an aNE, where the approximation factor depends on a novel coefficient, which is necessary to be bounded, otherwise obtaining the aNE is infeasible. The authors also provide positive results in the interactive setting; given best response oracles they provide an algorithm which finds an aNE with polynomial, in the parameters of the game, samples. Moreover, they provide an algorithm that computes an aNE without the need of Best Response oracles but with worse sample complexity.

**Questions:**

1. Is the single policy deviation concentrability coefficient a property of the game or the data trajectories? Can you explain more intuitively what having a very large coefficient means?

**Ethical Concerns:**

["NO or VERY MINOR ethics concerns only"]

**Final Justification:**

I do not have any modifications to my original score.

**Limitations:**

Yes

**Paper Formatting Concerns:**

No concerns.

**Quality:**

4

**Strengths And Weaknesses:**

Strengths:
1. The paper is well written, easy to read, and has nice illustrations that are intuitive.
2. The authors provide strong theoretical results that are coupled with experiments which verify them.
3. The authors provide a comprehensive study of finding equilibria from data in two player Markov Chains. They study both the interactive and non-interactive setting, and they provide efficient algorithms in terms of sample complexity that compute aNE in each case. In the non-interactive setting, they identify exactly the cases where learning an aNE is infeasible.

---

> ### Author Rebuttal · Authors · 2025-07-31
>
> We sincerely thank the reviewer for their thoughtful and positive evaluation. We are especially grateful for the kind comments that *“the paper is well written, easy to read, and has nice illustrations that are intuitive”* and that we *“provide strong theoretical results that are coupled with experiments which verify them”.* Please find below the answer to your questions.
>
> **Is the single policy deviation concentrability coefficient a property of the game or the data trajectories?**
>
> This is an excellent and important question. In the behavior cloning upper bound, the concentrability coefficient appears in terms of the estimated policies and, therefore, depends on the data.
> However, in our hardness result (Theorem 3.2), we pay for $\mathcal{C}(\mu_E,\nu_E)$, which depends only on the game structure and on the expert policies pair that collected the dataset.
> Importantly, this is a data and algorithm independent quantity that no non-interactive imitation learning algorithm can avoid.
>
> To summarize, the hardness result depends on a property of the game and of the particular expert policies pair that generates the dataset.
>
> **Intuition on concentrability coefficient**
>
> To build more intuition, consider the example used in our experiments:
>
>  - In the provided Markov Game, there exist two paths resulting from different pure strategies taken in $s_0$ with the same value.
> - The upper path is sensitive to deviations and therefore samples in the sensitive (worst case state) $s_1$ are required to recover a robust policy to deviations, i.e,. the Nash policy.
> - The concentrability coefficient can therefore intuitively be interpreted as the coverage of this worst-case state (the coverage is an upper bound on it). If the expert distribution does not cover this sensitive state, the coefficient can be infinite and if the observed equilibrium is a convex combination of both, then the concentrability coefficient decays with the convex coefficient for this upper exploitable path, reaching its minimum if both are covered equally (see Figure 2).
>
> We would be happy to clarify this further and add a more intuitive explanation to the camera-ready version to improve accessibility.
>
>  We thank the reviewer again for reading our paper and helping us to improve its quality. Happy to answer any further questions the reviewer might have.
>
> Best,
> Authors

---

> > ### Comment · Reviewer_KjGo · 2025-08-03
> >
> > I thank the authors for the response -- I have no further questions at this point.

---

### Official Review · Reviewer_sKo6 · 2025-07-02

**Clarity:** 4
**Significance:** 3
**Originality:** 3
**Rating:** 5
**Confidence:** 3

**Summary:**

The paper investigates a cutting-edge research topic, namely Multi-Agent Imitation Learning, for which theoretical results are currently scarce. It is mostly closely related to [Tang 2024] in the reference. The first result (Theorem 3.1) says that the Nash-Gap is close to zero if we use a behavior-cloning algorithm to learn the behavior very well, and a constant C(\hat{\mu},\hat{\nu}) is bounded. If the behavior are welled learned, C(\hat{\mu},\hat{\nu}) is expected to be close to C(\mu^E,\nu^E), where \mu^E and \nu^E are expert policies that achieve the Nash equilibrium. However, even if the nash equilibrium is unique, it may still be possible that C(\hat{\mu},\hat{\nu})=\infty, since fixing one player's policy other player may still have non-unique policy. This observation is formalized in Theorem 3.2, which says that there are examples of offline two-player games where the Nash gap is not vanishing even when the transition probability matrix is exactly known. This highlights an interesting difference between multi-agent and single-agent imitation learning, since for the latter it was known that behavior cloning can be optimal. Then Section 4 discusses alternative strategies and information that can be used to again close the gap.

**Questions:**

Just to make sure I understand the setup, the reason for the counterexample in Theorem 3.2 is that even if the algorithm knows the (action,state)-> state transition exactly, it does not know the reward function r, so that if the trajectory deviates from the equilibrium, the algorithm will not know how to act, is that correct?

**Ethical Concerns:**

["NO or VERY MINOR ethics concerns only"]

**Final Justification:**

The rebuttal confirmed my understandings and I believe that this is a clear accept.

**Limitations:**

The experimental results seem to be artificially constructed Markov chains. Is there any practical settings where the framework and theoretical results in this paper can be applied? Also, the present results are limited to the tabular settings, and there is a gap to the widely used deep learning models.

**Paper Formatting Concerns:**

No concern

**Quality:**

3

**Strengths And Weaknesses:**

This paper is very well written and provides necessary background, literature review, and proof sketch for people who are not intimately familiar with the topic. I haven't checked the proofs, but the intuitions seem to make sense. The paper only deals with the tabular setting and some of the results are expected, so I suppose that the proofs are not very technical sophisticated/challenging.

---

> ### Author Rebuttal · Authors · 2025-07-31
>
> We sincerely thank the reviewer for their insightful and encouraging review. We are especially grateful for the recognition that our paper is *“very well written and provides necessary background, literature review, and proof sketch for people who are not intimately familiar with the topic.”* Please find some comments on your questions below.
>
> **Theoretical contributions**
>
> We appreciate the reviewer’s assessment of our theoretical contributions. While the high-level ideas may seem intuitive, we would like to highlight that our analysis includes several technically non-trivial components.
> - In particular, our analysis tackles the non-convex optimization problem of interactive Imitation Learning: $$\min_{\widehat{\mu} \in \Pi} \max_{\nu \in \mathrm{br}(\widehat{\mu})} \mathbb{E}_{(\mu_E, \nu)} \left[ \sum\_{t=0}^{\infty}\gamma^t , \mathrm{TV}(\mu_E(\cdot \mid s), \widehat{\mu}(\cdot \mid s)) \right].$$
> The key difficulty lies in the fact that  the data  generation process depends on the minimizer itself $\widehat{\mu}$. To be precise, the data is generated from the occupancy measure of $(\mu_E, \nu)$ and $\nu$ is one of the best responses to the minimizer  $\widehat{\mu}$.
> This makes the minimization too hard to be tackled in a computationally efficient manner.
> To bypass these problems, we derived an upper bound on the optimization problem above (Equation 4 in the paper), which is easy to optimize leveraging a Best Response Oracle. For more details, see lines 230 to 237 of our paper.
> Additionally, when the reward function is not known and a best response oracle is not available, we introduce the UCBVI inner loop to compute the maximum uncertainty response, which is by itself a new principle that could find further applications.
>
> These new techniques allow us to provide a computationally efficient algorithm with sample complexity guarantees for MAIL, which was an important open problem.
>
> **Theorem 3.2**
>
> - Regarding Theorem 3.2, the reviewer correctly understood the issue: without knowledge of the reward function, the learner cannot be guided to recover a good policy. We tried to give some more intuition why knowing the transition dynamics is not sufficient in Appendix E(l.725-726) of the paper: *“Additionally, note that even if the learner has access to the transition dynamics, the learner can not differentiate the actions from s1 as all actions lead to different states. Therefore, she cannot use this knowledge to recover an action that would lead to [a good state]”.* As the reviewer has identified correctly, this is due to the missing reward function.
>
> **Practical applications**
>
> - A practical example could be route navigation recommendations. In this example, minimizing the Nash gap ensures robustness of route recommendations against individuals that may deviate from recommendations to increase their personal utility to e.g. to avoid traffic jams.
>
> **Tabular setting and deep imitation learning**
>
> - The reviewer has raised a valuable point regarding the limitation of the tabular setting. Since this is the first work that gives a sample-efficient imitation learning algorithm, we leave the extension to function approximation as future work (see Appendix C).
> - As stated in lines 609-613, we believe that the same algorithmical framework presented in this work can carry over to deep settings and give insight into the main steps that need to be adjusted: *“..., the main conceptual ideas easily carry on to deep imitation learning experiments. The largest theory-practice gap would be in the inner loop where UCBVI would need to be replaced by a Deep RL algorithm..”.* To summarize, while our theoretical analysis may be limited to the tabular case, the ideas carry over to more general applied settings.
>
> We hope this response has addressed the reviewer’s thoughtful questions and clarified the theoretical depth and practical relevance of our work. We are happy to incorporate any additional suggestions in the final version and thank the reviewer again for their constructive review.
>
>
> Best,
>
> Authors

---

### Official Review · Reviewer_8iNH · 2025-07-05

**Clarity:** 3
**Significance:** 3
**Originality:** 3
**Rating:** 5
**Confidence:** 3

**Summary:**

This work concerns multi-agent imitation learning and, more specifically, learning equilibria of zero-sum Markov games (MG) using a dataset of trajectories. I.e., an algorithm is provided with a number of trajectories sampled from a Nash equilibrium, and it needs to output a policy for each agent s.t. they jointly form a Nash equilibrium. The authors further demonstrate the impossibility of this task when the dataset does not satisfy a rather strict condition. To circumvent this, the authors allow *interactive* imitation learning, meaning that the algorithms is allowed to query the expert (who collected the dataset in the first place) for further demonstrations. In the latter setting, the authors manage to offer two algorithmic solutions for the task of equilibrium learning from data with only polynomial sample complexity. Further, behavior cloning and its shortcomings is extensively studied in this paper. Finally, the authors provide some experimental verification of their theoretical claims.

Behavior cloning is a common heuristic to tackle the problem in single-agent Markov decision processes (and also a previous multi-agent work Tang et al 2024). In behavior cloning, one needs to compute a maximum likelihood estimate of the two player policies for that particular dataset of trajectories. Under a bounded concentration coefficient, this works well and the authors prove this in Theorem 3.1.

The contributions further include:
* a clear separation between single-agent and multi-agent imitation learning without interactions (Theorem 3.2). The authors construct a MG where behavior cloning will output a pair of policies with exploitability lower bounded by a constant of the game.

* using an interactive expert assumption to design two algorithms for Nash equilibrium learning. The first algorithm assumes access to a policy best-response oracle and the second one circumvents by tweaking the first algorithm.


In Algorithm 1,
* at every iteration $k$, the best-response oracle is queried for a best response to minimizer's policy $\mu_k$ and maximizer's policy $\nu_k$.

* then, the algorithm queries the expert to provide trajectory samples.

* the samples are used to construct a feedback vector which will in turn be used with a policy optimization update (exponential weight updates).

* the algorithm samples a joint strategy from all iteration and outputs it.

The guarantee is that in expectation, the output is an approximate Nash equilibrium. (Theorem 4.1)


Algorithm 2 essentially goes beyond the need of a best response oracle by using an upper confidence bound value iteration subroutine. This extension requires non-trivial technical work.

**Questions:**

* Can you reduce MAIL to a (potentially) general-utility Markov decision process? (single agent) What are the connections to convex MDPs or convex Markov games? (Gemp et al 2024)

* In algorithm 2 for uncertain states, why did you not pick the KL divergence between $\mu_E$ and $\mu_k$ (lines 266-271)?

* Would you be able to extend your results to games with some low rank interaction assumption, e.g. (Kalogiannis and Panageas 2023, Park et al 2023, Zhan et al 2024)?

* Does the uniqueness of the equilibrium of zero-sum games play an crucial role in your results so much so that it would not extend your results as asked by previous question?

* What would be some further benchmarks for this setting?



Gemp, I., Haupt, A.A., Marris, L., Liu, S. and Piliouras, G., Convex Markov Games: A New Frontier for Multi-Agent Reinforcement Learning

Zhan, W., Fujimoto, S., Zhu, Z., Lee, J.D., Jiang, D.R. and Efroni, Y., 2024. Exploiting Structure in Offline Multi-Agent RL: The Benefits of Low Interaction Rank.

Park, C., Zhang, K. and Ozdaglar, A., 2023. Multi-player zero-sum markov games with networked separable interactions.

Kalogiannis, F. and Panageas, I., 2023. Zero-sum polymatrix markov games: Equilibrium collapse and efficient computation of nash equilibria

**Ethical Concerns:**

["NO or VERY MINOR ethics concerns only"]

**Limitations:**

yes

**Quality:**

3

**Strengths And Weaknesses:**

Among the strengths of this paper are its clear-cut improvement from previous work (Tang et al 2024). They set apart single-agent and multi-agent imitation learning with a clean and simple MG example. The separation is robust to even the strong assumption of knowledge of the game dynamics; the corresponding result by (Tang et al 2024) was not as strong as knowledge of the dynamics was enough to ameliorate things.

Having proven the impossibility of learning a policy pair whose exploitability does not depend on the *single policy deviation concentrability coefficient*, they consider the setting of interactive imitation learning. Then, they provide a strong algorithmic solution to the problem of learning a Nash equilibrium. The algorithm and its guarantees' proof make use of a broad array of the theoretical toolbox of RL and learning theory.

To the paper's weaknesses, I would list, the weak empirical evaluation without any code. There is no benchmark. Further, the setting the authors consider is only tabular MGs and they did not offer any function approximation guarantees.

---

> ### Author Rebuttal · Authors · 2025-07-31
>
> We thank the reviewer for their thoughtful, detailed, and positive feedback, and for recognizing that our work presents a *“clear-cut improvement from previous work”* and  *“the algorithm and its guarantees' proof make use of a broad array of the theoretical toolbox of RL and learning theory.”*
>
> In the following, we would like to address the different questions and few mentioned weaknesses raised by the reviewer:
>
> **Benchmarking**
>
> In the current submission, we focus on providing theoretical guarantees, leaving more practical extension for future work.  Our numerical simulation on the same Markov Game used for the hardness result in Theorem 3.2 were thought to verify our theoretical results, in particular showing that the degradation of BC under high concentrability is a real effect.
> Moving forward, we plan to explore experiments in route recommendation, inspired by the motivating example in [1]. Importantly, in this setting, a low Nash gap ensures that no agent (car driver) has benefits in taking a different route.
>
> Other interesting experiments can be performed in recent libraries such as BenchMARL [2].
> However, the design of Deep MAIL experiments requires care in estimating how the concentrability coefficient behaves in practical environments and in computing the Nash Gap for evaluation purposes.
>
> **Relation with Convex Markov Games**
>
> We see a twofold relation with the work of Gemp et al. 2024, which we outline below:
>
> - *Imitation as Convex Markov Game:* First, notice that in their section 5.2, they use the convex formulation for imitation learning. However, they use this technique in presence of reward for equilibrium selection purposes. When one sets the reward to zero in their formulation, we are left with an occupancy measure matching problem which is akin to minimize the value gap according to (Tang et al 2024) terminology. Therefore, their technique does not seem to be suitable to learn equilibria from data in presence of high concentrability coefficient.
> - *Imitating equilibria in Convex Markov Game:* As a final connection, we point out that one might study the question of learning equilibria from data in convex markov games which is a more general setting than the markov games studied in our work. If the markov game functional has G-bounded gradients our techniques still apply via a standard reduction from convex to linear losses in online learning. However, studying imitation in convex markov games with unbounded gradients is an interesting open question.
>
>
> **Extension to Low-rank interactions games**
>
> The low rank interaction assumption might help in bypassing the concentrability coefficient.
> In particular, if the norm of the difference between the learnt and expert policy is a low interaction rank function. It is, in our opinion, unclear under which conditions this can be proven. Moreover, notice that in [4] this assumption allows the authors to avoid the exponential dependency on the number of players in Offline MAIL. Under our setting, the dependence is polynomial even without the low rank assumption, see Remark 4.1.
>
> **KL divergence in algorithm 2**
>
> We did not use the KL divergence in Algorithm 2 because our analysis relies on the Cauchy–Schwarz inequality (see Equation 8 in Appendix G), which naturally leads to a squared norm. While one could derive an upper bound on Equation 8 using the strong convexity of KL and formulate updates that minimize this bound, doing so would likely result in suboptimal performance. Specifically, such an alternative would optimize a possibly looser bound than what Murmail currently minimizes (Equation 8).
>
> **Uniqueness of equilibria**
>
> Our analysis does not rely on a uniqueness assumption for equilibria. In Zero-Sum Games, the value is unique, but not the strategy. Under the uniqueness assumption, the lower bound construction is no longer valid, and the concentrability coefficient could be avoided. This is an interesting open question. Moreover, our theorems can be extended to the general-sum Markov Game setting, where not even the uniqueness of equilibrium value holds true as stated in Remark 4.1 and sketched in Appendix I.
>
> We hope these responses clarify the raised points. We are grateful for the reviewer’s detailed and constructive feedback and would be happy to incorporate any further suggestions in the camera-ready version.
>
> We would be glad to further clarify any additional points the reviewer may have.
>
> Best,
> Authors
>
> [1] J. Tang, G. Swamy, F. Fang, and Z. S. Wu. Multi-agent imitation learning: Value is easy, regret is hard. In A. Globerson, L. Mackey, D. Belgrave, A. Fan, U. Paquet, J. Tomczak, and C. Zhang, NIPS2024
>
> [2] Matteo Bettini, Amanda Prorok, and Vincent Moens. 2023. BenchMARL: Benchmarking Multi-Agent Reinforcement Learning.
>
> [3] Gemp, I., Haupt, A.A., Marris, L., Liu, S. and Piliouras, G., Convex Markov Games: A New Frontier for Multi-Agent Reinforcement Learning
>
> [4] Zhan, W., Fujimoto, S., Zhu, Z., Lee, J.D., Jiang, D.R. and Efroni, Y., 2024. Exploiting Structure in Offline Multi-Agent RL: The Benefits of Low Interaction Rank.

---

> > ### Comment · Reviewer_8iNH · 2025-08-05
> > **Thanks**
> >
> > Thank you for answering my questions. I will keep my positive score recommending acceptance.

---

### Note · Authors · 2025-08-12

We would like to thank the reviewers for taking the time to read our paper and for their valuable feedback, which helped us improve it. We are glad that our rebuttal addressed the remaining questions.

We appreciate the reviewers’ recognition of our contributions, particularly that *“the paper is well written, easy to read, and has nice illustrations that are intuitive.”* We are also grateful for their acknowledgment of the theoretical foundations we provide, including the first sample complexity guarantees for Multi-Agent Imitation Learning, which required *"use of a broad array of the theoretical toolbox of RL and learning theory."*

We believe this paper lays an important theoretical foundation and offers practical insights for future research in Multi-Agent Imitation Learning.

---

### Decision · Program_Chairs · 2025-09-17

**Decision:**

Accept (poster)

**Comment:**

The paper provides the first sample complexity bounds for learning Nash equilibria from expert data in Markov Games, introducing the critical single policy deviation concentrability coefficient. It shows behavioral cloning's (BC's) limitations in high concentrability settings and proposes two algorithms, MAIL-BRO, and MURMAIL, which are validated both theoretically and, to a lesser extent, also numerically. Strengths include clearly advancing current state-of-the-art work, robust RL theory, and useful illustrations. The main weakness is the restriction to tabular settings also in the empirical part. Four reviewers supported acceptance, praising theory and clarity, with authors addressing some issues are the concept of concentrability and the applicability of the work. The only remaining major concern after the panel discussion is the limited benchmarking that would allow more direct comparisons with prior work.

Overall, all reviewers engaged in discussion with authors ensuring a thorough and high-quality review process. On these grounds, I recommend acceptance. Nevertheless, I still strongly encourage the authors to implement the minor revisions that resulted from the discussion with the reviewers for the camera-ready version to further strengthen their contribution.